# Sampling Multimodal Distributions with the Vanilla Score: Benefits of Data-Based Initialization

**Frederic Koehler**
Department of Statistics and Data Science Institute
University of Chicago
Chicago, IL 60637
`fkoehler@uchicago.edu`

**Thuy-Duong Vuong**
Department of Computer Science
Stanford University
Stanford, CA 94305
`tdvuong@stanford.edu`

## Abstract

There is a long history, as well as a recent explosion of interest, in statistical and generative modeling approaches based on *score functions* — derivatives of the log-likelihood of a distribution. In seminal works, Hyvärinen proposed vanilla score matching as a way to learn distributions from data by computing an estimate of the score function of the underlying ground truth, and established connections between this method and established techniques like Contrastive Divergence and Pseudolikelihood estimation. It is by now well-known that vanilla score matching has significant difficulties learning multimodal distributions. Although there are various ways to overcome this difficulty, the following question has remained unanswered — is there a natural way to sample multimodal distributions using just the vanilla score? Inspired by a long line of related experimental works, we prove that the Langevin diffusion with early stopping, initialized at the empirical distribution, and run on a score function estimated from data successfully generates natural multimodal distributions (mixtures of log-concave distributions).

## 1 Introduction

Score matching is a fundamental approach to generative modeling which proceeds by attempting to learn the gradient of the log-likelihood of the ground truth distribution from samples ("score function") Hyvärinen (2005). This is an elegant approach to learning *energy-based models* from data, since it circumvents the need to compute the (potentially intractable) partition function which arises in Maximum Likelihood Estimation (MLE). Besides the original version of the score matching method (often referred to as *vanilla score matching*), many variants have been proposed and have seen dramatic experimental success in generative modeling, especially in the visual domain (see e.g. Song & Ermon (2019); Song et al. (2020b); Rombach et al. (2022)).

In this work, we revisit the vanilla score matching approach. It is known that learning a distribution via vanilla score matching generally fails in the multimodal setting (Wenliang et al., 2019; Song & Ermon, 2019; Koehler et al., 2022). However, there are also many positive aspects of modeling a distribution with the vanilla score. To name a few:

1. Simplicity to fit: computing the best estimate to the vanilla score is easy in many situations. For example, there is a simple closed form solution the class of models being fit is an exponential family (Hyvärinen, 2007b), and this in turn lets us compute the best fit in a kernel exponential family (see e.g. Sriperumbudur et al. (2017); Wenliang et al. (2019)).

2. Compatibility with energy-based models: for a distribution $p(x) \propto \exp(E(x))$, the vanilla score function is $\nabla E(x)$ so it is straightforward to go between the energy and the score function. This is related to the previous point (why exponential families are simple to score match), and also why it is easy to implement the Langevin chain for sampling an energy-based model.

3. Statistical inference: in cases where vanilla score matching does work well, it comes with attractive statistical features like $\sqrt{n}$-consistency, asymptotic normality, relative efficiency

guarantees compared to the MLE, etc. — see e.g. Barp et al. (2019); Forbes & Lauritzen (2015); Koehler et al. (2022); Song et al. (2020a).

In addition, score matching is also closely related to other celebrated methods for fitting distributions which have been successfully used for a long time in statistics and machine learning — pseudo-likelihood estimation (Besag, 1975) and contrastive divergence training (Hinton, 2002). (See e.g. Hyvärinen (2007a); Koehler et al. (2022).)

For these reasons, we would like to better understand the apparent failure of score matching in the multimodal setting. In this work, we study score matching in the context of the most canonical family of multimodal distributions — mixtures of log-concave distributions. (As a reminder, any distribution can be approximated by a sufficiently large mixture, see e.g. Wasserman (2006).) While vanilla score matching itself does not correctly estimate these distributions, we show that the trick of using "data-based initialization" when sampling, which is well-known in the context of CD/MLE training of energy based models (see e.g. Hinton (2012); Xie et al. (2016) and further references below), provably corrects the bias of any model which accurately score matches the ground truth distribution.

## 1.1 OUR RESULTS

We now state our results in full detail. We are interested in the question of generative modeling using the vanilla score function. Generally speaking, there is some ground truth distribution $\mu$, which for us we will assume is a mixture of log-concave distributions, and we are interested in outputing a good estimate $\hat{\mu}$ of it. We show that this is possible provided access to:

1. A good estimate of the score function of $\nabla \log \mu$. (In many applications, this would be learned from data using a procedure like score matching.)
2. A small number of additional samples from $\mu$, which are used for data-based initialization.

To make the above points precise, the following is our model assumption on $\mu$:

**Assumption 1.** *We assume probability distribution $\mu$ is a mixture of $K$ log-concave components: explicitly, $\mu = \sum_{i=1}^{K} p_i \mu_i$ for some weights $p_1, \ldots, p_K$ s.t. $p_i > 0$ and $\sum_i p_i = 1$. Furthermore, we suppose the density of each component $\mu_i$ is $\alpha$ strongly-log-concave and $\beta$-smooth with $\beta \geq 1$[1] i.e. $\alpha I \preceq -\nabla^2 \log \mu_i(x) \preceq \beta I$ for all $x$. We define the notation $p_* = \min_i p_i$ and $\kappa = \beta/\alpha \geq 1$.*

**Remark 1.** *The assumption that $\mu_i$ is $\alpha$-strongly log-concave and $\beta$-smooth is the most standard setting where the Langevin dynamics are guaranteed to mix rapidly (see e.g. Dalalyan (2017)).*

and the following captures formally what we mean by a "good estimate" of the score function:

**Definition 1.** *For $\mu$ a probability distribution with smooth density $\mu(x)$, an $\epsilon_{score}$-accurate estimate of the score in $L_2(\mu)$ is a function $s$ such that*

$$\mathbb{E}_{x \sim \mu}[||s(x) - \nabla \log \mu(x)||^2] \leq \epsilon_{score}^2. \tag{1}$$

As discussed in the below remark, this is the standard and appropriate assumption to make when score functions are learned from data. There are also other settings of interest where the ground truth score function is known exactly (e.g. $\mu$ is an explicit energy-based model which we have access to, and we want to generate more samples from it[2]) in which case we can simply take $\epsilon_{\text{score}} = 0$.

**Remark 2.** *Assumption (1) says that on average over a fresh sample from the distribution, $s(x)$ is a good estimate of the true score function $\nabla \log \mu(x)$. This is the right assumption when score functions are estimated from data, because it is generally impossible to learn the score function far from the support of the true distribution. See the previous work e.g. Chen et al. (2023); Lee et al. (2022a;b); Block et al. (2020) where the same distinction is discussed in more detail.*

*Given a class of functions which contains a good model for the true score function and has a small Rademacher complexity compared to the number of samples, the function output by vanilla score matching will achieve small $L_2$ error (see proof of Theorem 1 of Koehler et al. (2022)). In*

---

[1]We can always re-scale the domain so that $\beta \geq 1$.

[2]For example, one use case of generative modeling is when we have the ground truth and want to accelerate an existing sampler which is expensive to run, see e.g. Albergo et al. (2021); Lawrence & Yamauchi (2021).

*particular, this can be straightforwardly applied to parametric families of distributions like mixtures of Gaussians. We would also generally expect this assumption to be satisfied when the distribution is successfully learned via other learning procedures, such as MLE/contrastive divergence. (See related simulation in Appendix I.)*

We show the distribution output by Langevin dynamics on an approximate score function will be close to the ground truth provided (1) we initialize the Langevin diffusion from the empirical distribution of samples, and (2) we perform early stopping of the diffusion, so that it does not reach its stationary distribution. Formally, let the *Langevin Monte Carlo* (LMC, a.k.a. discrete-time Langevin dynamics) chain with initial state $X_0$, score function $s$, and step size $h > 0$ be defined by the recursion

$$X_{h(i+1)} = X_{hi} + h\, s(X_{hi}) + \sqrt{2h}\, \Delta_{hi}$$

where each noise variable $\Delta_{hi} \sim N(0, I)$ is independent of the previous ones. Our main result gives a guarantee for sampling with LMC started from a small set of samples and run for time $T$:

**Theorem 1.** *Let $\epsilon_{TV} \in (0, 1/2)$. Suppose $\mu$ is a mixture of strongly log-concave measures as in Assumption 1 and $s$ is a function which estimates the score of $\mu$ within $L_2$ error $\epsilon_{score}$ in the sense of Definition 1. Let*

$$T = \tilde{\Theta}\left(\left(\frac{\exp(K)d\kappa}{p_*\epsilon_{TV}}\right)^{O_K(1)}\right), \qquad h = \tilde{\Theta}\left(\frac{\epsilon_{TV}^4}{(\beta\kappa^2 K \exp(K))^4 d^3 T}\right).$$

*Let $U_{sample}$ be a set of $M$ i.i.d. samples from $\mu$ and $\nu_{sample}$ be the uniform distribution over $U_{sample}$. Suppose that $M = \Omega(p_*^{-2}\epsilon_{TV}^{-4} K^4 \log(K/\epsilon_{TV}) \log(K/\tau))$, and that*

$$\epsilon_{score} \leq \frac{p_*^{1/2}\sqrt{h}\epsilon_{TV}^2}{7T} = \tilde{\Theta}\left(\frac{p_*^{1/2}\epsilon_{TV}^4}{(\beta\kappa^2 K \exp(K))^2 d^{3/2}T^{3/2}}\right).$$

*Let $(X_{nh}^{\nu_{sample}})_{n\in\mathbb{N}}$ be the LMC chain with score $s$ and step size $h$ initialized at $\nu_{sample}$. Then with probability at least $1-\tau$ over the randomness of $U_{sample}$, the conditional law $\hat{\mu} = \mathcal{L}(X_T^{\nu_{sample}} \mid U_{sample})$ satisfies*

$$d_{TV}(\hat{\mu}, \mu) \leq \epsilon_{TV}. \tag{2}$$

We now make a few comments to discuss the meaning of the result. Conclusion (2) says that we have successfully found an $\epsilon_{TV}$-close approximation of the ground truth distribution $\mu$. Unpacking the definitions, it says that with high probability over the sample set: (1) picking a uniform sample from the training set, and (2) running the Langevin chain for time $T$ will generate an $\epsilon_{TV}$-approximate sample from the distribution $\mu$. Note in particular that we can draw as many samples as we like from the distribution without needing new training data. The fact that this is *conditional on the dataset* is a key distinction: the *marginal* law of any element of the training set would be $\mu$, but its *conditional* law is a delta-distribution at that training sample, and the conditional law is what is relevant for generative modeling (being able to draw new samples from the right distribution). See also Figure 1 for a simulation which helps illustrate this distinction.

**Remark 3.** *Provided the number of components in the mixture is $O(1)$, i.e. upper bounded by a constant, the dependence on all other parameters is polynomial or logarithmic. It is possible to remove the dependence on the minimum weight $p_*$ completely — see Corollary 2 in Appendix H.*

**Remark 4.** *It turns out Theorem 1 is a new result even in the very special case that the ground truth is unimodal. The closest prior work is Theorem 2.1 of Lee et al. (2022a), where it was proved that the Langevin diffusion computed using an approximate score function succeeds to approximately sample from the correct distribution given a (polynomially-)warm start in the $\chi_2^2$-divergence. However, while the empirical distribution of samples is a natural candidate for a warm start, in high dimensions it will not be anywhere close to the ground truth distribution unless we have an exponentially large (in the dimension) number of samples, due to the "curse of dimensionality", see e.g. Wasserman (2006).*

## 1.2 FURTHER DISCUSSION

**One motivation: computing score functions at substantial noise levels can be computationally difficult.** In some cases, computing/learning the vanilla score may be a substantially easier task than

alternatives; for example, compared to learning the score function for all noised versions of the ground truth (as used in diffusion models like Song & Ermon (2019)). As a reminder, denoising diffusion models are based on the observation that the score function of a noised distribution $N(0, \sigma^2 I) \star p$ exactly corresponds to a Bayesian denoising problem: computing the posterior mean on $X \sim p$ given a noisy observation $Y \sim N(x, \sigma^2 I)$ Vincent (2011); Block et al. (2020), via the equation

$$y + \sigma^2 \nabla \log(N(0, \sigma^2 I) \star p)(y) = \mathbb{E}[X \mid Y = y].$$

Unlike the vanilla score function this will not be closed form for most energy-based models; the optimal denoiser might be complex when the signal is immersed in a substantive amount of noise.

For example, results in the area of computational-statistical gaps tell us that for certain values of the noise level $\sigma$ and relatively simple distributions $p$, approximate denoising can be average-case computationally hard under widely-believed conjectures. For example, let $p$ be a distribution over matrices of the form $N(rr^T, \epsilon^2)$ with $r$ a random sparse vector and $\epsilon > 0$ small. Then the denoising problem for this distribution will be "estimation in the sparse spiked Wigner model". In this model, for a certain range of noise levels $\sigma$ performing optimal denoising is as hard as the (conjecturally intractible) "Planted Clique" problem (Brennan et al., 2018); in fact, even distinguishing this model from a pure noise model with $r = 0$ is computationally hard despite the fact it is statistically possible — see the reference for details. So unless the Planted Clique conjecture is false, there is no hope of approximately computing the score function of $p \star N(0, \sigma^2)$ for these values of $\sigma$. On the other hand, there is no computational obstacle to computing the score of $p$ itself provided $\epsilon > 0$ is small — denoising is only tricky once the noise level becomes sufficiently large.

**Related Experimental Work.** As mentioned before, many experimental works have found success generating samples, especially of images, by running the Langevin diffusion (or other Markov chain) for a small amount of time. One aspect which varies in these works is how the diffusion is initialized. To use the terminology of Nijkamp et al. (2020), the method we study uses an *informative/data-based initialization* similar to contrastive divergence Hinton (2012); Gao et al. (2018); Xie et al. (2016). While in CD the early stopping of the dynamics is usually motivated as a way to save computational resources, the idea that stopping the sampler early can improve the quality of samples is consistent with experimental findings in the literature on energy-based models. As the authors of Nijkamp et al. (2020) say, "it is much harder to train a ConvNet potential to learn a steady-state over realistic images. To our knowledge, long-run MCMC samples of all previous models lose the realism of short-run samples." One possible intuition for the benefit of early stopping, consistent with our analysis and simulations, is that it reduces the risk of stepping into low-probability regions where the score function may be poorly estimated. Some works have also found success using random/uninformative initializations with appropriate tweaks (Nijkamp et al., 2019; 2020), although they still found informative initialization to have some advantages — for example in terms of output quality after larger numbers of MCMC steps. Finally, we recall that the success of many recent experimental works which fit score functions with neural networks (e.g. Song & Ermon (2019); Song et al. (2020b); Rombach et al. (2022); Ho et al. (2020)).

**Related Theoretical Work.** The works Block et al. (2020); Lee et al. (2022a) established results for learning unimodal distributions (in the sense of being strongly log-concave or satisfying a log-Sobolev inequality) via score matching, provided the score functions are estimated in an $L_2$ sense. The work Koehler et al. (2022) showed that the sample complexity of vanilla score matching is related to the size of a restricted version of the log-Sobolev constant of the distribution, and in particular proved negative results for vanilla score matching in many multimodal settings. The works Lee et al. (2022b); Chen et al. (2023) proved that even for multimodal distributions, *annealed* score matching will successfully learn the distribution provided all of the annealed score functions can be successfully estimated in $L_2$. In our work we only assume access to a good estimate of the vanilla score function, but still successfully learn the ground truth distribution in a multimodal setting.

In the sampling literature, our result can be thought of establishing a type of *metastability* statement, where the dynamics become trapped in local minima for moderate amounts of time — see e.g. Tzen et al. (2018) for further background. Also in the sampling context, the works Lee et al. (2018); Ge et al. (2018) studied a related problem, where the goal is to sample a mixture of isotropic Gaussians given black-box access to the score function (which they do via simulated tempering). This problem ends up to be different to the ones arising in score matching: they need exact knowledge of the true score function (far away from the support of the distribution), but they do not have access to training

data from the true distribution. As a consequence of the differing setup, they prove an impossibility result (Ge et al., 2018, Theorem F.1) for a mixture of two Gaussians with covariances $I$ and $2I$ (it will not be possible to find both components), but our result proves this is not an issue in our setting.

**Questions for future work.**    In our result, we proved the first bound for sampling with the vanilla score, estimated from data, which succeeds in the multimodal setting, but it is an open question if the dependence on the number of components is optimal; it seems likely that the dependence can be improved, at least in many cases. Finally, it is interesting to ask what the largest class of distributions our result can generalize to — with data-based initialization, multimodality itself is no longer an obstruction to sampling with Langevin from estimated gradients, but are there other possible obstructions?

## 2    TECHNICAL OVERVIEW

We first review some background and notation which is helpful for discussing the proof sketch. We leave complete proofs of all results to the appendices.

**Notation.**    We use standard big-Oh notation and use tildes, e.g. $\tilde{O}(\cdot)$, to denote inequality up to log factors and $O_B(\cdot)$ to denote an inequality with a constant allowed to depend on $B$. We let $d_{TV}(\mu, \nu) = \sup_A |\mu(A) - \nu(A)|$ be the usual total variation distance between probability measures $\mu$ and $\nu$ defined on the same space, where the supremum ranges over measurable sets. Given a random variable $X$, we write $\mathcal{L}(X)$ to denote its law.

**Log-Sobolev inequality.**    We say probability distribution $\pi$ satisfies a log-Sobolev inequality (LSI) with constant $C_{LS}$ if for all smooth functions $f$, $\mathbb{E}_\pi[f^2 \log(f^2/\mathbb{E}_\pi[f^2])] \leq 2C_{LS}\mathbb{E}_\pi[||\nabla f||^2]$. Due to the Bakry-Emery criterion, if $\pi$ is $\alpha$-strongly log-concave then $\pi$ satisfies LSI with constant $C_{LS} = 1/\alpha$. LSI is equivalent to a statement about mixing of the Langevin dynamics — if we let $\pi_t$ denote the law of the diffusion at time $t$ then an LSI is equivalent to the inequality

$$\mathcal{D}_{\mathrm{KL}}(\pi_t || \pi) \leq \exp(-2t/C_{LS})\mathcal{D}_{\mathrm{KL}}(\pi_0 || \pi)$$

holding for an arbitrary initial distribution $\pi_0$. Here $\mathcal{D}_{KL}(P, Q) = \mathbb{E}_P[\log \frac{dP}{dQ}]$ is the Kullback-Liebler divergence. See Bakry et al. (2014); Van Handel (2014) for more background.

**Stochastic calculus.**    We will need to use stochastic calculus to compare the behavior of similar diffusion processes — see Karatzas & Shreve (1991) for formal background. Let $(X_t)_{t \geq 0}$ and $(Y_t)_{t \geq 0}$ be two Ito processes defined by SDEs: $dX_t = s_1(X_t)dt + dB_t$ and $dY_t = s_2(X_t)dt + dB_t$. Let $P_T, Q_T$ be the laws of the paths $(X_t)_{t \in [0,T]}$ and $(Y_t)_{t \in [0,T]}$ respectively. The following follows by Girsanov's theorem (see (Chen et al., 2023, Eq. (5.5) and Theorem 9))

$$d_{TV}(Y_T, X_T)^2 \leq d_{TV}(Q_T, P_T)^2 \leq \frac{1}{2}\mathbb{E}_{Q_T}\left[\int_0^T ||s_2(Y_t) - s_1(Y_t)||^2 dt\right]$$

In particular, this is useful to compare continuous and discrete time Langevin diffusions. If $(Y_t)$ be the continuous Langevin diffusion with score function $s$, and $(X_t)$ is a linearly interpolated version of the discrete-time Langevin dynamics defined by $dX_t = s(X_{\lceil t/h \rceil h})dt + dB_t$, then

$$d_{TV}(Y_T, X_T)^2 \leq \frac{1}{2}\mathbb{E}_{Q_T}\left[\int_0^T ||s(Y_t) - s(Y_{\lceil t/h \rceil h})||^2 dt\right] \tag{3}$$

### 2.1    PROOF SKETCH

**High-level discussion.**    At a high level, our argument proceeds by (1) grouping the components of the mixture into larger "well-connected" pieces, and (2) showing that the process mixes well within each of these pieces, while preserving the correct relative weight of each piece. One of the challenges in proving our result is that, contrary to the usual situation in the analysis of Markov chains (as in e.g. Bakry et al. (2014); Levin & Peres (2017)), we *do not* want to run the Langevin diffusion until it mixes to its stationary distributions. If we ran the process until mixing, then we would be performing

the vanilla score matching procedure which provably fails in most multimodal settings because it incorrectly weights the different components (Koehler et al., 2022). So what we want to do is prove the process succeeds at some intermediate time $T$ (See Figure 1 for a simulation illustrating this.)

To build intuition, consider the special case where all of the components in the mixture distributions are very far from each other. In this case, one might guess that taking $T$ to be the maximum of the mixing times of each of the individual components will work. Provided there are enough samples in the dataset, the initialization distribution will accurately model the relative weights of the different clusters in the data, and running the process up to time $T$ will approximately sample from the cluster that the initialization is drawn from. We could hope to prove the result by arguing that the dynamics on the mixture is close to the dynamics on one of the mixture components.

**Some challenges to overcome in the analysis.** This is the right intuition, but for the general case the behavior of the dynamics is more complicated. When components are close, the score function of the mixture distribution may not be close to the score function of either component in the region of overlap; relatedly, particles may cross over between components. Also, the following remark shows that natural variants of our main theorem are actually false.

**Remark 5.** *We might think that initializing from the* center *of each mixture component would work just as well as initializing from samples. This is fine if the clusters are all very far from each other, but wrong in general. If the underlying mixture distribution is $\frac{1}{2}N(0, I_d) + \frac{1}{2}N(0, 2I_d)$ and the dimension $d$ is large, then the first component will have almost all of its mass within distance $O(1)$ of a sphere of radius $\sqrt{d}$ and the second component will similarly concentrate about a sphere of radius $\sqrt{2d}$. (See Theorem 3.1.1 of Vershynin (2018).) As a consequence, the dynamics initialized at the origin will mix within the shell of radius $\sqrt{d}$ but take $\exp(\Omega(d))$ time to cross to the larger $\sqrt{2d}$ shell. (This can be proved by observing that the gap between the two spheres forms a "bottleneck" for the dynamics, see Levin & Peres (2017).) In contrast, if we initialize from samples then approximately half of them will lie on the outer shell and, as we prove, the dynamics mix correctly.*

We now proceed to explain in more detail how we prove our result. We start with the analysis of an idealized diffusion process, and then through several comparison arguments establish the result for the real LMC algorithm.

**Analysis of idealized diffusion.** To start out, we analyze an idealized process in which:

1. The score function $\nabla \log \mu$ is known exactly. (Our result is still new in this case.)

2. The dynamics is the *continous-time* Langevin diffusion given by the Ito process
   $$d\bar{X}_t = \nabla \log \mu(\bar{X}_t)\,dt + \sqrt{2}\,dB_t.$$
   This is the scaling limit of the discrete-time LMC chain as we take the step size $h \to 0$, where $dB_t$ is the differential of a Brownian motion $B_t$.

3. For purposes of exposition, we make the fictitious assumption that the ground truth distribution $\mu$ is supported in a ball of radius $R$. This will not be literally true, but for sufficiently large $R$ $\mu$ will be almost entirely contained within a radius $R$ ball. (In the supplement, we handle this rigorously using concentration, see e.g. proof of Lemma 11 of Appendix F).

Additionally, for the purpose of illustration, in this proof sketch we assume the target distance in TV is 0.01 and consider the case where there are two $\alpha$-strongly log concave and $\beta$-smooth components $\mu_1$ and $\mu_2$, and $\mu = \frac{1}{2}\mu_1 + \frac{1}{2}\mu_2$. After we complete the proof sketch for this setting, we will go back and explain how to generalize the analysis to arbitrary mixtures, handle the error induced by discretization, and finally make the analysis work with an $L_2$ estimate of the true score function.

*Overlap parameter.* We define
$$\delta_{12} := 1 - d_{TV}(\mu_1, \mu_2) = \int \min\{\mu_1(x), \mu_2(x)\}dx$$

as a quantitative measure of how much components 1 and 2 overlap; for example, $\delta_{12} = 1$ iff $\mu_1$ and $\mu_2$ are identical. The analysis splits into cases depending on whether $\delta_{12}$ is large; we let $\delta > 0$ be a parameter which determines this split and which will be optimized at the end.

*High overlap case (Appendix C).* If $\mu_1$ and $\mu_2$ has high overlap, in the sense that $\delta_{12} \geq \delta$, then we show that $\mu$ satisfies a log Sobolev inequality with constant at most $O(1/(\alpha\delta))$, by applying our Theorem 2, an important technical ingredient which is discussed in more detail below. Thus for a typical sample $x$ from $\mu$, the continuous Langevin diffusion $(X_t^{\delta_x})_{t\geq 0}$ with score function $\nabla \log \mu$ initialized at $x$ converges to $\mu$ i.e. $d_{TV}(\mathcal{L}(\bar{X}_t^{\delta_x}), \mu) \leq \epsilon$ for $T \geq \Omega(\frac{1}{\alpha\delta} \log(d\epsilon^{-1}))$.[3]

*Low overlap case (Appendix F, Lemma 11).* When $\mu_1$ and $\mu_2$ have small overlap i.e. $\delta_{12} \leq \delta$, we will show that for $x \sim \mu$, with high probability, the gradient of the log-likelihood of the mixture distribution $\mu$ at $x$ is close to that of one of the components $\mu_1, \mu_2$ *(Appendix F.1)*. This is because, supposing that $||x|| \leq R$, for $i \in \{1, 2\}$ we can upper bound

$$||\nabla \log \mu(x) - \nabla \log \mu_i(x)|| \leq 2\beta R \left( 1 - \frac{\mu_i(x)}{\mu_1(x) + \mu_2(x)} \right),$$

and low overlap implies that $\min_i \left( 1 - \frac{\mu_i(x)}{\mu_1(x)+\mu_2(x)} \right)$ is small for *typical* $x \sim \mu$.

Consider the continuous Langevin diffusion $(\bar{X}_t^{\delta_x})$ initialized at $\delta_x$ i.e. $\bar{X}_0 = x$. Observe that the *marginal* law of $\bar{X}_t^{\delta_x}$ where $x \sim \mu$ is exactly $\mu$, since $\mu$ is the stationary distribution of the Langevin diffusion. Let $H > 0$ be a parameter to be tuned later. The above discussion and Markov's inequality allows us to argue that for a typical sample $x$, the gradient of the log-likelihood of $\mu$ at $\bar{X}_{nH}^{\delta_x}$ is close to that of either components $\mu_1, \mu_2$ with high probability.

Next, we perform a union bound over $n \in \{0, \cdots, N-1\}$ and bound the drift $||\nabla \log \mu(x) - \nabla \log \mu_i(x)||$ in each small time interval $[nH, (n+1)H]$. By doing so, we can argue that for a typical sample $x \sim \mu$, with probability at least $1 - \epsilon^{-1}\beta RN\delta_{12}$ over the randomness of the Brownian motion driving the Langevin diffusion, the gradient of the log-likelihood at $\bar{X}_t^{\delta_x}$ for $t \in [0, NH]$ is close to that of the component distribution $\mu_i$ closest to the initial point $x$ (see Proposition 26 of Appendix F).

In other words, assuming that the initial point $x$ satisfies $\mu_1(x) \geq \mu_2(x)$ and letting $T = NH$, we can show that with high probability,

$$\sup_{t\in[0,T]} ||\nabla \log \mu(\bar{X}_t^{\delta_x}) - \nabla \log \mu_1(\bar{X}_t^{\delta_x})|| \leq 1.1\epsilon.$$

This allows us, using (3), to compare our Langevin diffusion with the one with score function $\nabla \log \mu_1$ and show the output at time $T$ is approximately a sample from $\mu_1$.

In a typical set $U_{\text{sample}}$ of i.i.d. samples from $\mu$, roughly 50% of the samples $x \in U_{\text{sample}}$ satisfy $\mu_1(x) \geq \mu_2(x)$ and the other 50% samples satisfy $\mu_2(x) \geq \mu_1(x)$, thus the Langevin dynamics $(\bar{X}_t^{\nu_{\text{sample}}})_{t\geq 0}$ initialized at the uniform distribution $\nu_{\text{sample}}$ over $U_{\text{sample}}$ will be close to $\frac{\mu_1+\mu_2}{2} = \mu$ after time $T$ provided we set $H, T, \epsilon, \delta$ appropriately.

*Concluding the idealized analysis.* Either $\delta_{12} \geq \delta$ in which case the high-overlap analysis above based on the log-Sobolev constant succeeds, or $\delta_{12} < \delta$ in which case the low-overlap analysis succeeds. Optimizing over $\delta$, we find that in either case, with high probability over the set $U_{\text{sample}}$ of samples from $\mu$, for $t \geq \tilde{\Omega}(\frac{(\beta R)^3}{\alpha^{5/2}})$ we have

$$d_{TV}(\mathcal{L}(\bar{X}_t^{\nu_{\text{sample}}} \mid U_{\text{sample}}), \mu) \leq 0.01$$

as desired.

**Generalizing idealized analysis to arbitrary mixtures.** *(Appendix F, Theorem 5)* When there are more than two components, we can generalize this analysis — the key technical difficulty, alluded to earlier, is analyzing the overlap between different mixture components. We do this by defining, for each $\delta > 0$, a graph $\mathbb{G}^\delta$ where there is an edge between $i, j \in [K]$ when $\delta_{ij} := 1 - d_{TV}(\mu_i, \mu_j) \leq \delta$. As long as the minimum of the weights $p_* := \min_i p_i$ is not too small, each connected component $C$ of $\mathbb{G}^\delta$ is associated with a probability distribution $\mu_C = \frac{\sum_{i\in C} p_i \mu_i}{\sum_{i\in C} p_i}$ that has log Sobolev constant on the order of $O_{K, p_*^{-1}}(1/\alpha\delta)$.

---

[3]This follows as LSI yields exponential convergence in KL-divergence. While the KL-divergence of the initialization $\delta_x$ with respect to $\mu$ is unbounded, we can bound the KL-divergence of $\bar{X}_h^{\delta_x}$ for some small $h$.

Suppose for a moment that the connected components are well separated compared to the magnitude of $\delta$. More precisely, suppose that for $i, j$ in different connected components and some $\delta > 0$ we have

$$\delta_{ij} \leq f(\delta) := \Theta\left(\frac{(\alpha\delta)^{3/2}}{(\beta R)^3}\right). \tag{4}$$

Then, a direct generalization of the argument for two components shows that for a typical set $U_{\text{sample}}$ of i.i.d. samples from $\mu$, the continuous Langevin diffusion $(\bar{X}_t^{\nu_{\text{sample}}})_{t \geq 0}$ initialized at the uniform distribution over $U_{\text{sample}}$ converges to $\mu$ after time $T_\delta = (\alpha\delta)^{-1}$.

It remains to discuss how we select $\delta$ so that (4) is satisfied. We consider a decreasing sequence $1 = \delta_0 > \delta_1 > \cdots > \delta_{K-1}$ where $\delta_{r+1} = f(\delta_r)$ as in Eq. (4). Let $\mathbb{G}^r := \mathbb{G}^{\delta_r}$. If any two vertices from different connected components of $\mathbb{G}^r$ have overlap at most $\delta_{r+1}$, then the above argument applies. Otherwise, $\mathbb{G}^{r+1}$ must have one less connected component than $\mathbb{G}^r$, and since $\mathbb{G}^0$ has at most $K$ connected components, $\mathbb{G}^{K-1}$ must have 1 connected component and the above argument applies to it. Thus, in all cases, the distribution of $\bar{X}_{T_{\delta_{K-1}}}^{\nu_{\text{sample}}}$ is close to $\mu$ in total variation distance.

**Discretization analysis.** *(Appendix G, Lemma 14)* We now move from a continuous-time to discrete-time process. Let $(X_{nh})_{n \in \mathbb{N}}$ and $(\bar{X}_t)_{t \geq 0}$ be respectively the LMC with step size $h$ and the continuous Langevin diffusion. Both are with score function $\nabla \log \mu$ and have the same initialization. By an explicit calculation, we can bound $||\nabla^2 \log \mu(x)||_{OP}$ along the trajectory of the continuous process. This combined with the consequence of Girsanov's theorem (3) allows us to bound the total variation distance between the continuous $(\bar{X}_t)$ and discretized $(X_{nh})$ processes. For appropriate choices of step size $h$ and time $T = Nh$, using triangle inequality and the bound $d_{TV}(\bar{X}_T, \mu)$, we conclude that the discretized process $X_{Nh}$ is close to $\mu$.

**Sampling with an $L_2$-approximate score function.** *(Appendix G)* In many cases, score functions are learned from data, so we only have access to an $L_2$-estimate $s$ of the score such that $\mathbb{E}_\mu[||s(x) - \nabla \log \mu(x)||^2] \leq \epsilon_{\text{score}}^2$. We now describe how to make the analysis work in this setting. Using Girsanov's theorem, we can bound the total variation distance between the LMC $(X_{nh}^{s,\mu})_{n \in \mathbb{N}}$ initialized at $\mu$ with score estimate $s$ and the continuous Langevin diffusion $(\bar{Z}_{nh}^\mu)_{n \in \mathbb{N}}$ with true score function $\nabla \log \mu$, thus we can bound the probability that the LMC $(X_{nh}^{s,\mu})_{n = \{0, \cdots, N-1\}}$ hits the bad set

$$B_{\text{score}} := \{x : ||s(x) - \log \mu(x)|| \geq \epsilon_{\text{score},1}\}.$$

(The idea of defining a "bad set" is inspired by the analysis of Lee et al. (2022a).) Similar to the argument for the continuous process, let $X_{nh}^{s,\nu_{\text{sample}}}$ denote the LMC with score function $s$ and step size $h$ initialized at the empirical distribution $\nu_{\text{sample}}$. Since we know that $X_{nh}^{s,\mu}$ avoids the bad set and that $\mathcal{L}(X_{nh}^{s,\mu}) = \mathbb{E}_{U_{\text{sample}} \sim \mu^{\otimes M}}[\mathcal{L}(X_{nh}^{s,\nu_{\text{sample}}})]$, we have by Markov's inequality that for a typical $U_{\text{sample}}$, with high probability over the randomness of the Brownian motion, $X_{nh}^{s,\nu_{\text{sample}}}$ also avoids the bad set $B_{\text{score}}$ for all $0 \leq n < N$. Thus, we can compare $X_{nh}^{s,\nu_{\text{sample}}}$ with the LMC with true score function $\nabla \log \mu$, and conclude that $\mathcal{L}(X_{Nh}^{s,\nu_{\text{sample}}})$ is close to $\mu$ in total variation distance.

## 2.2 Technical ingredient: log-Sobolev constant of well-connected mixtures

The following theorem, which we prove in the appendix, is used in the above argument to bound the log-Sobolev constant of mixture distributions where the components have significant overlap.

**Theorem 2.** *Let $I$ be a set, and consider probability measures $\{\mu_i\}_{i \in I}$, nonnegative weights $(p_i)_{i \in I}$ summing to one, and mixture distribution $\mu = \sum_i p_i \mu_i$. Let $G$ be the graph on vertex set $I$ where there is an edge between $i, j$ if $\mu_i, \mu_j$ have high overlap i.e.*

$$\delta_{ij} := \int \min\{\mu_i(x), \mu_j(x)\}dx \geq \delta.$$

*Suppose $G$ is connected and let $p_* = \min p_i$. The mixture distribution $\mu = \sum_{i \in I} p_i \mu_i$ has log-Sobolev constant*

$$C_{LS}(\mu) \leq \frac{C_{|I|,p_*}}{\delta} \max_i C_{LS}(\mu_i)$$

*where $C_{|I|,p_*} = 4|I|(1 + \log(p_*^{-1}))p_*^{-1}$ only depends on $|I|$ and $p_*$.*

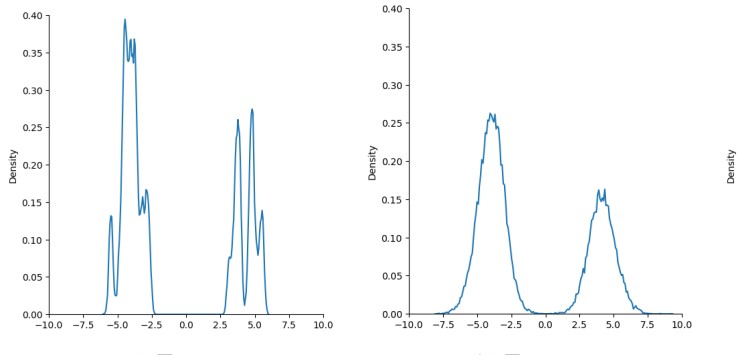

(a) $T = 0$          (b) $T = 200$          (c) $T = \infty$ & truth (orange)

Figure 1: Visualization of the distribution of the Langevin dynamics after $T$ iterations when initialized at the empirical distribution and run with an approximate score function estimated from data. Orange density (rightmost figure) is the ground truth mixture of two Gaussians; the empirical distribution (leftmost figure, $T = 0$) consists of 40 iid samples from the ground truth. Langevin dynamics with step size $0.01$ is run with an estimated score function, which was fit using vanilla score matching with a one hidden-layer neural network trained on fresh samples; densities (blue) are visualized using a Gaussian Kernel Density Estimate (KDE). Matching our theory, we see that the ground truth is accurately estimated at time $T = 200$ even though it is not at $T = 0$ or $\infty$.

A version of Theorem which bounds the (weaker) Poincaré constant instead appeared before as Theorem 1.2 of Madras & Randall (2002), but the result for the log-Sobolev constant is new to the best of our knowledge. Compared to Chen et al. (2021), our assumption is milder than their assumption that the chi-square divergence between any two components is bounded. (For example, two non-isotropic Gaussians might have infinite chi-square divergence (see e.g. (Schlichting, 2019, Section 4.3)), so in that case their result doesn't imply a finite bound on the LSI of their mixture.) Schlichting (2019) bounds LSI of $\mu = p\mu_1 + (1 - p)\mu_2$ when either $\chi^2(\mu_1\|\mu_2)$ or $\chi^2(\mu_2\|\mu_1)$ are bounded; our bound applies to mixtures of more than two components.

## 3   SIMULATIONS

In Figure 1, we simulated the behavior of the Langevin dynamics with step size $0.01$ and an estimated score function initialized at the ground truth distribution on a simple 1-dimensional example, a mixture of two Gaussians. If the Langevin dynamics are run until mixing, this corresponds to exactly performing the standard vanilla score matching procedure and this will fail to estimate the ground truth distribution well, which we see in the rightmost subfigure. The empirical distribution (time zero for the dynamics) is also not a good fit to the ground truth, but as our theory predicts the early-stopped Langevin diffusion (subfigure (b)) is indeed a good estimate for the ground truth.

In Figure 2 we simulated the trajectories of Langevin dynamics with step size $0.001$, again with initialization from samples and a learned score function, in a 32-dimensional mixture of Gaussians. Similar to the one-dimensional example, we can see that at moderate times the trajectories have mixed well within their component, and at large times the trajectories sometimes pass through the region in between the components where the true density is very small. Additional simulations (including an experiment with Contrastive Divergence training) and information is in Appendix I.

ACKNOWLEDGMENTS

F.K. was supported in part by NSF award CCF1704417, NSF award IIS1908774, and N. Anari's Sloan Research Fellowship.

REFERENCES

Michael S Albergo, Denis Boyda, Daniel C Hackett, Gurtej Kanwar, Kyle Cranmer, Sébastien Racaniere, Danilo Jimenez Rezende, and Phiala E Shanahan. Introduction to normalizing flows for

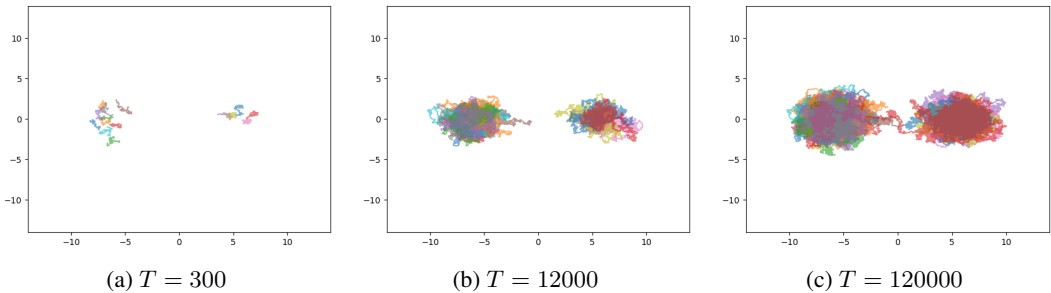

(a) $T = 300$      (b) $T = 12000$      (c) $T = 120000$

Figure 2: 2D projected trajectories of Langevin dynamics up to $T$ iterations with step size $0.001$ in a 32-dimensional mixture of Gaussians $\frac{2}{3}N(-6e_1, 1.5I) + \frac{1}{3}N(6e_1, 1.5I)$. The projection is the first two coordinates and the direction of separation of the components is the first axis direction. Langevin is initialized from the empirical distribution (15 iid samples) and run with an approximate score function learned from samples using a one hidden-layer neural network.

lattice field theory. *arXiv preprint arXiv:2101.08176*, 2021.

Dominique Bakry, Ivan Gentil, Michel Ledoux, et al. *Analysis and geometry of Markov diffusion operators*, volume 103. Springer, 2014.

Alessandro Barp, Francois-Xavier Briol, Andrew Duncan, Mark Girolami, and Lester Mackey. Minimum stein discrepancy estimators. *Advances in Neural Information Processing Systems*, 32, 2019.

Julian Besag. Statistical analysis of non-lattice data. *Journal of the Royal Statistical Society: Series D (The Statistician)*, 24(3):179–195, 1975.

Adam Block, Youssef Mroueh, and Alexander Rakhlin. Generative modeling with denoising auto-encoders and langevin sampling. *arXiv preprint arXiv:2002.00107*, 2020.

Matthew Brennan, Guy Bresler, and Wasim Huleihel. Reducibility and computational lower bounds for problems with planted sparse structure. In *Conference On Learning Theory*, pp. 48–166. PMLR, 2018.

Hong-Bin Chen, Sinho Chewi, and Jonathan Niles-Weed. Dimension-free log-sobolev inequalities for mixture distributions. *Journal of Functional Analysis*, 281(11):109236, 2021.

Sitan Chen, Sinho Chewi, Jerry Li, Yuanzhi Li, Adil Salim, and Anru R. Zhang. Sampling is as easy as learning the score: theory for diffusion models with minimal data assumptions, 2023.

Sinho Chewi, Murat A. Erdogdu, Mufan Bill Li, Ruoqi Shen, and Matthew Zhang. Analysis of langevin monte carlo from poincaré to log-sobolev, 2021.

Arnak S Dalalyan. Theoretical guarantees for approximate sampling from smooth and log-concave densities. *Journal of the Royal Statistical Society. Series B (Statistical Methodology)*, pp. 651–676, 2017.

Persi Diaconis and Laurent Saloff-Coste. Logarithmic sobolev inequalities for finite markov chains. *The Annals of Applied Probability*, 6(3):695–750, 1996.

Peter GM Forbes and Steffen Lauritzen. Linear estimating equations for exponential families with application to gaussian linear concentration models. *Linear Algebra and its Applications*, 473: 261–283, 2015.

Ruiqi Gao, Yang Lu, Junpei Zhou, Song-Chun Zhu, and Ying Nian Wu. Learning generative convnets via multi-grid modeling and sampling. In *Proceedings of the IEEE Conference on Computer Vision and Pattern Recognition*, pp. 9155–9164, 2018.

Rong Ge, Holden Lee, and Andrej Risteski. Simulated tempering langevin monte carlo ii: An improved proof using soft markov chain decomposition. *arXiv preprint arXiv:1812.00793*, 2018.

Geoffrey E Hinton. Training products of experts by minimizing contrastive divergence. *Neural computation*, 14(8):1771–1800, 2002.

Geoffrey E Hinton. A practical guide to training restricted boltzmann machines. *Neural Networks: Tricks of the Trade: Second Edition*, pp. 599–619, 2012.

Jonathan Ho, Ajay Jain, and Pieter Abbeel. Denoising diffusion probabilistic models. *Advances in neural information processing systems*, 33:6840–6851, 2020.

Aapo Hyvärinen. Estimation of non-normalized statistical models by score matching. *Journal of Machine Learning Research*, 6(4), 2005.

Aapo Hyvärinen. Connections between score matching, contrastive divergence, and pseudolikelihood for continuous-valued variables. *IEEE Transactions on neural networks*, 18(5):1529–1531, 2007a.

Aapo Hyvärinen. Some extensions of score matching. *Computational statistics & data analysis*, 51 (5):2499–2512, 2007b.

Ioannis Karatzas and Steven E Shreve. *Brownian motion and stochastic calculus*, volume 113. Springer Science & Business Media, 1991.

Frederic Koehler, Alexander Heckett, and Andrej Risteski. Statistical efficiency of score matching: The view from isoperimetry. *arXiv preprint arXiv:2210.00726*, 2022.

Scott Lawrence and Yukari Yamauchi. Normalizing flows and the real-time sign problem. *Physical Review D*, 103(11):114509, 2021.

Holden Lee, Andrej Risteski, and Rong Ge. Beyond log-concavity: Provable guarantees for sampling multi-modal distributions using simulated tempering langevin monte carlo. In S. Bengio, H. Wallach, H. Larochelle, K. Grauman, N. Cesa-Bianchi, and R. Garnett (eds.), *Advances in Neural Information Processing Systems*, volume 31. Curran Associates, Inc., 2018. URL https://proceedings.neurips.cc/paper/2018/file/c6ede20e6f597abf4b3f6bb30cee16c7-Paper.pdf.

Holden Lee, Jianfeng Lu, and Yixin Tan. Convergence for score-based generative modeling with polynomial complexity. *arXiv preprint arXiv:2206.06227*, 2022a.

Holden Lee, Jianfeng Lu, and Yixin Tan. Convergence of score-based generative modeling for general data distributions. *arXiv preprint arXiv:2209.12381*, 2022b.

David A Levin and Yuval Peres. *Markov chains and mixing times*, volume 107. American Mathematical Soc., 2017.

Neal Madras and Dana Randall. Markov chain decomposition for convergence rate analysis. *Annals of Applied Probability*, 12:581–606, 2002.

Ilya Mironov. Rényi differential privacy. In *2017 IEEE 30th computer security foundations symposium (CSF)*, pp. 263–275. IEEE, 2017.

Erik Nijkamp, Mitch Hill, Song-Chun Zhu, and Ying Nian Wu. Learning non-convergent non-persistent short-run mcmc toward energy-based model. *Advances in Neural Information Processing Systems*, 32, 2019.

Erik Nijkamp, Mitch Hill, Tian Han, Song-Chun Zhu, and Ying Nian Wu. On the anatomy of mcmc-based maximum likelihood learning of energy-based models. In *Proceedings of the AAAI Conference on Artificial Intelligence*, volume 34, pp. 5272–5280, 2020.

Phillippe Rigollet and Jan-Christian Hütter. High dimensional statistics. *Lecture notes for course 18S997*, 2017.

Robin Rombach, Andreas Blattmann, Dominik Lorenz, Patrick Esser, and Björn Ommer. High-resolution image synthesis with latent diffusion models. In *Proceedings of the IEEE/CVF Conference on Computer Vision and Pattern Recognition*, pp. 10684–10695, 2022.

André Schlichting. Poincaré and log–sobolev inequalities for mixtures. *Entropy*, 21(1):89, 2019. doi: 10.3390/e21010089. URL `https://doi.org/10.3390%2Fe21010089`.

Yang Song and Stefano Ermon. Generative modeling by estimating gradients of the data distribution. *Advances in Neural Information Processing Systems*, 32, 2019.

Yang Song, Sahaj Garg, Jiaxin Shi, and Stefano Ermon. Sliced score matching: A scalable approach to density and score estimation. In *Uncertainty in Artificial Intelligence*, pp. 574–584. PMLR, 2020a.

Yang Song, Jascha Sohl-Dickstein, Diederik P Kingma, Abhishek Kumar, Stefano Ermon, and Ben Poole. Score-based generative modeling through stochastic differential equations. *arXiv preprint arXiv:2011.13456*, 2020b.

Bharath Sriperumbudur, Kenji Fukumizu, Arthur Gretton, Aapo Hyvärinen, and Revant Kumar. Density estimation in infinite dimensional exponential families. *Journal of Machine Learning Research*, 18, 2017.

Belinda Tzen, Tengyuan Liang, and Maxim Raginsky. Local optimality and generalization guarantees for the langevin algorithm via empirical metastability. In *Conference On Learning Theory*, pp. 857–875. PMLR, 2018.

Ramon Van Handel. Probability in high dimension. Technical report, PRINCETON UNIV NJ, 2014.

Santosh Vempala and Andre Wibisono. Rapid convergence of the unadjusted langevin algorithm: Isoperimetry suffices. *Advances in neural information processing systems*, 32, 2019.

Roman Vershynin. *High-dimensional probability: An introduction with applications in data science*, volume 47. Cambridge university press, 2018.

Pascal Vincent. A connection between score matching and denoising autoencoders. *Neural computation*, 23(7):1661–1674, 2011.

Larry Wasserman. *All of nonparametric statistics*. Springer Science & Business Media, 2006.

Li Wenliang, Danica J Sutherland, Heiko Strathmann, and Arthur Gretton. Learning deep kernels for exponential family densities. In *International Conference on Machine Learning*, pp. 6737–6746. PMLR, 2019.

Jianwen Xie, Yang Lu, Song-Chun Zhu, and Yingnian Wu. A theory of generative convnet. In *International Conference on Machine Learning*, pp. 2635–2644. PMLR, 2016.

# A ORGANIZATION OF APPENDIX

In Appendix B, we review some basic mathematical preliminaries and notation, such as the definition of log-Sobolev and Poincaré inequalities. In Appendix C we prove Theorem 4 (Theorem 2 of the main text), which shows that when clusters have significant overlap that the Langevin dynamics for the mixture distribution will successfully mix. Appendix D and Appendix E contain intermediate results which are used in the following sections: Appendix D shows how to analyze the Langevin diffusion starting from a point, and Appendix E shows how to bound the drift of the continuous Langevin diffusion over a short period of time. In Appendix F we prove Theorem 5, which shows that the continuous Langevin diffusion with score function $\nabla V$ converges to $\mu$ after a suitable time $T$. In Appendix G, we prove our main results Theorem 6 and Corollary 1, which show that the discrete LMC with score function $s$ with appropriately chosen step size is close to $\mu$ in total variation distance at a suitable time. Corollary 1 corresponds to Theorem 1 of the main text. In Appendix H, we remove the dependency of the runtime and number of samples on the minimum weight of the components i.e. $p_* = \min_{i \in I} p_i$ (see Theorem 8 and Corollary 2 for the analogy of Theorem 6 and Corollary 1 respectively that has no dependency on $p_*$). Appendix I contains some additional simulations.

# B PRELIMINARIES

In the preliminaries, we review in more detail the needed background on divergences between probability measures, functional inequalities, log-concave distributions, etc. in order to prove our main results.

**Notation.** We use standard big-Oh notation and use tildes, e.g. $\tilde{O}(\cdot)$, to denote inequality up to log factors. We similarly use the notation $\lesssim$ to denote inequality up to a universal constant. We let $d_{TV}(\mu, \nu) = \sup_A |\mu(A) - \nu(A)|$ be the usual total variation distance between probability measures $\mu$ and $\nu$ defined on the same space, where the supremum ranges over measurable sets. Given a random variable $X$, we write $\mathcal{L}(X)$ to denote its law. In general, we use the same notation for a measure and its probability density function as long as there is no ambiguity. For random variables $X, Z$, we will write $d_{TV}(X, Z)$ to denote the total variation distance between their laws $\mathcal{L}(X)$ and $\mathcal{L}(Z)$.

## B.1 RENYI DIVERGENCE

The Renyi divergence, which generalizes the more well-known KL divergence, is a useful technical tool in the analysis of the Langevin diffusion — see e.g. Vempala & Wibisono (2019). The Renyi divergence of order $q \in (1, \infty)$ of $\mu$ from $\pi$ is defined to be

$$\mathcal{R}_q(\mu \| \pi) = \frac{1}{q-1} \ln \mathbb{E}_\pi \left[ \left( \frac{d\mu(x)}{d\pi(x)} \right)^q \right] = \frac{1}{q-1} \ln \int \left( \frac{d\mu(x)}{d\pi(x)} \right)^q d\pi(x)$$

$$= \frac{1}{q-1} \ln \int \left( \frac{d\mu(x)}{d\pi(x)} \right)^{q-1} d\mu(x) = \frac{1}{q-1} \ln \mathbb{E}_\mu \left[ \left( \frac{d\mu(x)}{d\pi(x)} \right)^{q-1} \right]$$

The limit $\mathcal{R}_q$ as $q \to 1$ is the Kullback-Leibler divergence $\mathcal{D}_{\mathrm{KL}}(\mu \| \pi) = \int \mu(x) \log \frac{\mu(x)}{\pi(x)} dx$, thus we write $\mathcal{R}_1(\cdot) = \mathcal{D}_{\mathrm{KL}}(\cdot)$. Renyi divergence increases as $q$ increases i.e. $\mathcal{R}_q \leq \mathcal{R}_{q'}$ for $1 \leq q \leq q'$.

**Lemma 1** (Weak triangle inequality, (Vempala & Wibisono, 2019, Lemma 7), Mironov (2017)). *For $q > 1$ and any measure $\nu$ absolutely continuous with respect to measure $\mu$,*

$$\mathcal{R}_q(\nu \| \mu) \leq \frac{q - 1/2}{q - 1} \mathcal{R}_{2q}(\nu \| \nu') + \mathcal{R}_{2q-1}(\nu' \| \mu)$$

**Lemma 2** (Weak convexity of Renyi entropy). *For $q > 1$, if $\mu$ is a convex combination of $\mu_i$ i.e. $\mu(x) = \sum p_i \mu_i(x)$ then*

$$\mathbb{E}_\nu \left[ \left( \frac{d\nu(x)}{d\mu(x)} \right)^{q-1} \right] \leq \sum_i p_i \mathbb{E}_\nu \left[ \left( \frac{d\nu(x)}{d\mu_i(x)} \right)^{q-1} \right].$$

*Consequently, $\mathcal{R}_q(\nu \| \mu) \leq \max_i \mathcal{R}_q(\nu \| \mu_i)$ and $\mathcal{R}_q(\mu \| \nu) \leq \max_i \mathcal{R}_q(\mu_i \| \nu)$*

*Proof.* By Holder's inequality

$$(\sum_i p_i \mu_i(x))^{q-1} \left( \sum_{i=1}^{d} \frac{p_i}{\mu_i(x)^{q-1}} \right) \geq (\sum_i p_i)^q = 1$$

thus

$$\left( \frac{\nu(x)}{\mu(x)} \right)^{q-1} \leq \sum_i p_i \left( \frac{\nu(x)}{\mu_i(x)} \right)^{q-1}$$

Taking expectation in $\nu$ gives the first statement. Similarly, since $q > 1 > 0$,

$$\mathbb{E}_\nu \left[ \left( \frac{\nu(x)}{\mu(x)} \right)^q \right] \leq \sum_i p_i \mathbb{E}_\nu \left[ \left( \frac{\nu(x)}{\mu_i(x)} \right)^q \right]$$

For the second statement

$$\mathcal{R}_q(\nu||\mu) = \frac{\ln \mathbb{E}_\nu[(\frac{d\nu(x)}{d\mu(x)})^{q-1}]}{q-1} \leq \frac{\ln(\max_i \mathbb{E}_\nu[(\frac{d\nu(x)}{d\mu_i(x)})^{q-1}])}{q-1} = \max_i \mathcal{R}_q(\nu||\mu_i)$$

and

$$\mathcal{R}_q(\mu||\nu) = \frac{\ln \mathbb{E}_\nu[(\frac{d\nu(x)}{d\mu(x)})^q]}{q-1} \leq \frac{\ln(\max_i \mathbb{E}_\nu[(\frac{d\nu(x)}{d\mu_i(x)})^q])}{q-1} = \max_i \mathcal{R}_q(\mu_i||\nu).$$

$\square$

## B.2 LOG-CONCAVE DISTRIBUTIONS

Consider a density function $\pi : \mathbb{R}^d \to \mathbb{R}_{\geq 0}$ where $\pi(x) = \exp(-V(x))$. Throughout the paper, we will assume $V$ is a twice continuously differentiable function. We say $\pi$ is $\beta$-smooth if $V$ has bounded Hessian for all $x \in \mathbb{R}^d$:

$$-\beta I \preceq \nabla^2 V(x) \preceq \beta I.$$

We say $\pi$ is $\alpha$-strongly log-concave if

$$0 \prec \alpha I \preceq \nabla^2 V(x)$$

for all $x \in \mathbb{R}^d$.

## B.3 FUNCTIONAL INEQUALITIES

For nonnegative smooth $f : \mathbb{R}^d \to \mathbb{R}_{\geq 0}$, let the entropy of $f$ with respect to probability distribution $\pi$ be

$$\text{Ent}_\pi[f] = \mathbb{E}_\pi[f \ln(f/\mathbb{E}_\pi[f])].$$

We say $\pi$ satisfies a log-Sobolev inequality (LSI) with constant $C_{LS}$ if for all smooth functions $f$,

$$\text{Ent}_\pi[f^2] \leq 2C_{LS}\mathbb{E}_\pi[||\nabla f||^2]$$

and $\pi$ satisfies a Poincare inequality (PI) with constant $C_{PI}$ if $\text{Var}_\pi[f] \leq 2C_{PI}\mathbb{E}_\pi[||\nabla f||^2]$. The log-Sobolev inequality implies Poincare inequality: $C_{PI} \leq C_{LS}$. Due to the Bakry-Emery criterion, if $\pi$ is $\alpha$-strongly log-concave then $\pi$ satisfies LSI with constant $C_{LS} = 1/\alpha$.

LSI and PI are equivalent to statements about exponential ergodicity of the continuous-time Langevin diffusion, which is defined by the Stochastic Differential Equation

$$d\bar{X}_t^\pi = \nabla \log \pi(\bar{X}_t^\mu) \, dt + \sqrt{2} \, dB_t.$$

Specifically, let $\pi_t$ denote the law of the diffusion at time $t$ initialized from $\pi_0$ then a LSI is equivalent to the inequality

$$\mathcal{D}_{\text{KL}}(\pi_t||\pi) \leq \exp(-2t/C_{LS})\mathcal{D}_{\text{KL}}(\pi_0||\pi)$$

holding for an arbitrary initial distribution $\pi_0$. Similarly, a PI is equivalent to $\chi^2(\pi_t||\pi) \leq \exp(-2t/C_{PI})\chi^2(\pi_0||\pi)$. Here $\mathcal{D}_{KL}(P,Q) = \mathbb{E}_P[\log \frac{dP}{dQ}]$ is the Kullback-Liebler divergence and $\chi^2(P,Q) = \mathbb{E}_Q[(dP/dQ - 1)^2]$ is the $\chi^2$-divergence. See Bakry et al. (2014); Van Handel (2014) for more background.

### B.4 CONCENTRATION

**Proposition 1** (Concentration of Brownian motion, (Chewi et al., 2021, Lemma 32))**.** *Let* $(B_t)_{t \geq 0}$ *be a standard Brownian motion in* $\mathbb{R}^d$*. Then, if* $\lambda \geq 0$ *and* $h \leq 1/(4\lambda)$,

$$\mathbb{E}\left[\exp\left(\lambda \sup_{t \in [0,h]} ||B_t||^2\right)\right] \leq \exp(6dh\lambda)$$

*In particular, for all* $\eta \geq 0$

$$\mathbb{P}\left[\sup_{t \in [0,h]} ||B_t||^2 \geq \eta\right] \leq \exp\left(-\frac{\eta^2}{6dh}\right)$$

**Proposition 2.** *Suppose a random non-negative real variable* $Z$ *satisfies*

$$\forall t : \mathbb{P}[Z \geq D + t] \leq 2\exp(-\gamma t^2)$$

*for some* $D \geq 0, \gamma > 0$*. Then there exists numerical constant* $C$ *s.t.*

$$\mathbb{E}[Z^p] \leq Cp^{p/2}(D + \gamma^{-1/2})^p$$

*Proof.* For some $R \geq D$ to be chosen later

$$\begin{aligned}
\mathbb{E}[Z^p] &= \int_0^\infty \mathbb{P}[Z^p \geq x]dx \\
&= \int_0^{R^p} \mathbb{P}[Z^p \geq x]dx + \int_{R^p}^\infty \mathbb{P}[Z^p \geq x]dx \\
&\leq \int_0^{R^p} 1dx + \int_R^\infty \mathbb{P}[Z \geq y]d(y^p) \\
&\leq R^p + 2p\int_R^\infty y^{p-1}\exp(-\gamma(y-D)^2)dy \\
&\leq R^p + p2^p\left(\int_R^\infty z^{p-1}\exp(-\gamma z^2)dz + D^{p-1}\int_R^\infty \exp(-\gamma z^2)dz\right) \\
&\leq R^p + 2^{p-1}(\gamma^{-p/2}(p/2)^{p/2} + pD^{p-1}\gamma^{-1/2}\sqrt{\pi})
\end{aligned}$$

where in the last inequality, we make a change of variable $u = \gamma z^2$ and note that $2p\int z^{p-1}\exp(-\gamma z^2)dz = \gamma^{-p}p\int u^{p/2-1}\exp(-u)du = \Gamma(p/2) \leq (p/2)^{p/2}$ and $\int_0^\infty \exp(-\gamma z^2)dz = (2\gamma)^{-1/2}\sqrt{2\pi}/2$. Take $R = D$ gives the desired result. $\square$

**Proposition 3** ((Bakry et al., 2014, 5.4.2), restated in (Lee et al., 2022a, Lemma E.2) )**.** *Suppose* $\pi : \mathbb{R}^d \to \mathbb{R}_{\geq 0}$ *satisfies LSI with constant* $1/\alpha$*. Let* $f : \mathbb{R}^d \to \mathbb{R}$ *be a L-Lipschitz function then*

$$\mathbb{P}_{x \sim \pi}[|f(x) - \mathbb{E}_\pi[f(x)]| \geq t] \leq \exp\left(-\frac{\alpha t^2}{2L^2}\right)$$

**Proposition 4** (Sub-Gaussian concentration of norm for strongly log concave measures)**.** *Let* $V : \mathbb{R}^d \to \mathbb{R}$ *be a* $\alpha$*-strongly convex and* $\beta$*-smooth function. Let* $\kappa = \beta/\alpha$*. Let* $\pi$ *be the probability measure with* $\pi(x) \propto \exp(-V(x))$*. Let* $x_* = \arg\min_x V(x)$ *then for* $D = 5\sqrt{\frac{d}{\alpha}}\ln(10\kappa)$ *we have*

$$\mathbb{P}_{x \sim \pi}[||x - x_*|| \geq D + t] \leq \exp(-\alpha t^2/4)$$

*thus by Proposition 2, for* $p \geq 1$.

$$\mathbb{E}_\pi[||x - x^*||^p]^{1/p} \leq O(1)\sqrt{p}\sqrt{\frac{d}{\alpha}}\ln(10\kappa)^p$$

*Proof.* By (Lee et al., 2022a, Lemma E.3), let $\bar{x} = \mathbb{E}_\pi[x]$ then $||\bar{x} - x^*|| \leq \frac{1}{2}\sqrt{\frac{d}{\alpha}}\ln(10\kappa)$. By Proposition 3, for any unit vector $v \in \mathbb{R}^d$, the function $\langle v, x - \bar{x}\rangle$ is 1-Lipschitz, since $|\langle v, x\rangle -$

$\langle v, y \rangle| \leq \sqrt{||v||_2}||x - y||_2 = ||x - y||_2$. Thus, by Proposition 3, $\langle v, x - \bar{x} \rangle$ has mean 0 and sub-Gaussian concentration for all unit vector $v$, thus $x - \bar{x} \rangle$ is a sub-Gaussian random vector. From sub-Gaussianity, a standard argument (see e.g. Theorem 1.19 of Rigollet & Hütter (2017)) shows that

$$\mathbb{P}_\pi \left[ ||x - \bar{x}|| \geq 4\sqrt{\frac{d}{\alpha}} + t \right] \leq \exp(-\alpha t^2/4)$$

thus by triangle inequality, using that $||\bar{x} - x^*|| \leq \sqrt{\frac{d}{\alpha}} \frac{1}{2} \ln(10\kappa)$, we have

$$\mathbb{P}_\pi \left[ ||x - x^*|| \geq (4 + 1/2 \ln(10\kappa))\sqrt{\frac{d}{\alpha}} + t \right] \leq \mathbb{P}_\pi \left[ ||x - \bar{x}|| \geq 4\sqrt{\frac{d}{\alpha}} + t \right] \leq \exp(-\alpha t^2/4)$$

$\square$

**Proposition 5** (Normalization factor bound). *Let $V : \mathbb{R}^d \to \mathbb{R}$ be a $\alpha$-strongly convex and $\beta$-smooth function. Let $\pi$ be the probability measure defined by $\pi(x) \propto \exp(-V(x))$ and $Z := Z_\pi = \int \exp(-V(x))dx$ be its normalization factor. For any $y \in \mathbb{R}^d$*

$$\exp\left(-V(y) + \frac{||\nabla V(y)||^2}{2\beta}\right)(2\pi\beta^{-1})^{d/2} \leq Z \leq \exp\left(-V(y) + \frac{||\nabla V(y)||^2}{2\alpha}\right)(2\pi\alpha^{-1})^{d/2}$$

*Let $y = x^* = \arg\min V(x)$ and assume w.l.o.g. $V(y) = 0$ gives*

$$\frac{d}{2}\ln\frac{1}{\beta} \leq \ln Z_\pi - \frac{d}{2}\ln(2\pi) \leq \frac{d}{2}\ln\frac{1}{\alpha}$$

*Proof.* Since $\alpha I \preceq \nabla^2 V(x) \preceq \beta I$,

$$\langle \nabla V(y), x - y \rangle + \alpha ||x - y||^2/2 \leq V(x) - V(y) \leq \langle \nabla V(y), x - y \rangle + \beta ||x - y||^2/2$$

$$Z \leq \int \exp(-V(y) - \langle \nabla V(y), x - y \rangle - \alpha ||x - y||^2/2)dx$$

$$= \exp\left(-V(y) + \frac{||\nabla V(y)||^2}{2\alpha}\right) \int \exp\left(-\frac{\alpha ||(x - y) + \alpha^{-1}\nabla V(y)||^2}{2}\right)dx$$

$$= \exp\left(-V(y) + \frac{||\nabla V(y)||^2}{2\alpha}\right)(2\pi\alpha^{-1})^{d/2}$$

The lower bound follows similarly. The second statement follows from the first since $\nabla V(x^*) = 0$.

$\square$

## B.5 GIRSANOV'S THEOREM

**Theorem 3** (Girsanov's Theorem (Karatzas & Shreve, 1991, Chapter 3.5)). *Let $(X_t)_{t\geq 0}$ be stochastic processes adapted to the same filtration. Let $P_T$ and $Q_T$ be probability measure on the path space $C([0, T]; \mathbb{R}^d)$ s.t. $X_t$ evolved according to*

$$dX_t = b_t^P dt + \sqrt{2}dB_t^P \text{ under } P_T$$
$$dX_t = b_t^Q dt + \sqrt{2}dB_t^Q \text{ under } Q_T$$

*Assume that Novikov's condition*

$$\mathbb{E}_{Q_T}\left[\exp\left(\frac{1}{4}\int_0^T ||b_t^P - b_t^Q||^2 dt\right)\right] < \infty \tag{5}$$

*holds. Then*

$$\frac{dP_T}{dQ_T} = \exp\left(\int_0^T \frac{1}{\sqrt{2}}\langle b_t^P - b_t^Q, dB_t^Q \rangle - \frac{1}{4}\int_0^T ||b_t^P - b_t^Q||^2 dt\right) \tag{6}$$

**Lemma 3** (Application of Girsanov with approximation argument (Chen et al., 2023, Equation 5.5, Proof of Theorem 9)). *Let $(X_t)_{t\geq 0}$ be stochastic processes adapted to the same filtration. Let $P_T$ and $Q_T$ be probability measure on the path space $C([0,T];\mathbb{R}^d)$ s.t. $X_t$ evolved according to*

$$dX_t = b_t^P dt + \sqrt{2}dB_t^P \text{ under } P_T$$
$$dX_t = b_t^Q dt + \sqrt{2}dB_t^Q \text{ under } Q_T$$

*Suppose $\mathbb{E}_{Q_T}[\int_0^T ||b_t^P - b_t^Q||^2 dt] < \infty$ then*

$$2d_{TV}(Q_T||P_T)^2 \leq \mathcal{D}_{\mathrm{KL}}(Q_T||P_T) \leq \mathbb{E}_{Q_T}\left[\int_0^T ||b_t^P - b_t^Q||^2 dt\right]$$

**Lemma 4** (Corollary of Theorem 3, (Chewi et al., 2021, Corollary 20)). *With the setup and preconditions in Theorem 3, For any event $\mathcal{E}$,*

$$\mathbb{E}_{Q_T}\left[\left(\frac{dP_T}{dQ_T}\right)^q \mathbf{1}_{\mathcal{E}}\right] \leq \sqrt{\mathbb{E}_{Q_T}\left[\exp\left(q^2 \int_0^T ||b_t^P - b_t^Q||^2 dt\right)\mathbf{1}_{\mathcal{E}}\right]}$$

### B.6 MIXTURE POTENTIAL

**Notation for indexing components.** Let $I = [K]$ be the set of indices $i$ for the components $\mu_i$ of the mixture distribution $\mu$. We will need to work with subsets $S$ of $I$ and the mixture distribution forms by components $\mu_i$ for $i \in S$.

**Definition 2.** *For $S \subseteq I$, let $p_S = \sum_{i \in S} p_i$, and $\mu_S = p_S^{-1} \sum_{i \in S} p_i \mu_i$. Let $V_S = -\log \mu_S$.*

*If $S = I$ we omit the subscript $S$.*

**Derivative computations.** For future use, we compute the derivatives of $V$.

**Proposition 6** (Gradient of $V$).

$$\nabla V(x) = \frac{\sum p_i \mu_i(x) \nabla V_i(x)}{\mu(x)} \tag{7}$$

*Consequently, $||\nabla V(x)|| \leq \max ||\nabla V_i(x)||$.*

*Proof.* The statement follows from

$$\nabla V(x) = \nabla \log \mu(x) = \frac{\nabla \mu(x)}{\mu(x)}$$

and

$$\nabla \mu(x) = \nabla(\sum p_i Z_i^{-1} \exp(-V_i(x))) = -\sum p_i \mu_i(x) \nabla V_i(x).$$

$\square$

**Proposition 7** (Hessian of $V$).

$$\nabla^2 V(x) = \frac{\sum_i p_i \mu_i(x) \nabla^2 V_i(x)}{\mu(x)} - \sum_{i,j} \frac{p_i p_j \mu_i(x) \mu_j(x)(\nabla V_i(x) - \nabla V_j(x))(\nabla V_i(x) - \nabla V_j(x))^\top}{4\mu^2(x)} \tag{8}$$

*hence if $\nabla^2 V_i \preceq \beta I$ for all $i \in I$ then $\nabla^2 V(x) \preceq \beta I$.*

*Proof.* Let $Z_i = \int \exp(-V_i(x))dx$ be the normalization factor of $\mu_i$. Note that

$$\nabla(\mu_i(x)\nabla V_i(x))$$
$$= \nabla(Z_i^{-1} \exp(-V_i(X))\nabla V_i(x)) = Z_i^{-1} \exp(-V_i(x))(-\nabla V_i(x)\nabla V_i(x)^\top + \nabla^2 V_i(x))$$
$$= \mu_i(x)(\nabla^2 V_i(x) - \nabla V_i(x)\nabla V_i(x)^\top)$$

and $\nabla\mu(x) = -\sum p_i\mu_i(x)\nabla V_i(x)$, thus

$$\nabla^2 V(x)$$
$$= \frac{\nabla(\sum_i p_i\mu_i(x)\nabla V_i(x))}{\mu(x)} - \frac{(\sum p_i\mu_i(x)\nabla V_i(x))\nabla\mu(x)}{\mu^2(x)}$$
$$= \frac{\sum p_i\mu_i(x)(\nabla^2 V_i(x) - \nabla V_i(x)\nabla V_i(x)^\top)}{\mu(x)} + \frac{(\sum p_i\mu_i(x)\nabla V_i(x))(\sum p_i\mu_i(x)\nabla V_i(x))^\top}{\mu^2(x)}$$

Next,

$$\left(\sum p_i\mu_i(x)\nabla V_i(x)\right)\left(\sum p_i\mu_i(x)\nabla V_i(x)\right)^\top - \left(\sum p_i\mu_i\nabla V_i(x)\nabla V_i(x)^\top\right)\left(\sum p_i\mu_i\right)$$
$$= \sum_{i,j} p_i p_j\mu_i(x)\mu_j(x)\nabla V_i(x)\nabla V_j^\top - \sum_{i,j} p_i p_j\mu_i(x)\mu_j(x)\nabla V_i(x)\nabla V_i(x)^\top$$
$$= \frac{1}{2}\sum_{i\neq j} p_i p_j\mu_i(x)\mu_j(x)(\nabla V_i(x)\nabla V_j^\top + \nabla V_j(x)\nabla V_i^\top - \nabla V_i(x)\nabla V_i^\top - \nabla V_j(x)\nabla V_j^\top)$$
$$= -\frac{1}{2}\sum_{i\neq j} p_i p_j\mu_i(x)\mu_j(x)(\nabla V_i(x) - \nabla V_j(x))(\nabla V_i(X) - \nabla V_j(x))^\top$$

thus the first statement follows. The second statement follows from noticing that $(\nabla V_i(x) - \nabla V_j(x))(\nabla V_i(X) - \nabla V_j(x))^\top \succeq 0$. $\qquad\square$

### B.7 PROPERTIES OF SMOOTH AND STRONGLY LOG-CONCAVE DISTRIBUTION

We record the consequences of $\alpha$-strongly log-concave and $\beta$-smooth that we will use.

**Lemma 5.** *Suppose $\mu_i$ is $\alpha$-strongly log-concave and $\beta$-smooth then for $\kappa = \beta/\alpha$, $u_i = \arg\min V_i(x)$, $D = 5\sqrt{\frac{d}{\alpha}}\ln(10\kappa)$, and $c_z = \frac{d}{2}\ln\kappa$, we have*

1. *For all $x$ : $||\nabla^2 V_i(x)||_{OP} \leq \beta$ and $||\nabla V_i(x)|| \leq \beta||x - u_i||$*

2. *$\alpha||x - u_i||^2 \leq V_i(x) \leq \beta||x - u_i||^2$.*

   *Consequently, for $Z_i = \int \mu_i(x)dx$, there exists $z_+ \leq z_-$ with $z_+ = z_- - c_z$ s.t.*
   $$\exp(-\beta||x - u_i||^2 - z_-) \leq \mu_i(x) = Z_i^{-1}\exp(-V_i(x)) \leq \exp(-\alpha||x - u_i||^2 - z_+)$$

3. *Sub-gaussian concentration:*
   $$\mathbb{P}[||x - u_i|| \geq D + t] \leq \exp(-\alpha t^2/4)$$
   *By Proposition 2, this implies that for all $p$*
   $$\mathbb{E}_{\mu_i}[||x - u_i||^p] \lesssim_p D^p.$$

4. *$\mu_i$ satisfies a LSI with constant $C_{LS} = \frac{1}{\alpha}$.*

*Proof.* This is due to Proposition 5 and Proposition 2, and the fact that $\nabla V_i(u_i) = 0$ for $u_i = \arg\min V_i(x)$. $\qquad\square$

### B.8 BASIC MATHEMATICAL FACTS

**Proposition 8.** *For any constant $a > 0, b, p \in \mathbb{N}_{\geq 0}$ $f(x) = \exp(-ax - b)x^p$ is decreasing on $[p/a, +\infty)$*

*Proof.* Let $g(x) = \log f(x) = -ax - b + p\log x$ and observe that
$$g'(x) = -a + p/x \leq 0$$
when $x \geq p/a$, so the claim follows by integrating. $\qquad\square$

**Proposition 9.** *Let $P_1, \ldots, P_k, Q_1, \ldots, Q_k$ be distributions s.t. $d_{TV}(P_i, Q_i) \leq \epsilon_i$. Let $\alpha_1, \cdots, \alpha_k, \beta_1, \cdots, \beta_k$ be s.t. $\alpha_i, \beta_i \geq 0 \forall i$ and $\sum_i \alpha_i = \sum_i \beta_i = 1$. Then*

$$d_{TV}(\sum_i \alpha_i P_i, \sum_i \alpha_i Q_i) \leq \sum_i \alpha_i \epsilon_i$$

*and*

$$d_{TV}(\sum_i \alpha_i Q_i, \sum_i \beta_i Q_i) \leq \frac{1}{2} \sum_i |\alpha_i - \beta_i|$$

*Proof.* By triangle inequality

$$2d_{TV}(\sum_i \alpha_i P_i, \sum_i \alpha_i Q_i) = \int_{x \in \Omega} |\sum_i \alpha_i P_i(x) - \sum_i \alpha_i Q_i(x)| dx$$

$$\leq \int_{x \in \Omega} \sum_i \alpha_i |P_i(x) - Q_i(x)| dx = 2 \sum_i \alpha_i d_{TV}(P_i, Q_i)$$

Similarly,

$$2d_{TV}(\sum_i \alpha_i Q_i, \sum_i \beta_i Q_i) = \int_{x \in \Omega} |\sum_i \alpha_i Q_i(x) - \sum_i \beta_i Q_i(x)| dx$$

$$\leq \int_{x \in \Omega} \sum_i |\alpha_i - \beta_i| Q_i(x) dx = \sum_i |\alpha_i - \beta_i|$$

$\square$

## C  LOG-SOBOLEV INEQUALITY FOR WELL-CONNECTED MIXTURES

In this section, we show that the mixture $\sum p_i \mu_i$ has a good log-Sobolev constant if its component distributions $\mu_i$ have high overlap. The below Theorem 4 corresponds to Theorem 2 of the main text.

**Definition 3.** *For distributions $\nu, \pi$, let $\delta(\nu, \pi) = \int \min\{\nu(x), \pi(x)\} dx$ be the overlap of $\nu$ and $\pi$. Let $\delta_{ij}$ denote $\delta(\mu_i, \mu_j)$. Note that*

$$1 - \delta(\nu, \pi) = \int (\nu(x) - \min\{\nu(x), \pi(x)\}) dx = \int_{x:\nu(x) \geq \pi(x)} (\nu(x) - \pi(x)) dx = d_{TV}(\nu, \pi).$$

**Theorem 4.** *Let $G$ be the graph on $I$ where $\{i, j\} \in E(G)$ iff $\mu_i, \mu_j$ have high overlap i.e.*

$$\delta_{ij} := \int \min\{\mu_i(x), \mu_j(x)\} dx \geq \delta.$$

*Suppose $G$ is connected. Let $M \leq |I|$ be the diameter of $G$. The mixture distribution $\mu = \sum_{i \in I} p_i \mu_i$ has*

1. *Poincare constant (Madras & Randall, 2002, Theorem 1.2)*

$$C_{PI}(\mu) \leq \frac{4M}{\delta} \max_{i \in I} \frac{C_{PI}(\mu_i)}{p_i}$$

2. *Log Sobolev constant*

$$C_{LS}(\mu) \leq \frac{4M C_{LS}(p)}{\delta} \max_i \frac{C_{LS}(\mu_i)}{p_i}$$

*where for $p_* = \min_i p_i$, $C_{LS}(p) = 1 + \log(p_*^{-1})$ is the log Sobolev constant of the instant mixing chain for $p$. Hence*

$$C_{LS}(\mu) \leq C_{|I|, p_*} \delta^{-1} \max_i C_{LS}(\mu_i)$$

*where $C_{|I|, p_*} = 4|I|^2(1 + \log(p_*^{-1})) p_*^{-1}$ only depends on $|I|$ and $p_*$*

Below we fix a test function $f$ s.t. $\mathbb{E}_\mu[f^2] \leq \infty$. Let

$$C_{i,j} = \int\int (f(x) - f(y))^2 \mu_i(x)\mu_j(x)dxdy. \tag{9}$$

**Lemma 6** (Triangle inequality).

$$C_{i_0,i_\ell} \leq \ell \sum_{j=0}^{\ell-1} C_{i_j,i_{j+1}}$$

*Proof.* Without loss of generality, assume $i_j = j$ for all $j$. Then

$$C_{i_0,i_\ell} = \int\int (f(x_0) - f(x_\ell))^2 \mu_0(x_0)\mu_\ell(x_\ell)dx_0 dx_\ell$$

$$= \int_{x_0} \cdots \int_{x_\ell} (f(x_0) - f(x_1) + \cdots + f(x_{\ell-1}) - f(x_\ell))^2 \prod_{j=0}^{\ell} \mu_j(x_j)dx_0 d_{x_1} \dots dx_\ell$$

$$\leq \int_{x_0} \cdots \int_{x_\ell} \ell \left( \sum_{j=0}^{\ell-1} (f(x_j) - f(x_{j+1}))^2 \right) \prod_{j=0}^{\ell} \mu_j(x_j)dx_0 d_{x_1} \dots dx_\ell$$

$$= \ell \sum_{j=0}^{\ell-1} \int_{x_j} \int_{x_{j+1}} (f(x_j) - f(x_{j+1}))^2 \mu_j(x_j)\mu_{j+1}x_{j+1}d_{x_j}d_{x_{j+1}} = \ell \sum_{j=0}^{\ell-1} C_{j,j+1}$$

where the inequality is Holder's inequality. $\qquad\qquad\square$

The following comes from (Madras & Randall, 2002, Proof of Theorem 1.2)

**Lemma 7.** *If $\int \min\{\mu_i(x), \mu_j(x)\}dx \geq \delta$ then*

$$C_{i,j} \leq \frac{2(2-\delta)}{\delta}(\mathrm{Var}_{\mu_i}(f) + \mathrm{Var}_{\mu_j}(f)).$$

**Proposition 10** (Variance decomposition).

$$2\,\mathrm{Var}_\mu(f) = \int_x \int_y (f(x) - f(y))^2 \mu(x)\mu(y)dxdy$$

$$= \sum_{i,j} p_i p_j C_{ij}$$

$$= 2\sum_i p_i^2 \,\mathrm{Var}_{\mu_i}(f) + 2\sum_{i<j} p_i p_j C_{ij}$$

*Proof.*

$$\mathrm{Var}_\mu(f) = \int_x \mu(x)f^2(x)dx - \left(\int_x \mu(x)f(x)dx\right)^2$$

$$= \int_x \int_y f^2(x)\mu(x)\mu(y)dxdy - \int_x \int_y \mu(x)f(x)\mu(y)f(y)dxdy$$

$$= \frac{1}{2}\int_x \int_y \mu(x)\mu(y)(f^2(x) + f^2(y) - 2f(x)f(y))dxdy$$

$$= \frac{1}{2}\int_x \int_y \mu(x)\mu(y)(f(x) - f(y))^2 dxdy$$

Since $\mu(x) = \sum_i p_i \mu_i(x)$, we can further rewrite

$$2 \operatorname{Var}_\mu(f) = \int_x \int_y (f(x) - f(y))^2 \left( \sum_i p_i \mu_i(x) \right) \left( \sum_i p_i \mu_i(y) \right) dx dy$$

$$= \int_x \int_y (f(x) - f(y))^2 \left( \sum_{i,j} p_i p_j \mu_i(x) \mu_j(y) \right) dx dy$$

$$= \sum_{i,j} p_i p_j \int_x \int_y (f(x) - f(y))^2 \mu_i(x) \mu_j(y) dx dy$$

$$= \sum_{i,j} p_i p_j C_{ij}$$

$$= \sum_i p_i^2 C_{ii} + \sum_{i<j} (C_{ij} + C_{ji})$$

$$= 2 \sum_i p_i^2 \operatorname{Var}_{\mu_i}[f] + 2 \sum_{i<j} p_i p_j C_{ij}$$

where the last equality is because $C_{ij} = C_{ji}$. $\qquad\square$

**Lemma 8.** *For $i, j$ let $\gamma_{ij}$ be the shortest path in $G$ from $i$ to $j$ and let $|\gamma_{ij}|$ be its length i.e. the number of edges in that path. For $u, v$, let $uv$ denote the edge $\{u, v\}$ of $G$ if it is in $E(G)$. Let $M = \max_{ij} |\gamma_{ij}|$ be the diameter of $G$. Then*

$$\sum_{i<j} p_i p_j C_{ij} \leq \sum_{uv \in E(G)} \left( C_{uv} \sum_{i<j : uv \in \gamma_{ij}} p_i p_j |\gamma_{ij}| \right)$$

$$\leq \frac{M(2-\delta)}{\delta} \sum_u \operatorname{Var}_{\mu_u}(f)$$

$$\leq \frac{M(2-\delta)}{\delta} \sum_u C_{PI}(\mu_u) \mathbb{E}_{\mu_u}[||\nabla f||^2]$$

*Proof.*

$$\sum_{i<j} p_i p_j C_{ij} \leq \sum_{uv \in E(G)} \left( C_{uv} \sum_{i<j : uv \in \gamma_{ij}} p_i p_j |\gamma_{ij}| \right) \leq M \sum_{uv \in E(G)} \left( C_{uv} \sum_{i<j : uv \in \gamma_{ij}} p_i p_j \right)$$

By Lemma 7 and the definition of $G$, $C_{uv} \leq \frac{2(2-\delta)}{\delta} (\operatorname{Var}_{\mu_u}(f) + \operatorname{Var}_{\mu_v}(f))$, thus

$$\sum_{i<j} p_i p_j C_{ij} \leq \frac{2M(2-\delta)}{\delta} \sum_{uv \in E(G)} \left[ (\operatorname{Var}_{\mu_u}(f) + \operatorname{Var}_{\mu_v}(f)) \sum_{i<j : uv \in \gamma_{ij}} p_i p_j \right]$$

$$\leq \frac{2M(2-\delta)}{\delta} \sum_u \left[ \operatorname{Var}_{\mu_u}(f) \sum_{v, uv \in E(G), i<j : uv \in \gamma_{ij}} p_i p_j \right]$$

$$= \frac{2M(2-\delta)}{\delta} \sum_u \left[ \operatorname{Var}_{\mu_u}(f) \sum_{i<j : u \in \gamma_{ij}} p_i p_j \right]$$

$$\leq \frac{M(2-\delta)}{\delta} \sum_u \operatorname{Var}_{\mu_u}(f)$$

$$\leq \frac{M(2-\delta)}{\delta} \sum_u C_{PI}(\mu_u) \mathbb{E}_{\mu_u}[||\nabla f||^2]$$

$\qquad\square$

**Proposition 11.** *For $C_{i,j}$ be as in Eq. (9)*

$$C_{i,j} = \frac{1}{2}(\text{Var}_{\mu_i}(f) + \text{Var}_{\mu_j}(f) + (\mathbb{E}_{\mu_i}[f] - \mathbb{E}_{\mu_j}[f])^2)$$

*Proof.* Let $\nu = \frac{1}{2}\mu_i + \frac{1}{2}\mu_j$. We write $\text{Var}(\nu)$ in two ways. First, $\mathbb{E}_\nu[f] = \frac{1}{2}(E_{\mu_i}[f] + E_{\mu_j}[f])$ thus

$$Var_\nu(f) = \mathbb{E}_\nu[f^2] - (\mathbb{E}_\nu[f])^2 = \frac{1}{2}(\mathbb{E}_{\mu_i}[f^2] + \mathbb{E}_{\mu_j}[f^2]) - \frac{1}{4}(E_{\mu_i}[f] + E_{\mu_j}[f])^2$$

$$= \frac{1}{2}\sum_{k \in \{i,j\}}(\mathbb{E}\mu_k[f^2] - (\mathbb{E}_{\mu_k}[f])^2) + \frac{1}{4}(E_{\mu_i}^2[f] + E_{\mu_j}^2\mu_j[f] - 2E_{\mu_i}[f]E_{\mu_j}[f])$$

$$= \frac{1}{2}(\text{Var}_{\mu_i}[f] + \text{Var}_{\mu_j}[f]) + \frac{1}{4}(\mathbb{E}_{\mu_i}[f] - \mathbb{E}_{\mu_j}[f])^2$$

On the other hand, by Proposition 10,

$$\text{Var}_\nu(f) = \frac{1}{4}(\text{Var}_{\mu_i}(f) + \text{Var}_{\mu_j}(f)) + \frac{1}{2}C_{ij}$$

Rearranging terms gives the desired equation. $\qquad\square$

**Proposition 12.** *Let $g \equiv f^2$. Let the projection of $g$ on $I$ be defined by $\bar{g}(i) = \mathbb{E}_{\mu_i}[g]$. Then*

$$\text{Ent}[f^2] = \sum_{i \in I} p_i \text{Ent}_{\mu_i}[f^2] + \text{Ent}_p[\bar{g}]$$

*Proof.*

$$\text{Ent}[f^2] = \int \mu(x)g(x)\log g(x)dx - \mathbb{E}_\mu[g(X)]\log(\mathbb{E}_\mu[g(x)])$$

$$= \int \left(\sum_i p_i\mu_i(x))g(x)\log g(x)dx - \mathbb{E}_\mu[g(x)]\log(\mathbb{E}_\mu[g(x)]\right)$$

$$= \sum_i p_i \left(\int \mu_i(x)g(x)\log g(x)dx - \mathbb{E}_{\mu_i}[g(x)]\log(\mathbb{E}_{\mu_i}[g(x)])\right)$$

$$+ \sum_i p_i\bar{g}(i)\log\bar{g}(i) - \mathbb{E}_\mu[g(x)]\log(\mathbb{E}_\mu[g(x)])$$

where in the last equality, we use the definition of $\bar{g}(i)$. Note that

$$\mathbb{E}_{i \sim p}[\bar{g}(i)] = \sum_i p_i\bar{g}(i) = \sum_i \left(p_i \int \mu_i(x)g(x)dx\right) = \int \left(\sum_i p_i\mu_i\right)g(x) = \mathbb{E}_\mu[g(x)]$$

thus

$$\text{Ent}[f^2] = \sum_i p_i \text{Ent}_{\mu_i}[f^2] + \text{Ent}_{i \sim p}[\bar{g}(i)]$$

$\qquad\square$

**Proposition 13.** *Let $\bar{g}$ be defined as in Proposition 12, then*

$$(\sqrt{\bar{g}}(i) - \sqrt{\bar{g}}(j))^2 \le \text{Var}_{\mu_i}[f^2] + \text{Var}_{\mu_j}[f^2] + (\mathbb{E}_{\mu_i}[f] - \mathbb{E}_{\mu_j}[f])^2 = 2C_{ij}$$

*Proof.* The first inequality comes from (Schlichting, 2019, Proof of Lemma 3) and the second part from Proposition 11. $\qquad\square$

**Proposition 14** (Log Sobolev inequality for the instant mixing chain, (Diaconis & Saloff-Coste, 1996, Theorem A.1))**.** *Let $p$ be the distribution over $I$ where the probability of sampling $i \in I$ is $p_i$. For a function $h : I \to \mathbb{R}_{\ge 0}$*

$$\text{Ent}_p[h] \le C_p \text{Var}_p[\sqrt{h}]$$

*with $C_p = \ln(4p_*^{-1})$ with $p_* = \min_i p_i$.*

**Lemma 9.** *With $\bar{g}$ defined as in Proposition 12,*

$$\mathrm{Var}_p[\sqrt{\bar{g}}] = \sum_{i<j} p_i p_j (\sqrt{\bar{g}}(i) - \sqrt{\bar{g}}(j))^2 \leq 2 \sum_{i<j} p_i p_j C_{ij}$$

*Proof of Theorem 4 part 2.* We can rewrite

$$\mathrm{Ent}_\mu[f^2] = \sum_{i \in I} p_i \mathrm{Ent}_{\mu_i}[f^2] + \mathrm{Ent}_p[\bar{g}]$$

$$\leq_{(1)} \sum_i p_i C_{LS}(\mu_i)\mathbb{E}_{\mu_i}[||\nabla f||^2] + C_{LS}(p) \mathrm{Var}_p(\sqrt{\bar{g}})$$

$$\leq_{(2)} \sum_i p_i C_{LS}(\mu_i)\mathbb{E}_{\mu_i}[||\nabla f||^2] + 2C_{LS}(p) \sum_{i<j} p_i p_j C_{ij}$$

$$\leq_{(3)} \sum_i p_i C_{LS}(\mu_i)\mathbb{E}_{\mu_i}[||\nabla f||^2] + \frac{2M(2-\delta)C_{LS}(p)}{\delta} \sum_u C_{PI}(\mu_u)\mathbb{E}_{\mu_u}[||\nabla f||^2]$$

$$\leq_{(4)} \frac{4MC_{LS}(p)}{\delta} \max_i \{\frac{C_{LS}(\mu_i)}{p_i}\} \sum_i p_i \mathbb{E}_{\mu_i}[||\nabla f||^2]$$

$$= \frac{4MC_{LS}(p)}{\delta} \max_i \{\frac{C_{LS}(\mu_i)}{p_i}\}\mathbb{E}_\mu[||\nabla f||^2]$$

where (1) is due to definition of $C_{LS}(\mu_i)$ and Proposition 14, (2) is due to Lemma 9, (3) is due to Lemma 8, and (4) is due to $C_{PI}(\mu_i) \leq C_{LS}(\mu_i)$ and $C_{LS}(p), M \geq 1$. □

## D    INITIALIZATION ANALYSIS

For the continuous Langevin diffusion $(\bar{X}_t)_{t \geq 0}$ initialized at a bounded support distribution $\nu_0$, we bound $\mathcal{R}_q(\mathcal{L}(\bar{X}_h)||\mu)$ for some small $h$. Consequently, for $\mu$ being the stationary distribution of the Langevin diffusion and satisfying a LSI with constant $C_{LS}$, we can use the fact that $\mathcal{D}_{\mathrm{KL}}(\mathcal{L}(\bar{X}_t)||\mu) \leq \exp(-\frac{t-h}{C_{LS}})\mathcal{D}_{\mathrm{KL}}(\mathcal{L}(\bar{X}_h)||\mu)$ to show that $\bar{X}_t$ converges to $\mu$.

**Lemma 10** (Initialization bound). *Let $\mu = \sum_{i \in I} p_i \mu_i$ be a mixture of distributions $\mu_i \propto \exp(-V_i(x))$ which are $\alpha$-strongly log concave and $\beta$-smooth. Let $V(x) = -\ln \mu(x)$. Let $(\bar{\nu}_t)_{t \in [0,h]}, (\nu_t)_{t \in [0,h]}$ be respectively the distribution of the continuous Langevin diffusion and the LMC with step size $h$ and score function $\nabla V$ initialized at $\delta_x$. Let $G(x) := \max_i ||\nabla V_i(x)||$. Suppose $h \leq 1/(30\beta)$ then for $q \in (2, \frac{1}{10\beta h})$,*

$$\mathcal{R}_q(\bar{\nu}_h||\nu_h) \leq O(q^2 h(G^2(x) + \beta^2 dh)),$$

$$\mathcal{R}_{q-1}(\nu_h||\mu) \leq \frac{d}{2}\ln((2\alpha h)^{-1}) + \alpha^{-1}G(x)$$

*and*

$$R_{q/2}(\bar{\nu}_h||\mu) \leq O(q^2 h(G^2(x) + \beta^2 dh)) + \frac{d}{2}\ln((2\alpha h)^{-1}) + \alpha^{-1}G^2(x)$$

*If we replace $\delta_x$ with any $\nu_0$ then by weak convexity of Renyi divergence (Lemma 2), the claim holds when we replace $G(x)$ with $G_\nu = \sup_{x \in supp(\nu_0)} G(x)$.*

**Proposition 15.** *Let $\nu = \mathcal{N}(y, \sigma^2 I)$. If $\pi(x) \propto \exp(-W(x))$ is $\alpha$-strongly log concave and $\beta$-Lipschitz and $\sigma^2 \beta \leq 1/2$ then*

$$\mathcal{R}_\infty(\nu||\pi) \leq -\frac{d}{2}\ln(\alpha\sigma^2) + ||\nabla W(y)||^2/\alpha$$

*Proof.* Since $\alpha I \preceq \nabla^2 W(x) \preceq \beta I$,

$$\langle \nabla W(y), x - y \rangle + \alpha ||x - y||^2/2 \leq W(x) - W(y) \leq \langle \nabla W(y), x - y \rangle + \beta ||x - y||^2/2$$

By Proposition 5, we can upper bound the normalization factor $Z = \int \exp(-W(x))dx$ by $\exp\left(-W(y) + \frac{||\nabla W(y)||^2}{2\alpha}\right)(2\pi\alpha^{-1})^{d/2}$.

For $x \in \mathbb{R}^d$, using the upper bound on $Z$

$$
\begin{aligned}
\nu(x)/\pi(x) &= (2\pi\sigma^2)^{-d/2} Z \exp\left(-\frac{||x-y||^2}{2\sigma^2} + W(x)\right) \\
&\leq (\alpha\sigma^2)^{-d/2} \exp\left(W(x) - W(y) + \frac{||\nabla W(y)||^2}{2\alpha} - \frac{||x-y||^2}{2\sigma^2}\right) \\
&= (\alpha\sigma^2)^{-d/2} \exp^{||\nabla W(y)||^2(\frac{1}{2\alpha} + \frac{\sigma^2}{2(1-\beta\sigma^2)})} \exp^{-(\sqrt{\frac{(1-\beta\sigma^2)||x-y||^2}{2\sigma^2}} - \sqrt{\frac{\sigma^2||\nabla W(y)||^2}{2(1-\beta\sigma^2)}})^2)} \\
&\leq (\alpha\sigma^2)^{-d/2} \exp\left(||\nabla W(y)||^2 \frac{1-(\beta-\alpha)\sigma^2}{2\alpha(1-\beta\sigma^2)}\right) \\
&\leq (\alpha\sigma^2)^{-d/2} \exp(||\nabla W(y)||^2/\alpha)
\end{aligned}
$$

where the last inequality follows from $1/2 \leq 1 - \beta\sigma^2 \leq 1 - (\beta-\alpha)\sigma^2 \leq 1$. $\qquad\square$

*Proof of Lemma 10.* We apply Theorem 3 with $T = h$, $P_T = (\bar{\nu}_t)_{t\in[0,h]}$ and $Q_T = (\nu_t)_{t\in[0,h]}$. Note that, $b_t^P = -\nabla V(X_t)$ and $b_t^Q = -\nabla V(x)$. We first check that Novikov's condition Eq. (5) holds.

$$
\mathbb{E}_{Q_T}\left[\exp\left(\frac{1}{4}\int_0^T ||b_t^P - b_t^Q||^2 dt\right)\right] = \mathbb{E}\left[\exp\left(\frac{1}{4}\int_0^h ||\nabla V(X_t) - \nabla V(x)||^2 dt\right)\right]
$$

with $(X_t)_{t\in[0,h]}$ be the solution of the interpolated Langevin process i.e.

$$
X_t - x = -t\nabla V(x) + \sqrt{2}B_t
$$

By $\beta$-Lipschitzness of $\nabla V_j$

$$
||\nabla V_j(X_t)|| - ||\nabla V_j(x)|| \leq \beta_j||X_t - x|| \leq \beta t||\nabla V(x)|| + \beta\sqrt{2}||B_t||
$$

thus

$$
||\nabla V(X_t)|| \leq G(X_t) = \max_{j\in I} ||\nabla V_j(X_t)|| \leq G(x) + \beta t G(x) + \beta\sqrt{2} \sup_{t\in[0,h]} ||B_t||
$$

$$
\leq 1.1 G(x) + \beta\sqrt{2} \sup_{t\in[0,h]} ||B_t||
$$

and

$$
\begin{aligned}
\int_0^h ||\nabla V(X_t) - \nabla V(x)||^2 dt &\leq 2\int_0^h (||\nabla V(X_t)||^2 + ||\nabla V(x)||^2) dt \\
&\leq h[2(1.1 G(x))^2 + 4\beta^2 \sup_{t\in[0,h]} ||B_t|| + G(x)^2] \qquad (10) \\
&\leq 4h G^2(x) + 4\beta^2 h \sup_{t\in[0,h]} ||B_t||^2
\end{aligned}
$$

We first prove the following.

**Proposition 16.** *For any $\lambda < \frac{1}{8\beta^2 h^2}$,*

$$
\mathbb{E}_{Q_T}\left[\exp\left(\lambda\int_0^T ||b_t^P - b_t^Q||^2 dt\right)\right] \leq \exp(4\lambda h G^2(x))\left(\frac{1 + 8\lambda\beta^2 h^2}{1 - 8\lambda\beta^2 h^2}\right)^d.
$$

*Proof.* By Proposition 1, for $\lambda \leq \frac{1}{16\beta^2 h^2}$

$$
\mathbb{E}\left[\exp\left(\lambda\int_0^h ||\nabla V(X_t) - \nabla V(x)||^2 dt\right)\right] \leq \mathbb{E}\left[\exp\left(4h\lambda G^2(x) + 4\lambda\beta^2 h \sup_{t\in[0,h]} ||B_t||^2\right)\right]
$$

$$
\leq \exp(4\lambda h G^2(x))\exp(6\beta^2 h^2 d\lambda)
$$

$\qquad\square$

Apply Proposition 16 with $\lambda = 1/4$ gives

$$\mathbb{E}_{Q_T}\left[\exp\left(\frac{1}{4}\int_0^T ||b_t^P - b_t^Q||^2 dt\right)\right] = \mathbb{E}\left[\exp\left(\frac{1}{4}\int_0^h ||\nabla V(X_t) - \nabla V(x)||^2 dt\right)\right]$$

$$\leq \exp\left(hG^2(x)\right)\exp(1.5\beta^2 h^2 d\lambda) < \infty$$

Next, let

$$H_t = \int_0^t \frac{1}{\sqrt{2}}\langle b_s^P - b_s^Q, dB_s^Q\rangle - \frac{1}{4}\int_0^t ||b_s^P - b_s^Q||^2 ds$$

then $\frac{dP_t}{dQ_t} = \exp(H_t)$ and

$$dH_t = -\frac{1}{4}||\nabla V(X_t) - \nabla V(x)||^2 dt + \frac{1}{\sqrt{2}}\langle -\nabla V(X_t) + \nabla V(x), dB_t^Q\rangle$$

By Ito's formula,

$$d\exp(qH_t)$$
$$= \frac{q^2 - q}{4}\exp(qH_t)||\nabla V(X_t) - \nabla V(x)||^2 + q\exp(qH_t)\frac{1}{\sqrt{2}}\langle \nabla V(x) - \nabla V(X_t), dB_t^Q\rangle$$

Thus

$$\mathbb{E}_{Q_T}[\exp(qH_T)] - 1 = \frac{q^2 - q}{4}\mathbb{E}\left[\int_0^h \exp(qH_t)||\nabla V(X_t) - \nabla V(x)||^2 dt\right]$$

$$\leq \frac{q^2}{4}\int_0^h \sqrt{\mathbb{E}[\exp(2qH_t)]}\cdot\sqrt{\mathbb{E}[||\nabla V(X_t) - \nabla V(x)||^4]}dt$$

We bound each term under the square root.

$$\mathbb{E}[||\nabla V(X_t) - \nabla V(x)||^4] \leq \mathbb{E}[(1.1G(x) + \beta\sqrt{2}\sup_{t\in[0,h]}||B_t|| + G(x))^4]$$

$$\leq 40G^4(x) + 32\beta^4\mathbb{E}[\sup_{t\in[0,h]}||B_t||^4]$$

$$\leq 40G^4(x) + O(\beta^4 d^2 h^2)$$

By Lemma 4 and Proposition 16, if $q^2 < \frac{1}{100\beta^2 h^2}$ then

$$(\mathbb{E}[\exp(2qH_t)])^2 \leq \mathbb{E}\left[\exp\left(4q^2\int_0^h ||\nabla V(X_t) - \nabla V(x)||^2 dt\right)\right]$$

$$\leq \exp(16q^2 hG^2(x))\exp(24q^2\beta^2 h^2)$$

$$\leq \exp(16q^2 hG^2(x) + 72q^2\beta^2 h^2 d)$$

Substitute back in gives

$$\mathbb{E}_{Q_T}[\exp(qH_T)] - 1 \leq \frac{q^2 h}{4}(7G^2(x) + O(\beta^2 dh))\exp(4q^2 hG^2(x) + 18q^2\beta^2 h^2 d)$$

By the data processing inequality

$$\mathcal{R}_q(\bar{\nu}_h||\nu_h) \leq \mathcal{R}_q(P_T||Q_T) = \frac{\ln\mathbb{E}_{Q_T}[\exp(qH_T)]}{q - 1}$$

$$\leq \ln\left(1 + \frac{q^2 h}{4}(7G^2(x) + 6C\beta^2 dh)\exp(4q^2 hG^2(x) + 18q^2\beta^2 h^2 d)\right)$$

$$\leq \ln\left[\left(1 + \frac{q^2 h}{4}(7G^2(x) + 6C\beta^2 dh\right)\exp(4q^2 hG^2(x) + 18q^2\beta^2 h^2 d)\right]$$

$$\leq \ln\left(1 + \frac{q^2 h}{4}(7G^2(x) + 6C\beta^2 dh)) + (4q^2 hG^2(x) + 18q^2\beta^2 h^2 d\right)$$

$$\leq 6q^2 h(G^2(x) + (3 + C/2)\beta^2 dh)$$

Now, note that $\nu_h = \mathcal{N}(y, \sigma^2 I)$ with $y = x - h\nabla V(x)$ and $\sigma^2 = 2h$. Note that $||\nabla V_i(y)|| \leq ||\nabla V_i(x)|| + \beta||y - x|| \leq ||\nabla V_i(x)|| + \beta h||\nabla V(x)|| \leq 1.1 G(x)$. By Lemma 2 and Proposition 15

$$\mathcal{R}_{2q-1}(\nu_h||\mu) \leq \max_i \mathcal{R}_{2q-1}(\nu_h||\mu_i) \leq \frac{d}{2}\ln((2\alpha h)^{-1}) + \alpha^{-1}\max_i ||\nabla V_i(y)||^2$$

$$\leq \frac{d}{2}\ln((2\alpha h)^{-1}) + 2\alpha^{-1}G^2(x)$$

The final statement follows from the weak triangle inequality (Lemma 1). $\qquad\square$

## E  PERTURBATION ANALYSIS

In this section, we bound the drift $||\bar{X}_t - \bar{X}_{kh}||$ for $t \in [kh, (k+1)h]$ of the continuous Langevin diffusion $\bar{X}_t$. These bounds will be used to bound the mixing time of the continuous Langevin diffusion and to compare the discrete LMC with the continuous process via Girsanov's theorem.

We will consider subset $S$ of $I$ such that the components $\mu_i$ for $i \in S$ have modes that are close together. We record the properties of the mixture distribution $\mu_S$ (see Definition 2 for definition) and and its log density function $V_S = -\log\mu_S$ in Assumption 2. To be clear, we are defining this assumption as it is shared between multiple lemmas (and will be satisfied when we apply the lemmas), it is not a new assumption for the final result.

**Assumption 2** (Cluster assumption). *We say a subset $S$ of $I$ satisfies the cluster assumption if there exists $u_S \in \mathbb{R}^d$, $A_{Hess,1}, A_{Hess,0}, A_{grad,1}, A_{grad,0}$ s.t.*

1. $||\nabla^2 V_S(x)||_{OP} \leq \min_{i \in S} A_{Hess,1}||x - u_i||^2 + A_{Hess,0}$

2. $||\nabla V_S(x)|| \leq A_{grad,1}||x - u_S|| + A_{grad,0}$.

**Proposition 17.** *Suppose for all $i \in S$, $\mu_i$ satisfies item 1 of Lemma 5. Let $u_i$ and $D$ be as in Lemma 5 and suppose $||u_i - u_j|| \leq L$ for $i, j \in S$ with $L \geq 10D$. Then $\mu_S$ satisfies Assumption 2 with $u_S = p_S^{-1}\sum_{i \in S} p_i u_i$, $A_{grad,1} = \beta$, $A_{grad,0} = \beta L$, $A_{Hess,1} = 2\beta^2$, $A_{Hess,0} = 2\beta^2 L^2$.*

*In addition, if $\mu_i$ satisfies item 3 of Lemma 5 then*
$$\mathbb{P}_{\mu_S}[||x - u_S|| \geq 1.1L + t] \leq \exp(-\alpha t^2/4).$$

*Proof.* First, $\forall i \in S : ||u_i - u_S|| = p_S^{-1}\sum_{j \in S} p_j||u_i - u_j|| \leq L$. By Proposition 6

$$p_S \nabla V_S(x) = \sum_{i \in S} p_i \nabla V_i(x) \leq \sum_{i \in S} p_i \beta ||x - u_i||$$

$$\leq \sum_{i \in S} p_i \beta(||x - u_S|| + ||u_i - u_S||) \leq p_S(\beta||x - u_S|| + L)$$

We replace $I$ with $S$ and use the formula from Proposition 7. By Holder's inequality
$$||\nabla V_i(x) - \nabla V_j(x)||^2 \leq 4\max_{k \in S} ||\nabla V_k(x)||^2 \leq 4\beta^2 \max_{k \in S} ||x - u_k||^2 \leq 8\beta^2 \min_{k \in S}(||x - u_k||^2 + L^2)$$

Next, for $\tilde{p}_i = p_i/p_S$, we have

$$\sum_{i,j \in S} \tilde{p}_i \tilde{p}_j \mu_i(x)\mu_j(x) = \left(\sum_{i \in S} \tilde{p}_i \mu_i(x)\right)^2 = \mu_C^2(x)$$

thus

$$\beta I \succeq \nabla^2 V_C(x) \succeq 0 - I\max_{i,j \in S} ||\nabla V_i(x) - \nabla V_j(x)||^2/4 \succeq -2I\beta^2 \min_{k \in S}(||x - u_k||^2 + L^2).$$

For $\tilde{D} = D + L \leq 1.1L$ and $\gamma = \frac{2}{\alpha}$.

$$\mathbb{P}_{\mu_S}[||\bar{Z} - u_S|| \geq \tilde{D} + \sqrt{\gamma\ln(1/\eta)}] = p_S^{-1}\sum_{i \in S} p_i \mu_i(\bar{Z} : ||\bar{Z} - u_S|| \geq \tilde{D} + \sqrt{\gamma\ln(1/\eta)})$$

$$\leq p_S^{-1}\sum_{i \in S} p_i \mu_i(\bar{Z} : ||\bar{Z} - u_i|| \geq D + \sqrt{\gamma\ln(1/\eta)})$$

$$\leq p_S^{-1}\sum_{i \in S} p_i \eta = \eta$$

where first inequality is due to $||u_i - u_S|| \leq L$ for all $i \in S$.

$\square$

**Proposition 18.** *Suppose $S \subseteq I$ satisfies item 1 and item 2 of Assumption 2. Let $(\bar{Z}_t)_{t \geq 0}$ be the continuous Langevin diffusion with score $\nabla V_S$ initialized at $\bar{Z}_0 \sim \nu_0$ then for $t \in [kh, (k+1)h)$*

$$\mathbb{E}[||\nabla V(\bar{Z}_{kh}) - \nabla V(\bar{Z}_t)||^2]$$
$$\lesssim \sqrt{\mathbb{E}[A_{Hess,1}^4(||\bar{Z}_{kh} - u_S||^8 + ||\bar{Z}_t - u_S||^8) + A_{Hess,0}^4]}$$
$$\times \sqrt{(t - kh)^3 \int_{kh}^t (A_{grad,1}^4 \mathbb{E}[||\bar{Z}_s - u_S||^4] + A_{grad,0}^4)ds + d^2(t - kh)^2}$$

*Proof.* By the mean value inequality

$$||\nabla V_S(\bar{Z}_{kh}) - \nabla V_S(\bar{Z}_t)||^2 \leq ||\bar{Z}_{kh} - \bar{Z}_t|| \max_{y = \eta \bar{Z}_{kh} + (1-\eta)\bar{Z}_t, \eta \in [0,1]} ||\nabla^2 V_S(y)||$$

By item 1 of Assumption 2, the fact that $y = \eta \bar{Z}_{kh} + (1 - \eta)\bar{Z}_t$ and Holder's inequality

$$||\nabla^2 V_S(y)||_{OP} \leq A_{\text{Hess},1}||y - u_S||^2 + A_{\text{Hess},0} \leq A_{\text{Hess},1}(||\bar{Z}_{kh} - u_S||^2 + ||\bar{Z}_t - u_S||^2) + A_{\text{Hess},0}$$

and so

$$\mathbb{E}[||\nabla V_S(\bar{Z}_{kh}) - \nabla V_S(\bar{Z}_t)||^2]$$
$$\leq \mathbb{E}\left[ \left(A_{\text{Hess},1}(||\bar{Z}_{kh} - u_S||^2 + ||\bar{Z}_t - u_S||^2) + A_{\text{Hess},0}\right)^2 \cdot || - \int_{kh}^t \nabla V_S(\bar{Z}_s)ds + \sqrt{2}B_{t-kh}||^2 \right]$$
$$\leq \sqrt{\mathbb{E}\left(A_{\text{Hess},1}(||\bar{X}_{kh} - u_S||^2 + ||\bar{X}_t - u_S||^2) + A_{\text{Hess},0}\right)^4}$$
$$\cdot \sqrt{\mathbb{E}|| - \int_{kh}^t \nabla V_S(\bar{Z}_s)ds + \sqrt{2}B_{t-kh}||^4}.$$

By item 2 of Assumption 2 and Holder's inequality, for $p = O(1)$

$$\mathbb{E}[|| - \int_{kh}^t \nabla V_S(\bar{Z}_s)ds + \sqrt{2}B_{t-kh}||^{2p}]$$
$$\lesssim \mathbb{E}[(t - kh)^{2p-1} \int_{kh}^t ||\nabla V_S(\bar{Z}_s)||^{2p}ds] + \mathbb{E}[||B_{t-kh}||^{2p}]$$
$$\lesssim (t - kh)^{2p-1} \int_{kh}^t (A_{\text{grad},1}^{2p}||\bar{Z}_s - u_S||^{2p} + A_{\text{grad},0}^{2p})ds + (d(t - kh))^p$$

The desired result follows from $p = 4$.

$\square$

**Proposition 19.** *Suppose $S \subseteq I$ satisfies item 2 of Assumption 2. Let $(\bar{Z}_t)_{t \geq 0}$ be the continuous Langevin diffusion wrt $\mu_S$ initialized at $\nu_0$. Suppose $h \leq \frac{1}{2A_{grad,1}}$ and $\sup_{k \in [0,N-1] \cap \mathbb{N}} ||\bar{Z}_{kh} - u_S|| \leq D$ then*

$$\sup_{k \in [0,N-1] \cap \mathbb{N}, t \in [0,h]} ||\bar{Z}_{kh+t} - \bar{Z}_{kh}|| \leq 2h(A_{grad,0} + A_{grad,1}||\bar{Z}_{kh} - u_S||) + \sqrt{48dh \ln \frac{6N}{\eta}}$$

*thus with probability $\geq 1 - \eta$*

$$\sup_{k \in [0,N-1] \cap \mathbb{N}, t \in [0,h]} ||\bar{Z}_{kh+t} - u_S|| \leq 2hA_{grad,0} + 2D + \sqrt{48dh \ln \frac{6N}{\eta}}$$

*Proof.* The proof is identical to (Chewi et al., 2021, Lemma 24). By triangle inequality,

$$||\bar{Z}_{kh+t} - \bar{Z}_{kh}||$$

$$\leq \int_0^t ||\nabla V_S(\bar{Z}_{kh+r})|| dr + \sqrt{2}||B_{kh+t} - B_{kh}||$$

$$\leq h A_{\text{grad},0} + A_{\text{grad},1} \int_0^t ||\bar{Z}_{kh+r} - u_S|| dr + \sqrt{2}||B_{kh+t} - B_{kh}||$$

$$\leq h A_{\text{grad},0} + A_{\text{grad},1} \left( h||\bar{Z}_{kh} - u_S|| + \int_0^t ||\bar{Z}_{kh+r} - \bar{Z}_{kh}|| dr \right) + \sqrt{2}||B_{kh+t} - B_{kh}||$$

where we use item 2 of Assumption 2 in the second inequality. Gronwall's inequality then implies

$$||\bar{Z}_{kh+t} - \bar{Z}_{kh}||$$

$$\leq \left( h(A_{\text{grad},0} + A_{\text{grad},1}||\bar{Z}_{kh} - u_S||) + \sqrt{2} \sup_{t \in [0,h]} ||B_{kh+t} - B_{kh}|| \right) \exp(h A_{\text{grad},1})$$

$$\leq 2h(A_{\text{grad},0} + A_{\text{grad},1}||\bar{Z}_{kh} - u_S||) + \sqrt{8} \sup_{t \in [0,h]} ||B_{kh+t} - B_{kh}||$$

as long as $h \leq \frac{1}{2A_{\text{grad},1}}$.

Thus by triangle inequality,

$$||\bar{Z}_{kh+t} - u_S|| \leq ||\bar{Z}_{kh} - u_S|| + ||\bar{Z}_{kh+t} - \bar{Z}_{kh}||$$

$$\leq 2h A_{\text{grad},0} + ||\bar{Z}_{kh} - u_S||(2h A_{\text{grad},1} + 1) + \sqrt{8} \sup_{t \in [0,h]} ||B_{kh+t} - B_{kh}||$$

By union bounds and concentration for Brownian motion (see (Chewi et al., 2021, Lemma 32)), with probability $1 - \eta$,

$$\sup_{k \in [0,N-1] \cap \mathbb{N}, t \in [0,h]} ||B_{kh+t} - B_{kh}|| \leq \sqrt{6dh \ln \frac{6N}{\eta}}$$

thus

$$\sup_{k \in [0,N-1] \cap \mathbb{N}, t \in [0,h]} ||\bar{Z}_{kh+t} - u_S|| \leq 2h A_{\text{grad},0} + 2D + \sqrt{48dh \ln \frac{6N}{\eta}}$$

$\square$

## F    ANALYSIS OF CONTINUOUS-TIME DIFFUSION

In this section, we analyze an idealized version of the final LMC chain: we assume knowledge of the exact score function and run the continuous time Langevin diffusion. First in Lemma 11 below, we prove that when the diffusion is initialized from a point, it converges in a certain amount of time to a sample from a mixture distribution corresponding to the clusters near the initialization. Then in Theorem 5 we deduce the analogue of our main result for the idealized process: the diffusion started from samples converges to the true distribution.

**Definition 4.** *For $S \subseteq I$ and $x \in \mathbb{R}^d$, let $i_{\max,S}(x) = \arg\max_{i \in S} \mu_i(x)$. We break ties in lexicographic order of $i$ i.e. we let $i_{\max,S}(x)$ be the maximum index among all indices $i$ s.t. $\mu_i(x) = \max_{j \in S} \mu_j(x)$.*

**Lemma 11.** *Fix $\epsilon_{TV}, \tau \in (0, 1/2), \delta \in (0, 1]$. Fix $S \subseteq I$. Let $\bar{p}_i = p_i p_S^{-1}$ and recall that $\mu_S = \sum_{i \in S} \bar{p}_i \mu_i$. Let $p_* = \min_{i \in S} \bar{p}_i$. Note that $p_* \geq \min_{i \in I} p_i$. Recall that $|I| = K$.*

*Suppose for $i \in S$, $\mu_i$ are $\alpha$-strongly log-concave and $\beta$-smooth with $\beta \geq 1$. Let $u_i = \arg\min_x V_i(x)$ and $D \geq 5\sqrt{\frac{d}{\alpha}}$ be as defined in Lemma 5. Suppose there exists $L \geq 10D$ such that for any $i, j \in S$, $||u_i - u_j|| \leq L$.*

*Let $\mathbb{G}^\delta := \mathbb{G}^\delta(S, E)$ be the graph on $S$ with an edge between $i, j$ iff $\delta_{ij} \leq \delta$. Let*

$$T = \frac{2C_{p_*, K}}{\delta \alpha} \left( \ln \left( \frac{\beta^2 L}{\alpha} \right) + \ln \ln \tau^{-1} + 2 \ln \tilde{\epsilon}_{TV}^{-1} \right).$$

*Suppose for all $i, j \in S$ which are not in the same connected component of $\mathbb{G}^\delta$, $\delta_{ij} \leq \delta'$ with*

$$\delta' = \frac{\delta^{3/2}\alpha^{3/2}p_*^{5/2}\epsilon_{TV}^2\tau}{10^5 K^5 d(\beta L)^3 \ln^{3/2}(p_*^{-1})\ln^{3/2}\frac{\beta^2 L \epsilon_{TV}^{-1}\ln\tau^{-1}}{\alpha}\ln^{2.51}\frac{16 d(\beta L)^2}{\epsilon_{TV}\tau\delta\alpha}}$$

*For $x \in \mathbb{R}^d$, let $(\bar{X}_t^{\delta_x})_{t\geq 0}$ denote the continuous Langevin diffusion with score $\nabla V_S$ initialized at $\delta_x$, and let $C_{\max}(x)$ be the unique connected component of $\mathbb{G}^\delta$ containing $i_{\max,S}(x) = \arg\max_{i \in S}\mu_i(x)$ as defined in Definition 4. Then*

$$\mathbb{P}_{x\sim\mu_S}[d_{TV}(\mathcal{L}(\bar{X}_T^{\delta_x}|x), \mu_{C_{\max}(x)}) \leq \epsilon_{TV}] \geq 1 - \tau$$

From the above lemma, we can deduce the following theorem. (The proof of the lemma is deferred until after the proof of the theorem.) In this result, the reader can consider simply the case $S = I$; the flexibility to pick a subset of indices is allowed for convenience later.

**Theorem 5.** *Fix $\epsilon_{TV}, \tau \in (0, 1/2)$. Fix $S \subseteq I$. Let $\bar{p}_i = p_i p_S^{-1}$ and recall that $\mu_S = \sum_{i\in S}\bar{p}_i\mu_i$. Let $p_* = \min_{i\in S}\bar{p}_i$. Note that $p_* \geq \min_{i\in I}p_i$. Recall that $|I| = K$.*

*Suppose for $i \in S$, $\mu_i$ are $\alpha$-strongly log-concave and $\beta$-smooth with $\beta \geq 1$. Let $u_i = \arg\min_x V_i(x)$ and $D \geq 5\sqrt{\frac{d}{\alpha}}$ be as defined in Lemma 5. Suppose there exists $L \geq 10D$ such that for any $i, j \in S$, $\|u_i - u_j\| \leq L$. Let $U_{sample}$ be a set of $M$ i.i.d. samples from $\mu_S$ and $\nu_{sample}$ be the uniform distribution over $U_{sample}$. Let $(\bar{X}_t^{\nu_{sample}})_{t\geq 0}$ be the continuous Langevin diffusion with score $\nabla V_S$ initialized at $\nu_{sample}$. Let*

$$\tilde{\Gamma} = \frac{p_*^{7/2}\epsilon_{TV}^3\alpha^{3/2}}{10^8 d(\beta L)^3\exp(K)\ln^{3/2}(p_*^{-1})\ln^5\frac{16 d(\beta L)^2}{\epsilon_{TV}\tau\alpha}},$$

*If $M \geq 600(\epsilon_{TV}^2 p_*)^{-1}K^2\log(K\tau^{-1})$ and*

$$T \geq \Theta\left(\alpha^{-1}K^2 p_*^{-1}\ln(10 p_*^{-1})\tilde{\Gamma}^{-2((3/2)^{K-1}-1)}\right)$$

*then*

$$\mathbb{P}_{U_{sample}}[d_{TV}(\mathcal{L}(\bar{X}_T^{\nu_{sample}}|U_{sample}), \mu_S) \leq \epsilon_{TV}] \geq 1 - \tau$$

**Remark 6.** *Note that after fixing $U_{sample}$, $\hat{\mu}_S^{U_{sample}} := \mathcal{L}(\bar{X}_T^{\nu_{sample}}|U_{sample})$ is a function of $U_{sample}$ and Brownian motions $(B_t)_{t\in[0,T]}$. Each run of the Langevin diffusion produces a sample from $\hat{\mu}_S^{U_{sample}}$ by choosing/sampling a value for the Brownian motions, thus we can produce as many samples as desired from $\hat{\mu}_S^{U_{sample}}$, while Theorem 5 guarantees that $\hat{\mu}_S^{U_{sample}}$ is approximately close to $\mu_S$ in total variation distance for a typical set of samples $U_{sample}$.*

*Proof of Theorem 5.* Let $\bar{p}_i = p_i p_S^{-1}$ then $\mu_S = \sum_{i\in S}\bar{p}_i\mu_i$. For $C \subseteq S$, let $\bar{p}_C = \sum_{i\in C}\bar{p}_i$.

Let $\tilde{\epsilon}_{TV} = \frac{\epsilon_{TV}}{9K}$ and $\tilde{\tau} = \frac{p_*\epsilon_{TV}}{9K} \leq \min\{\frac{\epsilon_{TV}}{9K^2}, p_*/3\}$. Define the sequence $1 = \delta_0 > \delta_1 > \cdots > \delta_K$ inductively as follow:

$$\delta_{s+1} = \frac{\delta_s^{3/2}\alpha^{3/2}p_*^{5/2}\tilde{\epsilon}_{TV}^2\tilde{\tau}}{10^5 K^5 d(\beta L)^3\ln^{3/2}(p_*^{-1})\ln^{3/2}\frac{\beta^2 L\tilde{\epsilon}_{TV}^{-1}\ln\tilde{\tau}^{-1}}{\alpha}\ln^{2.51}\frac{16 d(\beta L)^2}{\tilde{\epsilon}_{TV}\tilde{\tau}\delta_s\alpha}}$$

$$\geq \frac{\delta_s^{3/2}\alpha^{3/2}p_*^{7/2}\epsilon_{TV}^3}{10^8 K^8 d(\beta L)^3\ln^{3/2}(p_*^{-1})\ln^{3/2}\frac{\beta^2 L\epsilon_{TV}^{-1}K}{\alpha}\ln^{2.51}\frac{16 d(\beta L)^2 K}{\epsilon_{TV}\delta_s\alpha}}$$

Let $\mathbb{G}^s := \mathbb{G}^\delta(S, E)$ be the graph on $S$ with an edge between $i, j$ iff $\delta_{ij} \geq \delta_s$. Fix one such $s$ s.t. $s \leq K - 2$. Suppose $\delta_{ij} \leq \delta_{s+1}$ for all $i, j$ not in the same connected component of $\mathbb{G}^s$, y then Lemma 11 applies. Let the connected components of $\mathbb{G}^s$ be $C_1^s, \ldots, C_m^s$. For $x \in \mathbb{R}^d$, let $C_{\max}^s(x)$ be the unique connected component of $\mathbb{G}^s$ containing $i_{\max,S}(x)$ and let $(\bar{X}_t^{\delta_x})_{t\geq 0}$ denote the continuous Langevin diffusion with score $\nabla V$ initialized at $\delta_x$, then for $T_s = \frac{2C_{p_*,K}}{\delta_s\alpha}(\ln\frac{\beta^2 L}{\alpha} + \ln\ln\tilde{\tau}^{-1} + 2\ln\tilde{\epsilon}_{TV}^{-1})$,

$$\mathbb{P}_{x\sim\mu_S}[d_{TV}(\bar{X}_{T_s}^{\delta_x}, \mu_{C_{\max}^s(x)}) \leq \tilde{\epsilon}_{TV}] \geq 1 - \tilde{\tau}$$

and by Proposition 21,

$$\mathbb{P}_{x \sim \mu_S}[i_{\max,S}(x) \in C_r^s] \geq (1 - \delta_{s+1})\bar{p}_{C_r^s}.$$

It is easy to see that $\delta_{s+1} \leq \tilde{\epsilon}_{TV}/3$. By Proposition 23, as long as $M \geq 600(\epsilon_{TV}^2 p_*)^{-1} K^2(\log K + \log \tau^{-1})$

$$\mathbb{P}_{U_{\text{sample}}}[d_{TV}(\mathcal{L}(\bar{X}_{T_s}^{\nu_{\text{sample}}}|U_{\text{sample}}), \mu_S) \leq \epsilon_{TV}] \geq 1 - \tau$$

Since $\mu_S$ is the stationary distribution of the continuous Langevin with score function $\nabla V_S$, for any $T \geq T_{K-1} \geq T_s$, $d_{TV}(\mathcal{L}(\bar{X}_T^{\nu_{\text{sample}}}|U_{\text{sample}}), \mu_S) \leq d_{TV}(\mathcal{L}(\bar{X}_{T_s}^{\nu_{\text{sample}}}|U_{\text{sample}}), \mu_S)$ thus

$$\mathbb{P}_{U_{\text{sample}}}[d_{TV}(\mathcal{L}(\bar{X}_{T_s}^{\nu_{\text{sample}}}|U_{\text{sample}}), \mu_S) \leq \epsilon_{TV}] \geq 1 - \tau.$$

On the other hand, suppose for all $s \in [0, K-2] \cap \mathbb{N}$, there exists $i, j$ not in the same connected component of $\mathbb{G}^s$ s.t. $\delta_{ij} > \delta_{s+1}$, then $\mathbb{G}^{s+1}$ has one fewer connected components than $\mathbb{G}^s$. Thus $\mathbb{G}^{K-1}$ is connected then $\mu$ has LSI constant $\propto \delta_{K-1}^{-1}$, thus Lemma 11 apply with $\delta = \delta_{K-1}$ and Proposition 21 apply with $\delta' = 0$. For $T \geq T_{K-1}$,

$$\mathbb{P}_{U_{\text{sample}}}[d_{TV}(\mathcal{L}(\bar{X}_T^{\nu_{\text{sample}}}|U_{\text{sample}}), \mu_S) \leq \epsilon_{TV}] \geq 1 - \tau.$$

Let $\Gamma = \frac{p_*^{7/2} \epsilon_{TV}^3 \alpha^{3/2}}{10^8 K^8 d(\beta L)^3}$. If we ignore log terms, then $\delta_{s+1} = \delta_s^{3/2}\Gamma$ thus $\delta_s \approx \Gamma^{1+3/2+\cdots+(3/2)^{s-1}} = \Gamma^{2((3/2)^s-1)}$. To get the correct bound for $\delta_s$ and $T_s$, we can let

$$\Gamma_1 = \frac{p_*^{7/2} \epsilon_{TV}^3 \alpha^{3/2}}{8000 d(\beta L)^3 \exp(K) \ln^{3/2}(p_*^{-1}) \ln^{4.5} \frac{16 d(\beta L)^2}{\epsilon_{TV}\tau\alpha}} \leq \Gamma$$

then we can inductively prove $\delta_s \geq \Gamma_1^{2((3/2)^s-1)}$ and thus get the bound on $T_s$ i.e.

$$T_s \leq \Theta(\alpha^{-1} K^2 p_*^{-1} \ln(10 p_*^{-1}) \ln(\frac{\beta^2 L \epsilon_{TV}}{\alpha p_* K}) \Gamma_1^{-2((3/2)^s-1)})$$

$$= \Theta(\alpha^{-1} K^2 p_*^{-1} \ln(10 p_*^{-1}) \tilde{\Gamma}^{-2((3/2)^s-1)})$$

with $\tilde{\Gamma} = \frac{p_*^{7/2} \epsilon_{TV}^3 \alpha^{3/2}}{10^8 d(\beta L)^3 \exp(K) \ln^{3/2}(p_*^{-1}) \ln^5 \frac{16 d(\beta L)^2}{\epsilon_{TV}\tau\alpha}}$. $\qquad \square$

*Proof of Lemma 11.* Let $(\bar{X}_t)$ denote the continuous Langevin with score $\nabla V_S$ initialized at $\mu_S$. Since $\mu_S$ is the stationary distribution of continuous the Langevin with score $\nabla V_S$, the law $\mathcal{L}(\bar{X}_t)$ of $\bar{X}_t$ is $\mu_S$ at all time $t$. Let $\eta = \tau\epsilon_{TV}/2$ and $N = T/h$. Let $h > 0, \gamma \in (0, 1)$ to be chosen later. Let $\mathcal{C}$ be the partition of $S$ consisting of connected components of the graph $\mathbb{G}^\delta$, and $B_{S,\mathcal{C},\gamma}$ be defined as in Definition 5. Suppose $\delta', \gamma$ satisfies $K^2\gamma^{-1}\delta' \times T/h \leq \eta/2$, then by Lemma 12, $\mu_S(B_{S,\mathcal{C},\gamma})N \leq K^2\gamma^{-1}\delta' \times T/h \leq \eta/2$

Since the law of $\bar{X}_{kh}$ is $\mu_S$, we can bound $||\bar{X}_{kh} - u_S||$ using sub-Gaussian concentration of $\mu_S$ (due to Proposition 17). By the union bound, with probability $1 - \eta$, the event $\mathcal{E}_{\text{discrete}}$ happens where $\mathcal{E}_{\text{discrete}}$ is defined by: $\forall k \in [0, N-1] \cap \mathbb{N} : ||\bar{X}_{kh} - u_S|| \leq 2L + \sqrt{\frac{64}{\alpha} \ln \frac{16N}{\eta}}$ and $\bar{X}_{kh} \notin B_{S,\mathcal{C},\gamma}$.

Since $\mu_S = \mathbb{E}_{x \sim \mu_S}[\delta_x]$ and $\mu_S$ and $\delta_x$ are the initial distribution of $\bar{X}_t$ and $\bar{X}_t^{\delta_x}$ respectively, so $\mathcal{L}(\bar{X}_{kh}) = \mathbb{E}_{x \sim \mu_S}[\mathcal{L}(\bar{X}_{kh}^{\delta_x}|x)]$ where $\mathcal{L}(X)$ denote the law of the random variable $X$. Thus, let $\tilde{L} := 2L + \sqrt{\frac{64}{\alpha} \ln \frac{16N}{\eta}}$ and $\mathcal{G}_x$ is the event

$$\mathbb{P}_{\mathcal{F}_t}[\forall k \in [0, N-1] \cap \mathbb{N} : ||\bar{X}_{kh}^{\delta_x} - u_S|| \leq \tilde{L} \wedge \bar{X}_{kh}^{\delta_x} \notin B_{S,\mathcal{C},\gamma}] \geq 1 - \epsilon_{TV}/10$$

where the probability is taken over the randomness of the Brownian motions, then $\mathbb{P}_{x \sim \mu}[\mathcal{G}_x] \geq 1 - \tau/2$

Fix $x$, let $C = C_{\max}(x)$ and suppose $\mathcal{G}_x$ holds. Suppose $h$ satisfies the precondition of Proposition 26, then with probability $\geq 1 - \epsilon_{TV}/5$,

$$\sup_{t \in [0,T]} ||\nabla V_S(\bar{X}_t^{\delta_x}) - \nabla V_{C_{\max}(x)}(\bar{X}_t^{\delta_x})|| \leq \epsilon_{\text{score},1} := 36 p_*^{-1}\gamma\beta\tilde{L}$$

thus $\bar{X}_t^{\delta_x} \notin B$ for all $t \in [0, T]$, where $B$ is the "bad" set defined by $B = \{z \in \mathbb{R}^d : ||\nabla V_S(z) - \nabla V_C(z)|| > \epsilon_{\text{score},1}\}$. Let $\nu_0$ be the distribution of $\bar{X}_{h'}^{\delta_x}$ for some $h' \leq 1/(2\beta)$. Let $\mathcal{G}_{\text{init},x}$ be the event

that $||x - u_S|| \le L_1 := 2L + \log(10/\tau)$ then $\mathbb{P}_{x \sim \mu}[\mathcal{G}_{\text{init},x}] \ge 1 - \tau/10$. Suppose $\mathcal{G}_{\text{init},x}$ happens. Then $G_S(x) = \max_{i \in S} ||\nabla V_i(x)|| \lesssim \beta L_1$. Set $h' = \min\{\frac{1}{\beta \tilde{L}}, \frac{1}{\beta d}\}$ then by Lemma 10,

$$\mathcal{D}_{\text{KL}}(\nu_0 || \mu_C) \lesssim d \ln L_1 + \alpha^{-1} \beta^2 L_1^2$$

Pick $T = \frac{2C_{p_*,K}}{\delta \alpha}(\ln \frac{\beta^2 L}{\alpha} + \ln \ln \tau^{-1})$ then $T - h' \ge T_{\text{process}} := \frac{C_{p_*,K}}{\delta \alpha}(\ln \mathcal{D}_{\text{KL}}(\nu_0 || \mu_S) + 2 \ln \epsilon_{TV}^{-1})$

Let $(\bar{Z}_t^{\nu_0})_{t \ge 0}$ be the continuous Langevin initialized at $\nu_0$ with score $s_\infty$ defined by

$$s_\infty(z) = \begin{cases} \nabla V_S(z) \text{ if } x \notin B \\ \nabla V_C(z) \text{ if } x \in B \end{cases}$$

then $\sup_{z \in \mathbb{R}^d} ||s_\infty(z) - \nabla V_C(z)||^2 \le \epsilon_{\text{score},1}^2$. Note that if $\mathcal{G}_x$ holds then $\bar{X}_{t+h'}^{\delta_x} \notin B \forall t \in [0, T - h']$ and $\bar{Z}_t^{\nu_0} = \bar{X}_{t+h'}^{\delta_x} \forall t \in [0, T - h']$ thus

$$d_{TV}(\bar{X}_t^{\delta_x}, \bar{Z}_{T-h'}^{\nu_0}) \le \epsilon_{TV}/5$$

Proposition 20 gives

$$d_{TV}(\mathcal{L}(\bar{Z}_{T-h'}^{\nu_0}|x), \mu_C) \le \epsilon_{\text{score},1}\sqrt{T/2} + \epsilon_{TV}/5$$

Set $\gamma = \frac{p_* \epsilon_{TV}}{18 \beta \tilde{L} \sqrt{T}}$ then $\epsilon_{\text{score},1} = 18 p_*^{-1} \gamma \beta \tilde{L} \le \frac{\epsilon_{TV}}{\sqrt{T}}$ then by triangle inequality

$$d_{TV}(\mathcal{L}(\bar{X}_t^{\delta_x}|x), \mu_C) \le \epsilon_{TV}.$$

This holds conditioned on $\mathcal{G}_x$ and $\mathcal{G}_{\text{init},x}$ both happen, thus by union bound

$$\mathbb{P}_{x \sim \mu}[d_{TV}(\mathcal{L}(\bar{X}_t^{\delta_x}|x), \mu_{C_{\max}(x)}) \le \epsilon_{TV}] \ge 1 - \tau$$

Plug in $T, \gamma$ and set

$$h = \frac{1}{2000 d (\beta L)^2 \ln^2 \frac{16 d (\beta L)^2 T}{\epsilon_{TV} \tau}}$$

then $h \ln(1/h) \le \frac{1}{2000 d (\beta L)^2}$ and $h \ln^2(1/h) = \frac{1}{1000(\beta^2/\alpha)}$ and $h \le \frac{1}{100(\beta^2/\alpha) \ln^2(16T/\eta)}$. Hence $h$ satisfies the precondition of Proposition 26.

Finally, since $\tilde{L} \le L\sqrt{\ln \frac{16T}{h\eta}} \le 2L\sqrt{\ln(\beta L \epsilon_{TV}^{-1} \tau^{-1} T)}$, thus with

$$\delta' \le \frac{\delta^{3/2} \alpha^{3/2} p_*^{5/2} \epsilon_{TV}^2 \tau \ln^{3/2} \frac{\beta^2 L \epsilon_{TV}^{-1} \ln \tau^{-1}}{\alpha}}{10^5 K^5 d (\beta L)^3 \ln(p_*^{-1}) \ln^{2.51} \frac{16 d (\beta L)^2}{\epsilon_{TV} \tau \delta \alpha}} \le \frac{p_* \epsilon_{TV}^2 \tau}{10^5 K^2 T^{3/2} d (\beta L)^3 \ln^{2.51} \frac{16 d (\beta L)^2 T}{\epsilon_{TV} \tau}}$$

the precondition

$$K^2 \delta' \gamma^{-1} \times T/h = K^2 \delta' \times \frac{18 \beta \tilde{L} \sqrt{T}}{p_* \epsilon_{TV}} \times T/h$$

$$\le \delta' \times \frac{36 K^2 \beta L T^{3/2} \sqrt{\ln(\beta L \epsilon_{TV}^{-1} \tau^{-1} T)}}{p_* \epsilon_{TV} h}$$

$$\le \eta/2$$

$$= \epsilon_{TV} \tau/4$$

holds, so we are done.

$\square$

**Proposition 20** (Continuous chain with score estimation with $L_\infty$ error bound). *Fix $C \subseteq I$. Let $(\bar{Z}_t)_{t \ge 0}$ and $\bar{X}_t$ be the continuous Langevin diffusion with score functions $\nabla V_C$ and $s$ respectively and both $(\bar{Z}_t)$ and $(\bar{X}_t)$ are initialized at $\nu_0$. Suppose $\sup_{x \in \mathbb{R}^d} ||s(x) - \nabla V_C(x)||^2 \le \epsilon_{\text{score},1}^2$ then*

$$2d_{TV}(\bar{X}_T, \bar{Z}_T)^2 \le \mathcal{D}_{\text{KL}}(\bar{X}_T || \bar{Z}_T) \le \mathbb{E}\left[\int_0^T ||s(\bar{Z}_t) - \nabla V_C(\bar{Z}_t)||^2 dt\right] \le \epsilon_{\text{score},1}^2 T$$

*Suppose $\mu_S$ has log Sobolev constant $C_{LS}$ and $T \ge C_{LS}(\log(2\mathcal{D}_{KL}(\nu_0 || \mu_S)) + 2\log \epsilon_{TV}^{-1})$*

$$d_{TV}(\mathcal{L}(\bar{X}_T), \mu_C) \le d_{TV}(\bar{X}_T, \bar{Z}_T) + d_{TV}(\mathcal{L}(\bar{Z}_T), \mu_C) \le \epsilon_{\text{score},1}\sqrt{T/2} + \epsilon_{TV}/2$$

*Proof.* Clearly, by the assumption on $s$, $\mathbb{E}[\int_0^T ||s(\bar{Z}_t) - \nabla V_C(\bar{Z}_t)||^2 dt] \leq \int_0^T \epsilon_{\text{score},1}^2 dt = \epsilon_{\text{score},1}^2 T$. The first statement thus follows from Girsanov and the approximation argument in (Chen et al., 2023, Lemma 9) and Pinsker's inequality. Next, since $\mu_C$ has LSI constant $C_{LS}$, with this choice of $T$,

$$\mathcal{D}_{KL}(\mathcal{L}(\bar{Z}_T)||\mu_C) \leq \mathcal{D}_{KL}(\nu_0||\mu_S)\exp(-\frac{T}{C_{LS}}) \leq \epsilon_{TV}^2/2$$

and the second statement follows from Pinsker's inequality and triangle inequality for TV distance. $\square$

We need these propositions to go from Lemma 11 to Theorem 5

**Proposition 21.** *Suppose $\mu = \sum_{i \in I} p_i \mu_i$. Fix a set $C \subseteq I$. If the overlap between $\mu_i, \mu_j$ for $i \in C$ and $j \notin C$ is $\leq \delta'$ for all such $i, j$ then*

$$\mu(\{x : i_{\max}(x) \in C\}) \geq p_C(1 - \delta'|I|)$$

To remove dependency on $p_*$, we will use the following modified version of Proposition 21

**Proposition 22.** *Fix $C, C_* \subseteq I$ s.t. $C \cap C_* = \emptyset$. Let $I' = I \setminus C_*$. If for $i \in C, j \in I' \setminus C$, the overlap between $\mu_i$ and $\mu_j$ is $\leq \delta'$ then for $i_{\max,I'}(x) = \arg\max_{i \in I'} \mu_i(x)$*

$$\mu_I(\{x : i_{\max,I'}(x) \in C\}) \geq p_C(1 - \delta'|I|)$$

*Proof of Propositions 21 and 22.* We first prove Proposition 21. For $i \in C, j \notin C$

$$\mu_i(\{x : \mu_i(x) \leq \mu_j(x)\}) = \int_{x : \mu_i(x) \leq \mu_j(x)} \mu_i(x)dx$$

$$= \int_{x : \mu_i(x) \leq \mu_j(x)} \min\{\mu_i(x), \mu_j(x)\}dx$$

$$\leq \int \min\{\mu_i(x), \mu_j(x)\}dx \leq \delta'$$

By union bound, for $i \in C$

$$\mu_i(\{x \mid \exists j \notin C : \mu_i(x) \leq \mu_j(x)\}) \leq \delta'|I|$$

Let $\Lambda = \{x : i_{\max}(x) \in C\}$. If $\forall j \notin C : \mu_i(x) > \mu_j(x)$ then $i_{\max}(x) \in C$. Thus $\Lambda_i := \{x : \mu_i(x) > \mu_j(x) \forall j \notin C\} \subseteq \Lambda$ and $\mu_i(\Lambda_i) = 1 - \mu_i(\{x|\exists j \notin C : \mu_i(x) \leq \mu_j(x)\}) \geq 1 - \delta'|I|$. Since $\mu(x) \geq \sum_{i \in C} p_i \mu_i(x)$

$$\mu(\{x : i_{\max}(x) \in C\}) = \int_{x \in \Lambda} \mu(x)dx \geq \int_{x \in \Lambda} \sum_{i \in C} p_i \mu_i(x)dx = \sum_{i \in C} p_i \mu_i(\Lambda)$$

$$\geq \sum_{i \in C} p_i \mu_i(\Lambda_i) \geq \sum_{i \in C} p_i(1 - \delta'|I|) = p_C(1 - \delta'|I|)$$

The proof of Proposition 22 is identical, except we will consider $i \in C, j \in I' \setminus C$ and argue that $\mu_i(x : \mu_i(x) \leq \mu_j(x)) \leq \delta'$. Then $\mu_i(x|\exists j \in I' \setminus C : \mu_i(x) \leq \mu_j(x)) \leq \delta'|I|$. For $i \in C$, $\Lambda_i = \{x|\mu_i(x) > \mu_j(x) \forall j \in I' \setminus C\}$ then $\mu_i(\Lambda_i) \geq 1 - \delta'|I|$ and $\Lambda_i \subseteq \{x : i_{\max,I'}(x) \in C\}$. Finally,

$$\mu(\{x : i_{\max,I'}(x) \in C\}) \geq \sum_{i \in C} p_i \mu_i(\Lambda_i) \geq p_C(1 - \delta'|I|).$$

$\square$

**Proposition 23.** *Consider distributions $\mu_i$ for $i \in I$. Suppose $\mu = \sum_{i \in I} p_i \mu_i$ for $p_i > 0$ and $\sum_{i \in I} p_i = 1$. Suppose we have a partition $\mathcal{C}$ of $I$ into $C_1, \ldots, C_m$. For $x \in \mathbb{R}^d$, let $C = C_{\max}(x)$ be the unique part of the partition $\mathcal{C}$ containing $i_{\max}(x) = \arg\max_{i \in I} \mu_i(x)$. Let $p_* = \min_{i \in I} p_i$. For $x \in \mathbb{R}^d$, let $(X_t^{\delta_x})_t$ be a process initialized at $\delta_x$. Suppose for any $\tilde{\epsilon}_{TV} \in (0, 1/10), \tilde{\tau} \in (0, p_*/3)$, there exists $T_{\tilde{\epsilon}_{TV}, \tilde{\tau}}$ such that the following holds:*

$$\mathbb{P}_{x \sim \mu}[d_{TV}(\mathcal{L}(X_{T_{\tilde{\epsilon}_{TV}, \tilde{\tau}}}^x|x), \mu_{C_{\max}(x)}) \leq \tilde{\epsilon}_{TV}] \geq 1 - \tilde{\tau}.$$

*In addition, there exists $\delta' \in (0, \tilde{\epsilon}_{TV})$ s.t. for $C \in \{C_1, \ldots, C_m\}$*

$$\mathbb{P}_{x \sim \mu}[C_{\max}(x) = C] \geq p_C(1 - \delta').$$

*Let $U_{sample}$ be a set of $M$ i.i.d. samples from $\mu$ and $\nu_{sample}$ be the uniform distribution over $U_{sample}$. Let $(X_t^{\nu_{sample}})_{t \geq 0}$ be the process with score estimate $s$ initialized at $\nu_{sample}$. If $M \geq 6 \times 10^2 |I|^2 \epsilon_{TV}^{-2} p_*^{-1} \log(K\tau^{-1})$, with probability $\geq 1 - \tau$ over $U_{sample}$, let $T = T_{\frac{\epsilon_{TV}}{9|I|}, \min\{\frac{\epsilon_{TV}}{9|I|^2}, p_*/3\}}$ and $\hat{\mu} = \mathcal{L}(X_T^{\nu_{sample}}|U_{sample})$, then*

$$\mathbb{P}_{U_{sample}}[d_{TV}(\mathcal{L}(X_T^{\nu_{sample}}|U_{sample}), \mu) \leq \epsilon_{TV}] \geq 1 - \tau$$

To remove the dependency on $p_* = \min_{i \in I} p_i$, we will use this modified version of Proposition 23.

**Proposition 24.** *Consider distributions $\mu_i$ for $i \in I$. Suppose $\mu = \sum_{i \in I} p_i \mu_i$ for $p_i > 0$ and $\sum_{i \in I} p_i = 1$. For $x \in \mathbb{R}^d$, let $(X_t^{\delta_x})_t$ be a process initialized at $\delta_x$. Let $I' = \{i \in I : p_i \geq \frac{\tilde{\epsilon}_{TV}}{|I|}\}$ and $C_* = C \setminus I'$. Suppose we have a partition $\mathcal{C}$ of $I'$ into $C_1, \cdots, C_r$. For $x \in \mathbb{R}^d$, let $C_{\max}(x)$ be the unique part of the partition $\mathcal{C}$ containing $i_{\max, I'}(x) = \arg\max_{i \in I'} \mu_i(x)$.*

*Suppose for any $\tilde{\epsilon}_{TV} \in (0, 1/10), \tilde{\tau} \in (0, 1)$, there exists $T_{\tilde{\epsilon}_{TV}, \tilde{\tau}}$ such that the following holds:*

$$\mathbb{P}_{x \sim \mu}[d_{TV}(\mathcal{L}(X_{T_{\tilde{\epsilon}_{TV}, \tilde{\tau}}}^{\delta_x}|x), \mu_{C_{\max}(x)}) \leq \tilde{\epsilon}_{TV}] \geq 1 - \tilde{\tau}.$$

*In addition, there exists $\delta' \in (0, \tilde{\epsilon}_{TV})$ s.t. for $C \in \{C_1, \ldots, C_m\}$*

$$\mathbb{P}_{x \sim \mu}[C_{\max, I'}(x) = C] \geq p_C(1 - \delta').$$

*Let $U_{sample}$ be a set of $M$ i.i.d. samples from $\mu$ and $\nu_{sample}$ be the uniform distribution over $U_{sample}$. Let $(X_t^{\nu_{sample}})_{t \geq 0}$ be the process with score estimate $s$ initialized at $\nu_{sample}$. If $M \geq 2 \times 10^4 |I|^3 \epsilon_{TV}^{-3} \log(|I|\tau^{-1})$, then for $\tilde{\epsilon}_{TV} = \frac{\epsilon_{TV}}{9|I|}, \tilde{\tau} = \frac{\tilde{\epsilon}_{TV}}{9|I|}$ and $T = T_{\tilde{\epsilon}_{TV}, \tilde{\tau}}$*

$$\mathbb{P}_{U_{sample}}[d_{TV}(\mathcal{L}(X_T^{\nu_{sample}}|U_{sample}), \mu) \leq \epsilon_{TV}] \geq 1 - \tau$$

*Proof of Proposition 23 and Proposition 24.* We will prove Proposition 24. The proof of Proposition 23 is similar.

For $r \in I'$, let $\Omega_r = \{x : C_{\max}(x) = C_r \wedge d_{TV}(X_T^{\delta_x}, \mu_{C_{\max}(x)}) \leq \epsilon_{TV}\}$. Clearly, $\Omega_r$ are disjoint, and by union bound $\mu(\Omega_r) \geq \tilde{p}_{C_r} := (1 - \delta')p_{C_r} - \tilde{\tau} \geq \frac{\tilde{\epsilon}_{TV}}{10|I|}$.

Let $U_r = \Omega_r \cap U_{\text{sample}}$ then Chernoff bound gives

$$\mathbb{P}[|U_r| \geq M\tilde{p}_{C_r}(1 - \tilde{\epsilon}_{TV})] \geq 1 - \exp(-\tilde{\epsilon}_{TV}^2 \tilde{p}_{C_r} M/2) \geq 1 - \exp(-\frac{\tilde{\epsilon}_{TV}^3 M}{20|I|})$$

Let $\mathcal{E}$ be the event $\forall r : |U_r| \geq M\tilde{p}_{C_r}(1 - \tilde{\epsilon}_{TV})$. By union bound, $\mathbb{P}[\mathcal{E}] \geq 1 - |I| \exp(-\frac{\tilde{\epsilon}_{TV}^3 M}{20|I|})$.

Suppose $\mathcal{E}$ happens. Let $U_\emptyset = U_{\text{sample}} \setminus \bigcup_{r \in I'} U_r$ then

$$|U_\emptyset| \leq M - M(1 - \tilde{\epsilon}_{TV}) \sum_{r \in I'} (p_{C_r}(1 - \delta') - \frac{\tilde{\epsilon}_{TV}}{|I|})$$

$$\leq M[1 - (1 - \tilde{\epsilon}_{TV})((1 - \delta')(1 - \tilde{\epsilon}_{TV}) - \tilde{\epsilon}_{TV})]$$

$$\leq M(3\tilde{\epsilon}_{TV} + \delta') \leq 4M\tilde{\epsilon}_{TV}$$

where the second inequality is due to $\sum_{r \in I'} p_{C_r} \geq 1 - \sum_{r \notin J} p_{C_r} \geq 1 - |I| \times \tilde{\epsilon}_{TV}/|I|$.

Note that $\mathcal{L}(X_T^{\nu_{\text{sample}}}|U_{\text{sample}}) = \frac{1}{M} \sum_{x \in U_{\text{sample}}} \mathcal{L}(X_t^{\delta_x}|x)$. Thus, let $\hat{\mu} = \sum_{r \in I'} \frac{|U_r|}{|U_{\text{sample}} \setminus U_\emptyset|} \mu_{C_r}$ and $\tilde{\mu} := \sum_{r \in I'} \frac{|U_r|}{M} \mu_{C_r} + \frac{|U_\emptyset|}{M} \hat{\mu}$, we can apply part 1 of Proposition 9

$$d_{TV}(\mathcal{L}(X_T^{\nu_{\text{sample}}}|U_{\text{sample}}), \tilde{\mu})$$

$$\leq M^{-1}\left(\sum_{r \in I'} \sum_{x \in U_r} d_{TV}(\mathcal{L}(X_T^{\delta_x}|x), \mu_{C_r}) + \sum_{x \in U_\emptyset} d_{TV}(\mathcal{L}(X_T^{\delta_x}|x), \hat{\mu})\right)$$

$$\leq M^{-1}(\tilde{\epsilon}_{TV}(M - |U_\emptyset|) + |U_\emptyset|)$$

$$\leq \tilde{\epsilon}_{TV} + 4\tilde{\epsilon}_{TV} \leq 5\tilde{\epsilon}_{TV}$$

Next, note that $\mu = \sum_{r \in I'} p_{C_r} \mu_{C_r} + p_{C_*} \mu_{C_*}$ and $\tilde{\mu} = \sum_{r \in I'} \bar{p}_{C_r} \mu_{C_r}$ with $\bar{p}_{C_r} := \frac{|U_r|}{M}(1 + \frac{|U_\emptyset|}{|U_{\text{sample}} \setminus U_\emptyset|}) = \frac{|U_r|}{M - |U_\emptyset|}$. We bound $|\bar{p}_{C_r} - p_{C_r}|$.

$$\frac{|U_r|}{M - |U_\emptyset|} \geq \frac{|U_r|}{M} \geq \frac{M(1 - \tilde{\epsilon}_{TV})((1-\delta')p_{C_r} - \frac{\tilde{\epsilon}_{TV}}{|I|})}{M}$$

$$\geq p_{C_r}(1 - \tilde{\epsilon}_{TV} - \delta') - \frac{\tilde{\epsilon}_{TV}}{|I|} \geq p_{C_r} - \tilde{\epsilon}_{TV}(2 + \frac{1}{|I|})$$

We upper bound $|U_r|$. Since $U_r$'s are disjoint,

$$|U_r| \leq M - \sum_{s \in I', s \neq r} |U_s| \leq M - M \sum_{s \in I', s \neq r} \left[ p_{C_s}(1 - \tilde{\epsilon}_{TV} - \delta') - \frac{\tilde{\epsilon}_{TV}}{|I|} \right]$$

$$\leq M(p_{C_r} + 3\tilde{\epsilon}_{TV} + \delta') \leq M(p_{C_r} + 4\tilde{\epsilon}_{TV})$$

where the first inequality is due to the lower bound of $|U_s|$ above and the second inequality is due to $1 - \sum_{s \in I': s \neq r} p_{C_s} \leq p_{C_r} + \tilde{\epsilon}_{TV}$ and $\sum_{s: s \neq r} p_{C_s}(\tilde{\epsilon}_{TV} + \delta') \leq (\tilde{\epsilon}_{TV} + \delta')$. Thus

$$\frac{|U_r|}{M - |U_\emptyset|} - p_{C_r} \leq \frac{M(p_{C_r} + 4\tilde{\epsilon}_{TV})}{M(1 - 4\tilde{\epsilon}_{TV})} - p_{C_r} \leq \frac{4\tilde{\epsilon}_{TV}(p_{C_r} + 1)}{1 - 4\tilde{\epsilon}_{TV}} \leq 16\tilde{\epsilon}_{TV}$$

where in the last inequality, we use the bounds $\tilde{\epsilon}_{TV} \leq 1/10$ and $p_{C_r} \leq 1$. thus

$$|\bar{p}_{C_r} - p_{C_r}| \leq \max\{16\tilde{\epsilon}_{TV}, \tilde{\epsilon}_{TV}(2 + \frac{1}{|I|})\}$$

Part 2 of Proposition 9 gives

$$2d_{TV}(\mu, \tilde{\mu}) \leq \sum_r |\bar{p}_{C_r} - p_{C_r}| + p_{C_*} \leq |I| \max\{16\tilde{\epsilon}_{TV}, \tilde{\epsilon}_{TV}(2 + \frac{1}{|I|})\} + |I| \times \tilde{\epsilon}_{TV}/|I|$$

$$\leq (16|I| + 1)\tilde{\epsilon}_{TV}$$

Thus by triangle inequality,

$$d_{TV}(\mathcal{L}(X_T^{\nu_{\text{sample}}}|U_{\text{sample}}), \mu) \leq d_{TV}(\mathcal{L}(X_T^{\nu_{\text{sample}}}|U_{\text{sample}}), \tilde{\mu}) + d_{TV}(\mu, \tilde{\mu}) \leq 9|I|\tilde{\epsilon}_{TV}$$

Let $M \geq 2 \times 10^4 |I|^3 \epsilon_{TV}^{-3} \log(|I|\tau^{-1}) \geq 20|I|\tilde{\epsilon}_{TV}^{-3} \log(|I|\tau^{-1})$ gives the desired result.

In the proof of Proposition 23, $I' = I$, and we will set $\tilde{\tau} = \min\{\frac{\tilde{\epsilon}_{TV}}{|I|}, p_*/3\}$ which implies $\mu(\Omega_r) \geq p_*/3$ and the event $\mathcal{E}$ happens with probability $1 - |I| \exp(-\frac{p_* \tilde{\epsilon}_{TV}^2 M}{6})$. The rest of the argument follows through, and we need to set $M = 6 \times 10^2 p_*^{-1} |I|^2 \epsilon_{TV}^{-2} \log(|I|\tau^{-1})$ to ensure $\mathcal{E}$ happens with probability $\geq 1 - \tau$.

$\square$

### F.1 GRADIENT ERROR BOUND FOR CONTINUOUS PROCESS

**Definition 5** (Bad set for partition). *Let $\mathcal{C} = \{C_1, \ldots, C_m\}$ be a partition of $S$ i.e. $\bigcup C_r = S$ and $C_r \cap C_{r'} = \emptyset$ if $r \neq r'$. For $x \in \mathbb{R}^d$, let $\mu_{\max, S}(x) = \max_{i \in S} \mu_i(x)$, $i_{\max, S}(x) = \arg\max_{i \in S} \mu_i(x)^4$ and $C_{\max}(x)$ is the unique part of the partition containing $i_{\max, S}(x)$. For $\gamma \in (0, 1)$ let*

$$B_{S, \mathcal{C}, \gamma} = \{x \mid \exists j \in S \setminus C_{\max} : \mu_{\max, S}(x) \leq \gamma^{-1} \mu_j(x)\}$$

*If these are clear from context, we omit $S, \mathcal{C}$ in the subscript.*

**Lemma 12.** *Fix $S \subseteq I$, $\mathcal{C}$ is a partition of $S$, and define $B_\gamma = B_{S, \mathcal{C}, \gamma}$ as in Definition 5. If $\delta_{ij} \leq \delta$ for $i, j$ not being in the same part of the partition then $\mu(B_\gamma) \leq \gamma^{-1}\delta|I|^2/2$.*

---

[4]If there are ties, we break ties according to the lexicographic order of $I$.

**Proposition 25** (Absolute gradient difference bound). *Fix $S \subseteq I$. For $i \in S$, let $\bar{p}_i = p_i p_S^{-1}$ and recall that $\mu_S(x) = \sum_{i \in S} \bar{p}_i \mu_i(S)$. Let $i := i_{\max,S}(x) = \arg\max_{i' \in S'} \mu_{i'}(x)$. Suppose $i \in C \subseteq S$ and for all $j \in S \setminus C$, $\mu_i(x) \geq \gamma^{-1} \mu_j(x)$.*

*Let $G_S(x) = \max_{i \in S} \|\nabla V_i(x)\|$ then*

$$\|\nabla V_S(x) - \nabla V_C(x)\| \leq \frac{4\gamma}{\bar{p}_i} G_S(x)$$

In Appendix H, we will state generalized versions of Definition 5, Lemma 12, and Proposition 25. For proofs of Lemma 12 and Proposition 25, refers to proof of Lemma 16 and Proposition 32 respectively.

The following proposition shows that if the continuous Langevin process $(\bar{Z}_t^{\delta_x})$ initialized at $x$ doesn't hit the bad set $B_{S,\mathcal{C},\gamma}$, then the gradient $\nabla V_S(\bar{Z}_t)$ will be close to the gradient $\nabla V_C(\bar{Z}_t)$ where $C$ is the unique part of the partition $\mathcal{C}$ containing $i_{\max,S}(x)$.

**Proposition 26.** *Fix a set $S$. Suppose we have a partition $\mathcal{C}$ of $S$ as in Definition 5. Suppose for $i \in S$, $\mu_i$ satisfies item 1 of Lemma 5 with $\beta \geq 1$ and $\|u_i - u_j\| \leq L \forall i, j \in S$. Let $\bar{p}_i = p_S^{-1} p_i$, and recall that $\mu_S = \sum_{i \in S} \bar{p}_i \mu_i$. Let $(\bar{Z}_t^{\delta_x})_{t \geq 0}$ be the continuous Langevin diffusion with score function $\nabla V_S$ initialized at $\delta_x$. Fix $\gamma \in (0, 1/2)$. Suppose for any $\eta \in (0,1)$, with probability $1 - \eta/2$, the event $\mathcal{E}_{discrete,\eta}$ happens where $\mathcal{E}_{discrete,\eta}$ is defined by: for all $k \in [0, N-1] \cap \mathbb{N}$,*

$$\|\bar{Z}_{kh}^{\delta_x} - u_S\| \leq \tilde{L} := L + \sqrt{\frac{64}{\alpha} \ln \frac{16N}{\eta}}$$

*and*

$$\bar{Z}_{kh}^{\delta_x} \notin B_{S,\mathcal{C},\gamma}.$$

*Let $T = Nh$ and $C = C_{\max}(x) \in \mathcal{C}$ be the unique part of the partition $\mathcal{C}$ containing $i_{\max,S}(x)$.*

*Fix $\eta \in (0,1)$. Suppose $T \geq 1$,*

$$h \leq \min\{\frac{1}{(\beta^2/\alpha)\ln^2(16T/\eta)}, \frac{1}{40(\beta L)^2}, \frac{1}{2000d(\beta L)^2 \ln(16T/\eta)}\},$$

*$h \ln(1/h) \leq \frac{1}{2000d(\beta L)^2}$ and $h \ln^2(1/h) \leq \frac{1}{1000(\beta^2/\alpha)}$.*

*Then with probability $1 - \eta$,*

$$\forall t \in [0, T] : \|\nabla V_S(\bar{Z}_t^{\delta_x}) - \nabla V_C(\bar{Z}_t^{\delta_x})\| \leq \frac{18\gamma\beta\tilde{L}}{\min_{i \in C} \bar{p}_i}.$$

*Proof.* By Proposition 17, $S$ satisfies item 2 of Assumption 2 with $A_{\text{grad},0} = \beta L$ and $A_{\text{grad},1} = \beta$.

From Proposition 19, with probability $\geq 1 - \eta/2$, the following event $\mathcal{E}_{\text{drift},\eta/2}$ happens

$$\sup_{k \in [0,N-1] \cap \mathbb{N}, t \in [0,h]} \|\bar{Z}_{kh+t}^{\delta_x} - \bar{Z}_{kh}^{\delta_x}\| \leq 4\beta h L + 2\sqrt{\left(\frac{64(\beta h)^2}{\alpha} + 48dh\right) \ln \frac{16N}{\eta}} \leq 1/(20\beta\tilde{L})$$

Here we use the fact that $\ln(16N/\eta) = \ln(16T/(\eta h) = \ln(16T/\eta) + \ln(1/h)$ thus

$$h(\beta L)(\beta\tilde{L}) \leq h(\beta L)^2 + 2h(\beta L)\beta\sqrt{\frac{64}{\alpha} \ln \frac{16T}{\eta}} + 2h(\beta L)\beta\sqrt{\frac{64}{\alpha} \ln(1/h)}$$

$$\leq h(\beta L)^2 + 16\sqrt{h\frac{\beta^2}{\alpha}} \cdot \sqrt{h(\beta L)^2 \ln \frac{16T}{\eta}} + 16\sqrt{h\frac{\beta^2}{\alpha}} \cdot \sqrt{h(\beta L)^2 \ln(1/h)} \leq \frac{1}{160}$$

and

$$\sqrt{\left(\frac{64(\beta h)^2}{\alpha} + 48dh\right) \ln \frac{16N}{\eta}} \times (\beta \tilde{L})$$

$$\leq 10\sqrt{dh}(\sqrt{\ln \frac{16T}{\eta}} + \sqrt{\ln(1/h)}) \left(\beta L + 2\beta\sqrt{\frac{64}{\alpha} \ln \frac{16T}{\eta}} + 2\beta\sqrt{\frac{64}{\alpha} \ln(1/h)}\right)$$

$$\leq 10\left(\sqrt{hd(\beta L)^2 \ln \frac{16T}{\eta}} + \sqrt{hd(\beta L)^2 \ln(1/h)} + 48\sqrt{\frac{h\beta^2}{\alpha}(\ln^2 \frac{16T}{\eta} + \ln^2(1/h))}\right) \leq \frac{1}{80}$$

Suppose both events $\mathcal{E}_{\text{drift},\eta/2}$ and $\mathcal{E}_{\text{drift},\eta/2}$ happen. By union bound, this occurs with probability $\geq 1 - \eta$. We have, by triangle inequality

$$\sup_{k\in[0,N-1]\cap\mathbb{N}, t\in[0,h]} ||\bar{Z}_{kh+t}^{\delta_x} - u_S|| \leq \tilde{L} + 1/(10\beta\tilde{L}) \leq 1.1\tilde{L}$$

and for $i \in S$, by item 1 of Lemma 5 and $||u_i - u_S|| \leq L$

$$||\nabla V_i(\bar{Z}_{kh+t}^{\delta_x})|| \leq \beta(||\bar{Z}_{kh+t}^{\delta_x} - u_S|| + L) \leq 2.2\beta\tilde{L}. \tag{11}$$

For any $i, j \in S$ and $t \in [0, h]$

$$
\begin{aligned}
&\log \frac{\mu_j(\bar{Z}_{kh+t}^{\delta_x})}{\mu_i(\bar{Z}_{kh+t}^{\delta_x})} - \log \frac{\mu_j(\bar{Z}_{kh}^{\delta_x})}{\mu_i(\bar{Z}_{kh}^{\delta_x})} \\
&= V_j(\bar{Z}_{kh}^{\delta_x}) - V_j(\bar{Z}_{kh+t}^{\delta_x}) - (V_i(\bar{Z}_{kh}^{\delta_x}) - V_i(\bar{Z}_{kh+t}^{\delta_x})) \\
&\leq (||\nabla V_i(\bar{Z}_{kh}^{\delta_x})|| + ||\nabla V_j(\bar{Z}_{kh}^{\delta_x})||)||\bar{Z}_{kh+t}^{\delta_x} - \bar{Z}_{kh}^{\delta_x}|| + \beta||\bar{Z}_{kh+t}^{\delta_x} - \bar{Z}_{kh}^{\delta_x}||^2 \\
&\leq 5\beta\tilde{L}(20\beta\tilde{L})^{-1} + \beta(20\beta\tilde{L})^{-2} \leq 1/2
\end{aligned}
\tag{12}
$$

where we use the assumption $\beta \geq 1$.

Below we write $i_{\max}$ instead of $i_{\max,S}$ since $S$ is clear from context. We first argue by induction on $k$ that $i_{\max}(\bar{Z}_{kh}^{\delta_x}) \in C$. The base case $k = 0$ holds trivially. Let $y$ be a realization of $\bar{Z}_{kh}^{\delta_x}$. Condition on $\bar{Z}_{kh}^{\delta_x} = y$, we argue that $i_{\max}(\bar{Z}_{(k+1)h}^{\delta_x}) \in C_{\max}(y)$. Since $C_{\max}(y) = C$ by the inductive hypothesis for $k$, the inductive hypothesis for $k + 1$ follows. Apply Eq. (12) for $t = h$, $i := i_{\max}(y)$ and $j \notin C_{\max}(y)$ gives

$$\log \frac{\mu_j(\bar{Z}_{(k+1)h}^{\delta_x})}{\mu_i(\bar{Z}_{(k+1)h}^{\delta_x})} \leq \log \frac{\mu_j(\bar{Z}_{kh}^{\delta_x})}{\mu_i(\bar{Z}_{kh}^{\delta_x})} + 1/2 = \log \frac{\mu_j(y)}{\mu_{\max}(y)} + 1/2 \leq \log \gamma + 1/2 < 0$$

where the penultimate inequality follows from $\bar{Z}_{kh} \notin B_\gamma$ and $j \notin C_{\max}(y)$, and the final inequality from $\gamma < 1/2$. Thus, for all $j \notin C_{\max}(y)$, $\mu_i(\bar{Z}_{(k+1)h}^{\delta_x}) > \mu_j(\bar{Z}_{(k+1)h}^{\delta_x})$ thus $i_{\max}(\bar{Z}_{(k+1)h}^{\delta_x}) \in C_{\max}(y)$. Finally, we argue for $k \in [0, N-1] \cap \mathbb{N}$ and $t \in (0, h)$, $i_{\max}(\bar{Z}_{kh+t}^{\delta_x}) \in C$ and $\bar{Z}_{kh+t}^{\delta_x} \notin B_{2\gamma}$. Condition on $\bar{Z}_{kh}^{\delta_x} = y$, apply Eq. (12) for $t = h$, $i := i_{\max}(y)$ and $j \notin C_{\max}(y) = C$ gives

$$\log \frac{\mu_j(\bar{Z}_{kh+t}^{\delta_x})}{\mu_i(\bar{Z}_{kh+t}^{\delta_x})} \leq \log \frac{\mu_j(\bar{Z}_{kh}^{\delta_x})}{\mu_i(\bar{Z}_{kh}^{\delta_x})} + 1/2 = \log \frac{\mu_j(y)}{\mu_{\max}(y)} + 1/2 \leq \log \gamma + 1/2 < \log(2\gamma)$$

thus $\forall j \notin C : \mu_{\max}(\bar{Z}_{kh+t}^{\delta_x}) \geq \mu_i(\bar{Z}_{kh+t}^{\delta_x}) \geq (2\gamma)^{-1}\mu_j(\bar{Z}_{kh+t}^{\delta_x})$. Combine this with the bound on $\nabla V_i(\bar{Z}_{kh+t}^{\delta_x})$ in Eq. (11) and using Proposition 25 gives the desired result. Indeed,

$$||\nabla V_S(\bar{Z}_t^{\delta_x}) - \nabla V_C(\bar{Z}_t^{\delta_x})|| \leq \frac{4 \times (2\gamma)G_S(\bar{Z}_t^{\delta_x})}{\bar{p}_i} \leq \frac{18\beta\tilde{L}}{\bar{p}_i}.$$

$\square$

# G ANALYSIS OF LMC WITH APPROXIMATE SCORE

In this section, we prove the main result (Corollary 1).

**Definition 6.** *Let $\mathbb{H}^L$ be the graph where there is an edge between $i, j$ iff $||u_i - u_j|| \leq L$.*

**Proposition 27.** *Suppose $C$ is a connected component of $\mathbb{H}^L$ then for any $i, j \in C$, $||u_i - u_j|| \leq KL$.*

*Proof.* For any $i, j \in C$, there exists a path $i := p_0, p_1, \cdots, p_m := j$ s.t. $||u_{p_s} - u_{p_{s+1}}|| \leq L$. The statement then follows from triangle inequality. □

## G.1 EXPECTED SCORE ERROR BOUND

**Lemma 13.** *Suppose $\mu_i$ satisfies the conditions stated in Lemma 5. Let $u_i$ be as defined in Lemma 5. Fix $S, R \subseteq I, S \cap R = \emptyset$. Let $p_- = \max_{j \in R} p_j$. Suppose for $j \in I \setminus (S \cup R)$, $||u_i - u_j|| \geq L \forall i \in S$, with $L \geq 30 \max\{\sqrt{\frac{d}{\alpha}}, \kappa\sqrt{d}\} \ln(10\kappa)$. If score estimate $s$ satisfies $\mathbb{E}_\mu[||s(x) - \nabla V(x)||^2] \leq \epsilon_{score}^2$ then*

$$\mathbb{E}_{\mu_S}[||\nabla V(x) - \nabla V_S(x)||^2] \leq 3p_S^{-1}(\epsilon_{score}^2 + 8\beta^2 K \exp(-\frac{L^2}{80\kappa}) + 10K^2 p_- \beta^2 L^2)$$

*Proof.* Since $\nabla V_S(x) = p_S^{-1} \sum_{i \in S} \nabla V_i(x)$, we can write

$$||\nabla V(x) - \nabla V_S(x)|| = (\mu(x)p_S\mu_S(x))^{-1} \sum_{i \in S, j \notin S} p_i p_j \mu_i(x)\mu_j(x)||\nabla V_i(x) - \nabla V_j(x)||$$

$$\leq \sum_{i \in S, j \notin S: ||u_i - u_j|| < L} \frac{p_i p_j \mu_i(x)\mu_j(x)||\nabla V_i(x) - \nabla V_j(x)||}{\mu(x)p_S\mu_S(x)}$$

$$+ \sum_{i \in S, j \notin S: ||u_i - u_j|| \geq L} \frac{p_i p_j \mu_i(x)\mu_j(x)||\nabla V_i(x) - \nabla V_j(x)||}{\mu(x)p_S\mu_S(x)}$$

If $||u_i - u_j|| \leq L$ then $||\nabla V_i(x) - \nabla V_j(x)|| \leq \beta(||x - u_i|| + ||x - u_j||) \leq \beta(2||x - u_j|| + L)$ thus the first term can be bounded by

$$(p_S\mu_S(x))^{-1} \left( \sum_{i \in S} \sum_{j \notin S: ||u_i - u_j|| \leq L} \frac{p_i p_j \mu_i \mu_j(x)}{\mu(x)} \beta(2||x - u_j|| + L) \right)$$

$$\leq \beta \sum_{j \notin S: p_j \leq p_-} \frac{p_j \mu_j(x)(2||x - u_j|| + L)}{\mu(x)}$$

where in the last inequality we use the fact that if $||u_i - u_j|| \leq L$ for some $i \in S$ then $j \in R$ and $p_j \leq p_-$. Hence, by Holder's inequality

$$\mathbb{E}_{\mu_S}[||\nabla V(x) - \nabla V_S(x)||^2] \leq 3(\mathbb{E}_{\mu_S}[||s(x) - \nabla V(x)||^2] + A_1 + A_2) \tag{13}$$

with $A_2 = \mathbb{E}_{\mu_S}[(\sum_{i \in S, j \in T_2} \frac{p_i p_j \mu_i(x)\mu_j(x)}{p_S\mu_S(x)\mu(x)})^2]$ and

$$A_1 = \mathbb{E}_{\mu_S} \left[ \beta^2 \left( \sum_{j \notin S: p_j \leq p_-} \frac{p_j \mu_j(x)(2||x - u_j|| + L)}{\mu(x)} \right)^2 \right]$$

$$\leq 5\beta^2 K \sum_{j \notin S: p_j \leq p_-} \int \mu_S(x) \left( \frac{p_j \mu_j(x)}{\mu(x)} \right)^2 (||x - u_j||^2 + L^2)dx$$

$$\leq 5\beta^2 K p_S^{-1} \sum_{j \notin S: p_j \leq p_-} p_j \int \mu_j(x)(||x - u_j||^2 + L)dx$$

$$\leq 10\beta^2 K^2 p_S^{-1} p_- L^2$$

Now we bound the term $A_2$. Let $T_2 = \{j : j \notin S, p_j \geq p_-\}$.

$$\mathbb{E}_{\mu_S}\left[\left(\sum_{i \in S, j \in T_2} \frac{p_i p_j \mu_i(x) \mu_j(x)}{p_S \mu_S(x) \mu(x)}\right)^2\right]$$

$$\leq E_{\mu_S}\left[\frac{\left(\sum_{i \in S, j \in T_2} p_i p_j \mu_i(x) \mu_j(x)||\nabla V_i(x) - \nabla V_j(x)||^2\right)\left(\sum_{i \in S, j \in T_2} p_i p_j \mu_i(x) \mu_j(x)\right)}{(p_S \mu_S(x) \mu(x))^2}\right]$$

$$= p_S^{-1} \int \sum_{i \in S, j \in T_2} \frac{p_i p_j \mu_i(x) \mu_j(x)||\nabla V_i(x) - \nabla V_j(x)||^2}{\mu(x)} dx$$

$$= p_S^{-1} \sum_{i \in S, j \in T_2} p_i \mathbb{E}_{\mu_i}\left[\frac{p_j \mu_j(x)}{\mu(x)}||\nabla V_i(x) - \nabla V_j(x)||^2\right]$$

$$\leq 8 p_S^{-1} K \beta^2 \exp\left(-\frac{L^2}{40\kappa}\right)$$

where in the last inequality we use Proposition 29. Plug these inequalities back into Eq. (13), and use Proposition 28 gives the desired results. $\square$

**Proposition 28.** *Suppose $s$ satisfies Definition 1 then*

$$\mathbb{E}_{\mu_S}[||s(x) - \nabla V(x)||^2] \leq p_S^{-1} \epsilon_{score}^2$$

*Proof.*

$$p_S \mathbb{E}_{\mu_S}[||s(x) - \nabla V(x)||^2] = p_S \int \mu_S(x)||s(x) - \nabla V(x)||^2 dx$$

$$\leq \int (p_S \mu_S(x) + p_{S^c} \mu_{S^c}(x))||s(x) - \nabla V(x)||^2 dx$$

$$= \mathbb{E}_\mu[||s(x) - \nabla V(x)||^2] \leq \epsilon_{score}^2$$

$\square$

**Proposition 29** (Pairwise gradient difference for large $||u_i - u_j||$)**.** *Suppose $\mu_i, \mu_j$ satisfies items 2 and 3 in Lemma 5. Let $u_i, u_j$ be as defined in Lemma 5 and $r := ||u_i - u_j||$. If $\frac{\alpha r^2/2 + c_z}{17/16\alpha + \beta} \geq 4D^2$ then*

$$\mathbb{E}_{x \sim \mu_i}\left[\frac{p_j \mu_j(x)}{\mu(x)}||\nabla V_i(x) - \nabla V_j(x)||^2\right] \leq 8\beta^2 p_i^{-1} r^2 \exp\left(-\frac{\alpha r^2 + c_z}{17\alpha + 16\beta}\right)$$

*Consequently, suppose $\mu_i, \mu_j$ are $\alpha$-strongly log concave and $\beta$-smooth with $\beta \geq 1$ and $\kappa = \beta/\alpha$, and $||u_i - u_j|| \geq L$ with $L \geq 30 \max\{\sqrt{\frac{d}{\alpha}}, \kappa\sqrt{d}\} \ln(10\kappa)$*

$$p_i \mathbb{E}_{x \sim \mu_i}\left[\frac{p_j \mu_j(x)}{\mu(x)}||\nabla V_i(x) - \nabla V_j(x)||^2\right] \leq 8\beta^2 \exp\left(-\frac{L^2}{80\kappa}\right)$$

*Proof.* By Lemma 5, item 2

$$\frac{\mu_i(x)}{\mu_j(x)} \geq \exp\left(-\beta||x - u_i||^2 - z_- + \alpha||x - u_j||^2 + z_+\right)$$

$$\geq \exp\left(\frac{\alpha}{2}||u_i - u_j||^2 - (\alpha + \beta)||x - u_i||^2 + c_z\right)$$

where the second inequality follows from $||u_i - u_j||^2/2 \leq (||x - u_i|| + ||x - u_j||)^2/2 \leq ||x - u_i||^2 + ||x - u_j||^2$ thus

$$\frac{p_j \mu_j(x)}{\mu(x)} \leq \frac{p_j \mu_j(x)}{p_j \mu_j(x) + p_i \mu_i(x)} = \frac{1}{1 + \frac{p_i \mu_i(x)}{p_j \mu_j(x)}} \leq H(||x - u_i||^2)$$

where
$$H(y) = \frac{1}{1 + p_i \exp(\frac{\alpha}{2}||u_i - u_j||^2 - (\alpha + \beta)y + c_z)}.$$

Let $A := \mathbb{E}_{x \sim \mu_i}[H(||x - u_i||^2)]$ and $B := \mathbb{E}_{x \sim \mu_i}[||x - u_i||^2 H(||x - u_i||^2)]$. Using the fact that
$$||\nabla V_i(x) - \nabla V_j(x)||^2 \leq \beta^2(||x - u_i|| + ||x - u_j||)^2 \leq 2\beta^2(4||x - u_i||^2 + ||u_i - u_j||^2),$$

we can bound
$$\mathbb{E}_{x \sim \mu_i}\left[\frac{p_j \mu_j(x)}{\mu(x)}||\nabla V_i(x) - \nabla V_j(x)||^2\right] \leq 2\beta^2(r^2 A + 4B)$$

First we bound $A$. We have
$$\mathbb{E}_{\mu_i}[H(||x - u_i||^2)] = \int_{||x - u_i|| \geq R} H(||x - u_i||^2)\mu_i(x)dx + \int_{||x - u_i|| < R} H(||x - u_i||^2)\mu_i(x)dx$$
$$\leq \mathbb{P}_{x \sim \mu_i}[||x - u_i|| \geq R] + H(R^2)$$
$$\leq \exp(-\alpha(R - D)^2/4) + p_i^{-1}\exp(-\frac{\alpha}{2}r^2 + (\alpha + \beta)R^2 - c_z)$$

where the second inequality follows from $H$ being an increasing function bounded above by $1$, and the third inequality follows from $H(y) \leq p_i^{-1}\exp(-\alpha r^2 + (\alpha + \beta)y - c_z)$. Set $R^2 = \frac{\alpha r^2/2 + c_z}{\alpha + \beta + \alpha/16}$ then $R \geq 2D$ thus $\exp(-\alpha(R - D)^2/4) \leq \exp(-\alpha R^2/16) = \exp(-\alpha r^2/2 + (\alpha + \beta)R^2 - c_z)$. Hence, the rhs is bounded by $2p_i^{-1}\exp(-\frac{\alpha r^2/2 + c_z}{17\alpha + 16\beta})$.

Now we bound $B$. By Holder's inequality
$$\mathbb{E}_{\mu_i}[||x - u_i||^2 H(||x - u_i||^2)]$$
$$\leq \sqrt{\mathbb{E}_{\mu_i}[||x - u_i||^4]} \cdot \sqrt{\mathbb{E}_{\mu_i}[H^2(||x - u_i||^2)]}$$
$$\leq D^2\sqrt{\mathbb{P}_{x \sim \mu_i}[||x - u_i|| \geq \tilde{R}] + H^2(\tilde{R}^2)}$$
$$\leq D^2\sqrt{\exp(-\alpha(\tilde{R} - D)^2/4) + p_i^{-2}\exp(-2\alpha r^2 + 2(\alpha + \beta)\tilde{R}^2 - 2c_z)}$$

where we use the sub-Gaussian moment assumption to bound $\mathbb{E}_{\mu_i}[||x - u_i||^4]$ and the same argument as in the bound for $A$ to bound $\mathbb{E}_{\mu_i}[H^2(||x - u_i||^2)]$, noting that $H^2(\cdot)$ is also an increasing function bounded above by $1$. Set $\tilde{R}^2 = \frac{\alpha r^2/2 + c_z}{\alpha + \beta + \alpha/32}$ then $\tilde{R} \geq 2D$ thus $\exp(-\alpha(\tilde{R} - D)^2/4) \leq \exp(-\alpha \tilde{R}^2/16) = \exp(-2(\alpha r^2 + (\alpha + \beta)\tilde{R}^2 - c_z))$. Hence,
$$B \leq 2D^2 p_i^{-1}\exp(-\tilde{R}^2/32) = 2D^2 p_i^{-1}\exp\left(-\frac{\alpha r^2/2 + c_z}{33\alpha + 32\beta}\right)$$

For the second statement, plug in $D = 5\sqrt{\frac{d}{\alpha}}\ln(10\kappa)$ and $c_z = -\frac{d}{2}\ln(\kappa)$, and use the fact that $\beta \geq 1$, we have
$$\frac{\alpha r^2/2 + c_z}{17\alpha + 16\beta} \geq \frac{0.45\alpha r^2}{33\beta} \geq 80 \times \frac{\beta d}{\alpha}\ln^2(10\kappa) = 4D^2$$

Thus by Proposition 8 and the fact that $L^2 \geq 2\kappa$, $r^2\exp(-\frac{\alpha r^2/2 + c_z}{17\alpha + 16\beta}) \leq r^2\exp(-\frac{r^2}{80\kappa}) \leq L^2\exp(-\frac{L^2}{80\kappa})$ □

**Theorem 6.** *Suppose each $\mu_i$ is $\alpha$ strongly-log-concave and $\beta$-smooth for all $i \in I$ with $\beta \geq 1$. Recall that $|I| = K$. Let $u_i = \arg\min V_i(x)$, $p_* = \min_{i \in I} p_i$, $\kappa = \beta/\alpha$. Set*
$$L_0 = \Theta\left(\kappa^2 K\sqrt{d}(\ln(10\kappa) + \exp(K)\ln(dp_*^{-1}\epsilon_{TV}^{-1}))\right) = \tilde{\Theta}(\kappa^2 K\exp(K)\sqrt{d}).$$

*Let $S$ be a connected component of $\mathbb{H}^L$, where there is an edge between $i, j$ if $||u_i - u_j|| \leq L := L_0/(\kappa K)$. Let $U_{sample}$ be a set of $M$ i.i.d. samples from $\mu_S$ and $\nu_{sample}$ be the uniform distribution over $U_{sample}$. Let $(X_{nh}^{\nu_{sample}})_{n \in \mathbb{N}}$ be the LMC with score $s$ and step size $h$ initialized at $\nu_{sample}$. Set*
$$T = \Theta\left(\alpha^{-1}K^2 p_*^{-1}\ln(10p_*^{-1})\left(\frac{10^8 d(\beta L_0)^3\exp(K)\ln^{3/2}(p_*^{-1})\ln^5\frac{16d(\beta L_0)^2}{\epsilon_{TV}\tau\alpha}}{p_*^{7/2}\epsilon_{TV}^3\alpha^{3/2}}\right)^{2((3/2)^{K-1} - 1)}\right)$$

*Let the step size $h = \Theta\left(\frac{\epsilon_{TV}^4}{(\beta L_0)^4 dT}\right) = \tilde{\Theta}\left(\frac{\epsilon_{TV}^4}{(\beta \kappa^2 K \exp(K))^4 d^3 T}\right)$. Suppose $s$ satisfies Definition 1 with*

$$\epsilon_{score}$$

$$\leq \frac{p_*^{1/2} \epsilon_{TV}^2 \sqrt{h}}{7T}$$

$$= \Theta\left(\frac{p_*^{1/2} \epsilon_{TV}^4}{(\beta L_0)^2 T^{3/2}}\right)$$

$$= \tilde{\Theta}\left(\frac{p_*^{1/2} \epsilon_{TV}^4}{(\beta \kappa^2 K \exp(K))^2 d^{3/2} T^{3/2}}\right)$$

$$= \Theta\left(\frac{p_*^2 \epsilon_{TV}^4 \alpha^{3/2}}{K^3 \ln^{3/2}(10p_*^{-1})(\beta L_0)^2} \left(\frac{p_*^{7/2} \epsilon_{TV}^3 \alpha^{3/2}}{10^8 d(\beta L_0)^3 \exp(K) \ln^{3/2}(p_*^{-1}) \ln^5 \frac{16d(\beta L_0)^2}{\epsilon_{TV} \tau \alpha}}\right)^{3((3/2)^{K-1}-1)}\right)$$

*Suppose the number of samples $M$ satisfies $M \geq 4000 p_*^{-1} \epsilon_{TV}^{-4} K^2 \log(K\epsilon_{TV}^{-1}) \log(\tau^{-1})$, then*

$$\mathbb{P}_{U_{sample}}[d_{TV}(\mathcal{L}(X_T^{\nu_{sample}} \mid U_{sample}), \mu_S) \leq \epsilon_{TV}] \geq 1 - \tau$$

**Corollary 1.** *Suppose $\mu_i$ is $\alpha$ strongly-log-concave and $\beta$-smooth for all $i$ with $\beta \geq 1$. Let $p_* = \min_{i \in I} p_i$. Suppose $s$ satisfies Definition 1. Let $U_{sample}$ be a set of $M$ i.i.d. samples from $\mu$ and $\nu_{sample}$ be the uniform distribution over $U_{sample}$. With $T, h, \epsilon_{score}^2$ as in Theorem 6 and $M \geq 20000 p_*^{-2} \epsilon_{TV}^{-4} K^2 \log(K\epsilon_{TV}^{-1}) \log(K\tau^{-1})$. Let $(X_{nh}^{\nu_{sample}})_{n \in \mathbb{N}}$ be the LMC with score $s$ and step size $h$ initialized at $\nu_{sample}$, then*

$$\mathbb{P}_{U_{sample}}[d_{TV}(\mathcal{L}(X_T^{\nu_{sample}} \mid U_{sample}), \mu) \leq \epsilon_{TV}] \geq 1 - \tau$$

*Proof.* This is a consequence of Theorem 6 and Proposition 31. Here we apply Proposition 31 with

$$M_0 = 4000 p_*^{-1} \epsilon_{TV}^{-4} K^2 \log(K\epsilon_{TV}^{-1}) \log(\tau^{-1}).$$

$\square$

*Proof of Theorem 6.* Let $u_i = \arg\min_x V_i(x)$ then $\nabla V_i(u_i) = 0$. W.l.o.g. we can assume $V_i(u_i) = 0$. By Proposition 27, $\|u_i - u_j\| \leq \hat{L} := KL = L_0/\kappa$ for $i, j \in S$. By Proposition 17, with $u_S = p_S^{-1} \sum_{i \in S} p_i u_i$, $S$ satisfies Assumption 2 with $A_{\text{grad},1} = \beta$, $A_{\text{grad},0} = \beta\hat{L}$, $A_{\text{Hess},1} = 2\beta^2$ and $A_{\text{Hess},0} = 2\beta^2 \hat{L}^2$.

We first show the statement for $M = M_0 := 600 p_*^{-1} \epsilon_{TV}^{-2} K^2 (\log(K\tau^{-1}))$, where we set $\tau = \epsilon_{TV}$, then use Proposition 30 to obtain the result for $M \geq 4000 p_*^{-1} \epsilon_{TV}^{-4} K^4 \log(K\epsilon_{TV}^{-1})) \log(\tau^{-1}) \geq 6M_0 \epsilon_{TV}^{-2} \log(\epsilon_{\text{sample}}^{-1})$.

From this point onward set $\tau = \epsilon_{TV}$ and $M = M_0$ as defined above. Let $(X_{nh}^{\mu_S})_{n \in \mathbb{N}}$ be the LMC with score estimate $s$ and step size $h$ initialized at $\mu_S$ and $(\bar{X}_t^{\mu_S})_{t \geq 0}$ be the continuous Langevin diffusion with score $\nabla V_S$ initialized at $\mu_S$. Let $Q_T$ and $\bar{Q}_T$ denote the distribution of the paths $(X_{nh}^{\mu_S})_{n \in [0, T/h] \cap \mathbb{N}}$ and $(\bar{X}_t^{\mu_S})_{t \in [0, T]}$. Note that $L \geq 50\kappa\sqrt{d}\ln(10\kappa) \geq 10D$, so Lemma 15 gives

$$2d_{TV}(Q_T, \bar{Q}_T)^2 \leq 2h^2 T\beta^6 \hat{L}^6 + 2hTd\beta^4 \hat{L}^4 + T\epsilon_{\text{score},0}^2$$

with $\epsilon_{\text{score},0}^2 := 3p_S^{-1}(\epsilon_{\text{score}}^2 + 8\beta^2 K \exp(-\frac{L^2}{80\kappa}))$. Let $\epsilon_{\text{score},1}^2 = \frac{\epsilon_{TV}^2}{8T}$ and $B = \{z : \|s(z) - V_S(z)\| > \epsilon_{\text{score},1}\}$ then by Markov's inequality $\mu(B) \leq \frac{\epsilon_{\text{score},0}^2}{\epsilon_{\text{score},1}^2} = \frac{8\epsilon_{\text{score},0}^2 T}{\epsilon_{TV}^2}$.

Let $\eta = \epsilon_{TV}\tau$. Suppose $T\epsilon_{\text{score},0}^2 \leq \eta^2/100$ and $h \leq (100)^{-1} \min\{\frac{\eta}{(\beta\hat{L})^3 \sqrt{T}}, \frac{\eta^2}{(\beta\hat{L})^4 dT}\}$ then $d_{TV}(Q_T, \bar{Q}_T) \leq \eta/4$, thus

$$\mathbb{P}[\exists n \in [0, N-1] \cap \mathbb{N} : X_{nh}^{\mu_S} \in B] \leq \mathbb{P}[\exists n \in [0, N-1] \cap \mathbb{N} : \bar{X}_{nh}^{\mu_S} \in B] + d_{TV}(Q_T, \bar{Q}_T)$$

$$\leq A_1 := \frac{T}{h} \times \frac{8T\epsilon_{\text{score},0}^2}{\epsilon_{TV}^2} + \epsilon_{TV}\tau/4$$

Since $\mathbb{E}_{U_{\text{sample}}}[\nu_{\text{sample}}] = \mu_S$, $\mathcal{L}(X_{nh}^{\mu_S}) = \mathbb{E}_{U_{\text{sample}}}[\mathcal{L}(X_{nh}^{\nu_{\text{sample}}}|U_{\text{sample}})]$ and

$$\mathbb{E}_{U_{\text{sample}}}[\mathbb{P}_{\mathcal{F}_n}[\exists n \in [0, N-1] \cap \mathbb{N} : X_{nh}^{\nu_{\text{sample}}} \in B]] = \mathbb{P}[\exists n \in [0, N-1] \cap \mathbb{N} : X_{nh}^{\mu_S} \in B] \leq A_1$$

By Markov's inequality, let $\mathcal{E}_0$ be the event $\mathbb{P}_{\mathcal{F}_n}[\exists n \in [0, N-1] \cap \mathbb{N} : X_{nh}^{\nu_{\text{sample}}} \in B] \leq 2A_1/\tau$ then

$$\mathbb{P}_{U_{\text{sample}}}[\mathcal{E}_0 \text{ occurs}] \geq 1 - \tau/2$$

Suppose $\mathcal{E}_0$ occurs. Let $\nu := \nu_{\text{sample}}$. Let $(Z_{nh}^{\nu_{\text{sample}}})_{n \in \mathbb{N}}$ be the LMC initialized at $\nu$ with score estimate $s_\infty$ defined by

$$s_\infty(z) = \begin{cases} s(z) \text{ if } z \notin B \\ \nabla V_S(z) \text{ if } x \in B \end{cases}$$

then $\sup_{z \in \mathbb{R}^d} ||s_\infty(z) - \nabla V_S(z)||^2 \leq \epsilon_{\text{score},1}^2$.

Note that if $X_{nh} \notin B \forall n \in [0, N-1] \cap \mathbb{N}$ then $Z_{nh}^{\nu_{\text{sample}}} = X_{nh}^{\nu_{\text{sample}}} \forall n \in [0, N] \cap \mathbb{N}$ thus conditioned on $\mathcal{E}_0$ occurs, $d_{TV}(Z_{Nh}^{\nu_{\text{sample}}}, X_{Nh}^{\nu_{\text{sample}}}) \leq 2A_1/\tau$. Let $(\bar{Z}_t^{\nu_{\text{sample}}})_t$ be the continuous Langevin with score $\nabla V_S$ initialized at $\nu$. We want to bound $d_{TV}(Z_{Nh}^{\nu_{\text{sample}}}, \bar{Z}_T^{\nu_{\text{sample}}})$. By sub-Gaussian concentration of $\mu_i$ and union bound over $M$ samples, we have with probability $\geq 1 - \tau/3$, the following event $\mathcal{E}_1$ happens:

$$\sup_{x \sim \nu} \max_{i \in S} ||x - u_i|| \leq \tilde{L} := 2\hat{L} + \sqrt{\frac{4}{\alpha} \ln(\frac{8M}{\tau})} \leq 3\hat{L}$$

since for $\beta \geq 1$, $\sqrt{\frac{4}{\alpha^3} \ln(\frac{8M}{\tau})} \leq 3\kappa^{3/2} \sqrt{\ln(\epsilon_{TV}^{-1})} \leq \hat{L}$.

Let $\mathcal{E}_2$ be the event

$$d_{TV}(\mathcal{L}(\bar{Z}_T^{\nu_{\text{sample}}}|U_{\text{sample}}), \mu_S) \leq \epsilon_{TV}/4$$

By Lemma 11, if $M \geq 605(p_* \epsilon_{TV}^2)^{-1} K^2 \log(K\tau^{-1})$ then $\mathbb{P}_{U^{\text{sample}}}[\mathcal{E}_2] \geq 1 - \tau/6$.

Suppose $\mathcal{E}_0, \mathcal{E}_1, \mathcal{E}_2$ all hold; by union bound, this happens with probability $\geq 1 - \tau$. By Lemma 14, for

$$L_0 = \hat{L} + \kappa\tilde{L} + \sqrt{\frac{d}{\alpha} \ln((2\alpha h)^{-1})} + \sqrt{(16/\alpha + 200dh) \ln(\frac{8T}{h})}$$

we have

$$d_{TV}(Z_{Nh}^{\nu_{\text{sample}}}, \bar{Z}_T^{\nu_{\text{sample}}})^2 \lesssim h^2 T \beta^6 L_0^6 + hTd\beta^4 L_0^4 + \epsilon_{\text{score},1}^2 T/2 \leq \epsilon_{TV}^2/64$$

if $100h \leq \frac{\epsilon_{TV}^2}{(\beta L_0)^4 dT} \leq \frac{\epsilon_{TV}}{(\beta L_0)^3 \sqrt{T}}$.

By triangle inequality

$$\begin{aligned}
&d_{TV}(\mathcal{L}(X_{nh}^{\nu_{\text{sample}}}|U_{\text{sample}}), \mu_S) \\
&\leq d_{TV}(X_{nh}^{\nu_{\text{sample}}}, Z_{Nh}^{\nu_{\text{sample}}}) + d_{TV}(Z_{Nh}^{\nu_{\text{sample}}}, \bar{Z}_T^{\nu_{\text{sample}}}) + d_{TV}(\mathcal{L}(\bar{Z}_T^{\nu_{\text{sample}}}|U_{\text{sample}}), \mu_S) \\
&\leq \frac{16T^2 \epsilon_{\text{score},0}^2}{h\epsilon_{TV}^2 \tau} + \epsilon_{TV}/2 + \epsilon_{TV}/8 + \epsilon_{TV}/4 \leq \epsilon_{TV}
\end{aligned}$$

if $\frac{16T^2 \epsilon_{\text{score},0}^2}{h\epsilon_{TV}^2 \tau} \leq \epsilon_{TV}/8$.

Our choice of parameters satisfies all the conditions mentioned above. Since $h \leq 1/(100\beta d)$, $\beta \geq 1$ and $\hat{L} \geq \sqrt{\frac{d}{\alpha} \ln(10\kappa)} \geq \sqrt{\frac{d}{\alpha} \ln(\alpha^{-1})}$. we can bound $L_0$ by

$$\begin{aligned}
L_0 &\leq 2\hat{L}\kappa + \sqrt{\frac{d}{\alpha} \ln((2\alpha h)^{-1})} + \sqrt{\frac{17}{\alpha} \ln(\frac{8T}{h})} \\
&\leq 3\hat{L}\kappa + \sqrt{\frac{17}{\alpha} \ln(8T)} + 2\ln(1/h)\sqrt{\frac{d+1}{\alpha}} \\
&= 3\hat{L}\kappa + \exp(K)\sqrt{\kappa d} \ln(p_*^{-1} d\epsilon_{TV}^{-1} L_0 \kappa K)
\end{aligned}$$

where we use the bound on $T$ and $h$ to bound $\ln(T)$ and $\ln(1/h)$.

Set $L = 50\kappa\sqrt{d}\ln(10\kappa) + \sqrt{\kappa d}\exp(K)\ln(d\kappa p_*^{-1}\epsilon_{TV}^{-1})$. Since $3\kappa\hat{L} \leq L_0 \leq 5\kappa\hat{L}$,

$$L_0 = \Theta(\kappa^2\sqrt{d}(\ln(10\kappa) + \exp(K)\ln(dp_*^{-1}\epsilon_{TV}^{-1}))).$$

We need to check that $h \lesssim \frac{\epsilon_{TV}^4}{(\beta L_0)^4 dT}$ but this is true due to the choice of $h$. Next, we need to check $\frac{16T^2\epsilon_{score,0}^2}{h\epsilon_{TV}^2\tau} \leq \epsilon_{TV}/8$ and $T^{1/2}\epsilon_{score,0} \leq \eta/10 = \epsilon_{TV}^2/10$. We note that the former implies the latter, and the latter is true since

$$\epsilon_{score} \leq \frac{p_S^{1/2}\epsilon_{TV}^2\sqrt{h}}{7T}$$

and

$$p_S^{-1/2}\beta\sqrt{K}\exp(-\frac{L^2}{160\kappa})T \leq \sqrt{h}\epsilon_{TV}^2/20,$$

which in turn is implied by

$$L/\sqrt{\kappa} \geq \exp(K)\ln(d\kappa p_*^{-1}\epsilon_{TV}^{-1}) \geq 5\ln(Th^{-1}\beta K p_*^{-1}\epsilon_{TV}^{-1})$$

which is true for our choice of $L$, $h$ and $T$.

$\square$

**Lemma 14.** *Fix $S \subseteq I$. For $u_i$ and $D$ as defined in Lemma 5, suppose $||u_i - u_j|| \leq L \forall i, j \in S$ with $L \geq 10D$. Let $\nu_0$ be a distribution s.t. $\sup_{x\sim\nu_0}\max_{i\in S}||x - u_i|| \leq \tilde{L}$. Let $(\bar{Z}_t^{\nu_0})_{t\geq 0}$ the continuous Langevin with score $\nabla V_S$ initialized at $\nu_0$. Let $(Z_{nh}^{\nu_0})_{n\in\mathbb{N}}$ be the LMC with step size $h$ and score $s_\infty$ s.t. $\sup_{x\in\mathbb{R}^d}||s(x) - \nabla V_S(x)|| \leq \epsilon_{score,1}^2$. Suppose $h \leq 1/(30\beta)$ then for $\tilde{D} := 6L + O\left(\kappa\tilde{L} + \sqrt{\frac{d}{\alpha}\ln((2\alpha h)^{-1})}\right) + \sqrt{(\frac{16}{\alpha} + 200dh)\ln(8N)}$, we have*

$$d_{TV}(Z_T^{\nu_0}, \bar{Z}_T^{\nu_0})^2 \leq h^2T\beta^6\tilde{D}^6 + hTd\beta^4\tilde{D}^4 + \epsilon_{score,1}^2T/2$$

*Proof.* To simplify notations, we omit the superscript $\nu_0$ and write $Z_{nh}$ and $\bar{Z}_t$ in the proof instead of $Z_{nh}^{\nu_0}$ and $\bar{Z}_t^{\nu_0}$. Let $\bar{\nu}_h$ be the distribution of $\bar{Z}_h$. First, we bound $\mathcal{R}_2(\bar{\nu}_h||\mu_S)$. By Lemma 10,

$$\mathcal{R}_2(\bar{\nu}_h||\mu_S) \leq O(\alpha^{-1}(\beta\tilde{L})^2 + d\ln((2\alpha h)^{-1}))$$

By Proposition 17, let $u_S = p_S^{-1}\sum_{i\in S}p_i u_i$ then $\mu_S$ satisfies Assumption 2 so

$$\mathbb{P}_{\mu_S}[||x - u_S|| \geq 1.1L + t] \leq \exp(-\alpha t^2/4).$$

Let $N = T/h$. By the change of measure argument in (Chewi et al., 2021, Lemma 24), with probability $\geq 1 - \eta/2$

$$\max_{k\in[1,N-1]\cap\mathbb{N}}||\bar{Z}_{kh} - u_S|| \leq 1.1L + \sqrt{\frac{2}{\alpha}\mathcal{R}_2(\bar{\nu}_h||\mu_S)} + \sqrt{\frac{4}{\alpha}\ln\frac{8N}{\eta}}$$

$$\leq 1.1L + \kappa\tilde{L} + \sqrt{\alpha^{-1}d\ln((2\alpha h)^{-1})} + \sqrt{\frac{4}{\alpha}\ln\frac{8N}{\eta}}.$$

By Proposition 19, this implies that with probability $\geq 1 - \eta$, for $\gamma = \frac{16}{\alpha} + 200dh$

$$\sup_{t\in[0,T]}||\bar{Z}_t - u_S|| \leq \tilde{D} + \sqrt{\gamma\ln(1/\eta)}$$

with $\tilde{D} := 6L + O(\kappa\tilde{L} + \sqrt{\alpha^{-1}d\ln((2\alpha h)^{-1})}) + \sqrt{\gamma\ln(8N)}$. By Proposition 2, this implies, for $p = O(1)$

$$\mathbb{E}[||\bar{Z}_t - u_S||^p] \lesssim (\tilde{D} + \sqrt{\gamma})^p \lesssim \tilde{D}^p$$

where we use the fact that $\sqrt{\gamma} \leq \sqrt{\frac{d+16}{\alpha}} \leq \tilde{D}/50$.

By Proposition 18, for $t \in [kh, (k+1)h]$,

$$\mathbb{E}[||\nabla V(\bar{Z}_{kh}) - \nabla V(\bar{Z}_t)||^2]$$

$$\lesssim \sqrt{\mathbb{E}[A_{\text{Hess},1}^4(||\bar{Z}_{kh} - u_S||^8 + ||\bar{Z}_t - u_S||^8) + A_{\text{Hess},0}^4]}$$

$$\times \sqrt{(t-kh)^3 \int_{kh}^t (A_{\text{grad},1}^4 \mathbb{E}[||\bar{Z}_s - u_S||^4] + A_{\text{grad},0}^4) ds + d^2(t-kh)^2}$$

$$\lesssim (A_{\text{Hess},1}^2 \tilde{D}^4 + A_{\text{Hess},0}^2)(h^2(A_{\text{grad},1}^2 \tilde{D}^2 + A_{\text{grad},0}^2) + dh)$$

$$\lesssim \beta^4(\tilde{D}^4 + L^4)(h^2\beta^2(\tilde{D}^2 + L^2) + dh)$$

$$\lesssim \beta^4 \tilde{D}^4(h^2\beta^2 \tilde{D}^2 + dh)$$

where in the second inequality, we use the moment bounds for $||\bar{Z}_s - u_S||$, in the third inequality, we use Proposition 17 to substitute in the parameters $A_{\text{Hess},1}, A_{\text{Hess},0}, A_{\text{grad},1}, A_{\text{grad},0}$, and in the final bound, we use $\tilde{D} \geq 6L$. Then by Girsanov's theorem (see Lemma 3)

$$2d_{TV}(Z_T^{\nu_0}, \bar{Z}_T^{\nu_0})^2$$

$$\leq \mathbb{E}[\int_0^T ||s(\bar{Z}_{\lfloor t/h \rfloor h}) - \nabla V(\bar{Z}_t)||^2 dt]$$

$$\lesssim \epsilon_{\text{score},1}^2 T + \mathbb{E}[\int_0^T ||\nabla V(\bar{Z}_{\lfloor t/h \rfloor h}) - \nabla V(\bar{Z}_t)||^2 dt]$$

$$\lesssim \epsilon_{\text{score},1}^2 T + h^2 T \beta^6 \tilde{D}^6 + hTd\beta^4 \tilde{D}^4.$$

$\square$

**Lemma 15.** *Suppose the score estimate $s$ satisfies Definition 1. Let $u_i$ and $D$ be defined as in Lemma 5. Let $S$ be a connected component of $\mathbb{H}^L$ with $L \geq 10D$. Let $(X_{nh}^{\mu_S})_{n \in \mathbb{N}}$ be the LMC with score estimate $s$ and step size $h$ initialized at $\mu_S$ and $(\bar{X}_t^{\mu_S})_{t \geq 0}$ be the continuous Langevin diffusion with score $\nabla V_S$ initialized at $\mu_S$. Let $T = Nh$, $Q_T$ and $\bar{Q}_T$ denote the distribution of the paths of $(X_{nh}^{\mu_S})_{n \in [0,T/h] \cap \mathbb{N}}$ and $(\bar{X}_t^{\mu_S})_{t \in [0,T]}$. Then for $\hat{L} = LK$,*

$$2d_{TV}(\bar{Q}_T, Q_T)^2 \leq \mathbb{E}\left[\int_0^T ||s(\bar{X}_{\lfloor t/h \rfloor h}^{\mu_S}) - \nabla V_S(\bar{X}_t^{\mu_S})||^2 dt\right]$$

$$\lesssim 2h^2 T \beta^6 \hat{L}^6 + 2hTd\beta^4 \hat{L}^4 + T\epsilon_{score,0}^2$$

*with $\epsilon_{score,0}^2 := 3p_S^{-1}(\epsilon_{score}^2 + \beta^2 L^2 8K^3 \exp(-\frac{L^2}{40\kappa}))$.*

*Proof.* By Proposition 27, $||u_i - u_j|| \leq \hat{L}$ for $i, j \in S$ and $||u_i - u_j|| > L$ for $i \in S, j \notin S$. Note that since $\mu_S$ is the stationary distribution of the continuous Langevin diffusion with score $\nabla V_S$, the law of $\bar{X}_t^{\mu_S}$ is $\mu_S$ at all time $t$. Thus, for $t \in [kh, (k+1)h]$

$$\mathbb{E}[||s(\bar{X}_{kh}^{\mu_S}) - \nabla V_S(\bar{X}_t^{\mu_S})||^2]$$

$$\leq 2(\mathbb{E}[||s(\bar{X}_{kh}^{\mu_S}) - \nabla V_S(\bar{X}_{kh}^{\mu_S})||^2] + \mathbb{E}[||\nabla V_S(\bar{X}_{kh}^{\mu_S}) - \nabla V_S(\bar{X}_t^{\mu_S})||^2] \quad (14)$$

$$\leq 2(\epsilon_{\text{score},0}^2 + \beta^4 \hat{L}^4(h^2\beta^2 \hat{L}^2 + dh))$$

where in the second inequality, we use Lemma 13 with $R = \emptyset$ to bound the first term and Proposition 19 and Proposition 2 to bound the second term. The argument is similar to the one in the proof of Lemma 14. Let $u_S = p_S^{-1} \sum_{i \in S} p_i u_i$ then $||u_i - u_S|| \leq L \forall i \in S$. For $\tilde{D} = D + \hat{L} \leq 1.1\hat{L}$ and $\gamma = \frac{4}{\alpha}$, since the law of $\bar{X}_t^{\mu_S}$ is $\mu_S$, by Proposition 17

$$\mathbb{P}[||\bar{X}_t^{\mu_S} - u_S|| \geq \tilde{D} + \sqrt{\gamma \ln(1/\eta)}] \leq \eta$$

thus by Proposition 2 and $\tilde{D} \geq \sqrt{100/\alpha}$, for $p = O(1)$, $\mathbb{E}[||\bar{X}_t^{\mu_S} - u_S||^p] \lesssim \tilde{D}^p$. By Proposition 19,

$$\mathbb{E}[||\nabla V_S(\bar{X}_{kh}^{\mu_S}) - \nabla V_S(\bar{X}_t^{\mu_S})||^2] \leq \beta^4(\tilde{D}^4 + \hat{L}^4)(h^2\beta^2(\tilde{D}^2 + \hat{L}^2) + dh)$$
$$\lesssim \beta^4\hat{L}^4(h^2\beta^2\hat{L}^2 + dh)$$

The statement follows from integrating Eq. (14) from 0 to $T$ and Girsanov's theorem (see Lemma 3). $\square$

This proposition is used in Theorem 6 to go from a set of samples of fixed size $M_0$ to a set of samples with size $M$ that can be arbitrarily large.

**Proposition 30.** *Fix distributions $\mu_{sample}, \mu$. For a set $U_{sample} \subseteq \mathbb{R}^d$, let $(X_t^{\nu_{sample}})_t$ be a process initialized at $\nu_{sample}$, the uniform distribution over $U_{sample}$. Suppose there exists $T > 0, \epsilon_{TV} \in (0, 1)$ s.t. with probability $\geq 1 - \epsilon_{TV}/10$ over the choice of $U_{sample}$ consisting of $M_0$ i.i.d. samples from $\mu_{sample}$, $d_{TV}(\mathcal{L}(X_T^{\nu_{sample}}|U_{sample}), \mu) \leq \epsilon_{TV}/10$. Then, for $M \geq 6\epsilon_{TV}^{-2}M_0\log(\tau^{-1})$, with probability $\geq 1 - \tau$ over the choice of $U_{sample}$ consisting of $M$ i.i.d. samples from $\mu_{sample}$,*

$$d_{TV}(\mathcal{L}(X_T^{\nu_{sample}}|U_{sample}), \mu) \leq \epsilon_{TV}/2.$$

*Proof.* Let $U_{\text{sample}}$ be a set of $M$ i.i.d. samples $x^{(1)}, \cdots, x^{(M)}$ from $\mu_{\text{sample}}$. For $r \in \{1, \cdots, \lfloor M/M_0 \rfloor\}$ Let $U_r = \{x^{(i)} : (r-1)M_0 + 1 \leq rM_0\}$ and $U_\emptyset = U_{\text{sample}} \setminus \bigcup_r U_r$. Let $\nu_r$ be the uniform distribution over $U_r$ and $\nu_\emptyset$ be the uniform distribution over $U_\emptyset$. For $m = \lfloor M/M_0 \rfloor$

$$\nu = \frac{M_0}{M}\sum_r \nu_r + \frac{M - M_0 m}{M}\nu_\emptyset$$

Let $\Omega$ be the set of $U \in (\mathbb{R}^d)^{M_0}$ s.t. $d_{TV}(X_T^\nu, \mu) \leq \epsilon_{TV}/2$ with $\nu$ being the uniform distribution over $U$.

Similar to the proof of Proposition 23, if we choose $M/M_0 \geq 6\epsilon_{TV}^{-2}\log(\tau^{-1})$, then with probability $\geq 1 - \tau$, $|\{r : U_r \in \Omega\}| \geq m(1 - \epsilon_{TV}/5)$. By Proposition 9,

$$d_{TV}(\mathcal{L}(X_T^{\nu_{\text{sample}}}|U_{\text{sample}}), \mu) \leq \sum_{r:U_r \in \Omega}\frac{M_0}{M}d_{TV}(\mathcal{L}(X_T^{\nu_r}), \mu) + \frac{M - M_0 m(1 - \epsilon_{TV}/5)}{M}$$
$$\leq \epsilon_{TV}/10 + \epsilon_{TV}^2/6 + \epsilon_{TV}/5 \leq \epsilon_{TV}/2$$

where in the penultimate inequality, we use the definition of $\Omega$, $M_0 m \leq M$ and $M - m_0 M \leq M_0 \leq \epsilon_{TV}^2 M/6$. $\square$

The following proposition combined with Theorem 6 implies Corollary 1.

**Proposition 31.** *For a set $U_{sample} \subseteq \mathbb{R}^d$, let $(X_t^{\nu_{sample}})_t$ be a process initialized at the uniform distribution over $U_{sample}$. Consider distributions $\mu_C$ for $C \in \mathcal{C}$. Let $\mu = \sum p_C\mu_C$ with $p_C > 0$ and $\sum p_C = 1$. Let $p_* = \min p_C$. Suppose there exists $T > 0, \epsilon_{TV} \in (0, 1)$ s.t. with probability $\geq 1 - \frac{\tau}{10|\mathcal{C}|}$ over the choice of $U_{C,sample}$ consisting of $M \geq M_0$ i.i.d. samples from $\mu_C$, $d_{TV}(\mathcal{L}(X_T^{\nu_{C,sample}}|U_{C,sample}), \mu_C) \leq \epsilon_{TV}/10$, where $\nu_{C,sample}$ is the uniform distribution over $U_{C,sample}$. Then, for $M \geq \min p_*^{-1}\{M_0, 20\epsilon_{TV}^{-2}\log(|\mathcal{C}|\tau^{-1})\}$, with probability $\geq 1 - \tau$ over the choice of $U_{sample}$ consisting of $M$ i.i.d. samples from $\mu$,*

$$d_{TV}(\mathcal{L}(X_T^{\nu_{sample}}|U_{sample}), \mu) \leq \epsilon_{TV}.$$

*Proof of Proposition 31.* Since $\mu = \sum_C p_C\mu_C$, a sample $x^{(i)}$ from $\mu$ can be drawn by first sampling $C^{(i)} \in \mathcal{C}$ from the distribution defined by the weights $\{p_C\}_{C \in \mathcal{C}}$, then sample from $\mu_{C^{(i)}}$. Consider $M$ i.i.d. samples $x^{(i)}$ using this procedure, and let $U_C = \{x^{(i)} : C^{(i)} = C\}$. Since $M \geq 20p_*^{-1}\epsilon_{TV}^{-2}$, and $\mathbb{E}[|U_C|] = p_C M$, by Chernoff's inequality and union bound, with probability $1 - \tau/2$ over the randomness of $U_{\text{sample}}$, the following event $\mathcal{E}_1$ holds

$$\forall C : |\frac{|U_C|}{M} - p_C| \leq p_C\epsilon_{TV}/2$$

Suppose $\mathcal{E}_1$ holds. Then, $\frac{|U_C|}{M} \geq p_C(1 - \epsilon_{TV}/2)M \geq M_0$. Thus by union bound, with probability $1 - \epsilon_{TV}/10$ over the randomness of $U_{\text{sample}}$, the following event $\mathcal{E}_2$ holds with $\nu_C$ be the uniform distribution over $U_C$

$$\forall C : d_{TV}(\mathcal{L}(X_T^{\nu_C}|U_C), \mu_C) \leq \epsilon_{TV}/10$$

then let $\tilde{\mu} = \sum_C \frac{|U_C|}{M}\mu_C$, by part 1 of Proposition 9,

$$d_{TV}(\mathcal{L}(X_T^{\nu_{\text{sample}}}|U_{\text{sample}}), \tilde{\mu}) = d_{TV}\left(\sum_C \frac{|U_C|}{M}\mathcal{L}(X_T^{\nu_C}|U_C), \tilde{\mu}\right) \leq \sum_C \frac{|U_C|}{M}\epsilon_{TV}/10 = \epsilon_{TV}/10$$

and $d_{TV}(\tilde{\mu}, \mu) \leq \sum_C |\frac{|U_C|}{M} - p_C| \leq \epsilon_{TV}/2$. Condition on $\mathcal{E}_1$ and $\mathcal{E}_2$ both hold, which happens with probability $1 - \tau$, we have

$$d_{TV}(\mathcal{L}(X_T^{\nu_{\text{sample}}}|U_{\text{sample}}), \mu) \leq d_{TV}(\mathcal{L}(X_T^{\nu_{\text{sample}}}|U_{\text{sample}}), \tilde{\mu}) + d_{TV}(\tilde{\mu}, \mu)$$
$$\leq \epsilon_{TV}/10 + \epsilon_{TV}/2 \leq \epsilon_{TV}$$

$\square$

# H REMOVING THE DEPENDENCY ON $p_* = \min_{i \in I} p_i$.

In this section, we remove the dependency on the minimum weight $p_* = \min_{i \in I} p_i$. The idea is to consider only the components $\mu_i$ with significant weight $p_i$ i.e. $p_i \geq p_{\text{threshold}}$ for some chosen threshold $p_{\text{threshold}}$. In Lemma 17, Theorems 7 and 8, and Corollary 2, we prove analogs of Lemma 11, Theorems 5 and 6, and Corollary 1 respectively with no dependency on $p_*$.

We will need modified versions of Lemma 12 and Proposition 25, which are Lemma 16 and Proposition 32 respectively.

**Definition 7** (Bad set for partition (modified))**.** *Fix $S \subset I, C_* \subseteq S, S' = S \setminus C_*$. Suppose we have a partition $\mathcal{C} = \{C_1, \ldots, C_m\}$ of $S'$. For $x \in \mathbb{R}^d$, let $i_{\max,S'}(x) = \arg\max_{i \in S'} \mu_i(x)$ and $\mu_{\max,S'}(x) = \mu_{i_{\max,S'}(x)} = \max_{i \in S'} \mu_i(x)$ as in Definition 4. Let $C_{\max,S'}(x)$ is the unique part of the partition $\mathcal{C}$ containing $i_{\max,S'}(x)$. For $\gamma \in (0,1), \gamma_* > 0$ let*

$$\tilde{B}_{S,C_*,\mathcal{C},\gamma,\gamma_*}$$
$$= \{x | \exists j \in S' \setminus C_{\max,S'}(x) : \mu_{\max,S'}(x) \leq \gamma^{-1}\mu_j(x) \text{ or } \exists j \in C_* : \mu_{\max,S'}(x) \leq \gamma_*^{-1}\mu_j(x)\}$$

*Note that if $C_* = \emptyset$ then $\tilde{B}_{S,C_*,\mathcal{C},\gamma,\gamma_*} = B_{S,\mathcal{C},\gamma}$ as defined in Definition 5. If they are clear from context, we omit $S, C_*, \mathcal{C}$ in the subscript.*

**Lemma 16** (Bad set bound (generalized version of Lemma 12))**.** *Fix $S \subseteq I, C_* \subseteq C, \mathcal{C}$ be a partition of $S' = S \setminus C_*$. Let $p_S = \sum_{i \in S} p_i$ and $\bar{p}_i = p_i p_S^{-1}$. Recall that $\mu_S = \sum_{i \in S} \bar{p}_i \mu_i$. For $\gamma, \delta \in (0,1)$, define $\tilde{B}_\gamma = \tilde{B}_{S,C_*,\mathcal{C},\gamma,\gamma_*}$ as in Definition 7 with $\gamma_*^{-1} = \gamma^{-1}\delta K/8$. Suppose*

1. *If $i \in C_*$ then $\bar{p}_i \leq \delta/8$*

2. *$\delta_{ij} \leq \delta$ for $i,j$ which are in $S'$ and are not in the same part of the partition $\mathcal{C}$ of $S'$*

*then $\mu_S(\tilde{B}_\gamma) \leq \gamma^{-1}\delta K^2$.*

*Proof of Lemmas 12 and 16.* We prove Lemma 16, then Lemma 12 follows immediately by setting $C_* = \emptyset$ in Definition 7.

Consider $x \in \tilde{B}_\gamma$ s.t. $i_{\max,S'}(x) = i$. For $j \in S'$, let $C(j)$ denote the unique part of the partition $\mathcal{C}$ containing $j$. Let $k = i_{\max 2,S'}(x) = \arg\max_{j \in S' \setminus C(i)} \mu_j(x)$. If $j \in C(i)$ then by definition of $i_{\max,S'}(x) = i, \mu_j(x) \leq \mu_i(x)$. If $j \in S' \setminus C(i)$, then by definition of $k, \mu_j(x) \leq \mu_k(x)$. Let

$$B'_\gamma = \{x \mid \exists j \in S' \setminus C_{\max,S'}(x) : \mu_{\max,S'}(x) \leq \gamma^{-1}\mu_j(x)\}$$

and

$$B_* = \{x \mid \exists j \in C_* : \mu_{\max,S'} \leq \gamma_*^{-1}\mu_j(x)\}.$$

Let $\bar{p}_j = p_j p_S^{-1}$ for $j \in S$. If $x \in B'_\gamma$, $\mu_i(x) \le \gamma^{-1} \mu_k(x)$, and for

$$
\begin{aligned}
\mu_S(x) = \sum \bar{p}_j \mu_j(x) &= \sum_{j \in C(i)} p_j \mu_j(x) + \sum_{j \in S' \backslash C(i)} \bar{p}_j \mu_j(x) + \sum_{j \in C_*} \bar{p}_j \mu_j(x) \\
&\le \sum_{j \in C(i)} \bar{p}_j \mu_i(x) + \sum_{j \in S' \backslash C(i)} \bar{p}_j \mu_k(x) + \sum_{j \in C_*} \bar{p}_j \mu_j(x) \\
&\le \sum_{j \in S'} \bar{p}_j \gamma^{-1} \mu_k(x) + \sum_{j \in C_*} \bar{p}_j \mu_j(x) \\
&\le \gamma^{-1} \mu_k(x) + \sum_{j \in C_*} \bar{p}_j \mu_j(x)
\end{aligned}
$$

Let $\bar{p}_{C_*} := \sum_{j \in C_*} \bar{p}_j$ then $\bar{p}_{C_*} \le K \times \delta/8 \le \gamma^{-1} \delta K/8$ since $\gamma^{-1} > 1$. For $i, k \in S'$, let $\Omega_{i,k}$ be the set of $x$ s.t. $i_{\max,S'}(x) = i$ and $i_{\max 2, S'}(x) = k$. Since $\{\Omega_{i,k} | i, k \in S', C(i) \ne C(k)\}$ forms a partition of $\mathbb{R}^d$, we have

$$
\begin{aligned}
\mu_S(B'_\gamma) &= \sum_{i,k \in S': C(i) \ne C(k)} \int_{x \in B_\gamma \cap \Omega_{i,k}} \mu_S(x) dx \\
&\le \sum_{i,k: C(i) \ne C(k)} \int_{x \in B_\gamma \cap \Omega_{i,k}} (\gamma^{-1} \mu_k(x) + \sum_{j \in C_*} \bar{p}_j \mu_j(x)) dx \\
&= \gamma^{-1} \sum_{i < k: C(i) \ne C(k)} \left( \int_{x \in B_\gamma \cap \Omega_{i,k}} \mu_k(x) dx + \int_{x \in B_\gamma \cap \Omega_{k,i}} \mu_i(x) dx \right) \\
&\quad + \sum_{j \in C_*} \bar{p}_j \left( \sum_{i,k} \mu_j(B_\gamma \cap \Omega_{i,k}) \right) \\
&= \gamma^{-1} \sum_{i < k: C(i) \ne C(k)} \int_{x \in B_\gamma \cap (\Omega_{i,k} \cup \Omega_{k,i})} \min\{\mu_i(x), \mu_k(x)\} dx + \sum_{j \in C_*} \bar{p}_j \\
&\le \gamma^{-1} \sum_{i < k: C(i) \ne C(k)} \delta + \gamma^{-1} \delta K/8 \\
&\le \gamma^{-1} \delta K^2/2 + \gamma^{-1} \delta K/8
\end{aligned}
$$

where in the penultimate inequality, we use the fact that $\delta_{ik} \le \delta$ for $i, k$ which are not in $C_*$ and not in the same part of the partition, and $p_j \le \delta K/2 \le \gamma^{-1} \delta K/2$ for $j \in C_*$.

For $i \in C_*$, let $\Omega_i^*$ be the set of $x$ s.t. $i_{\max,C_*} = i$. If $x \in \Omega_i^* \cap B_*$ then

$$
\mu_S(x) = \sum_{j \in C_*} \bar{p}_j \mu_j(x) + \sum_{j \in S'} \bar{p}_j \mu_j(x) \le \sum_{j \in C_*} \bar{p}_j \mu_i(x) + \sum_{j \in S'} \bar{p}_j \gamma_*^{-1} \mu_i(x) = \mu_i(x)(\bar{p}_{C_*} + \gamma_*^{-1}).
$$

Thus

$$
\begin{aligned}
\mu_S(B_*) &= \sum_{i \in C_*} \int_{x \in B_* \cap \Omega_i^*} \mu_S(x) dx \\
&\le \sum_{i \in C_*} \int_{x \in B_* \cap \Omega_i^*} (\bar{p}_{C_*} + \gamma_*^{-1}) \mu_i(x) dx \\
&\le (\bar{p}_{C_*} + \gamma_*^{-1}) \sum_{i \in C_*} \mu_i(B_* \cap \Omega_i^*) \le (\gamma^{-1} \delta K/8 + \gamma^{-1} \delta K/8) K
\end{aligned}
$$

where in the last inequality we use the definition of $\gamma_*$ and the fact that $\mu_i(B_* \cap \Omega_i^*) \le 1$. Thus by union bound

$$
\mu_S(\tilde{B}_{S,C_*,\mathcal{C},\gamma,\gamma_*}) \le \mu_S(B'_\gamma) + \mu_S(B_*) \le \gamma^{-1} \delta K^2.
$$

$\square$

**Proposition 32** (Absolute gradient difference bound (generalized version of Proposition 25)). *Fix $S \subseteq I, C_* \subseteq S$. Let $S' = S \setminus C_*$. For $i \in S$, let $\bar{p}_i = p_i p_S^{-1}$ and recall that $\mu_S(x) = \sum_{i \in S} \bar{p}_i \mu_i(S)$. Suppose $\bar{p}_j \leq \frac{\delta}{8}$ for $j \in C_*$. Let $i := i_{\max,S'}(x) = \arg\max_{i' \in S'} \mu_{i'}(x)$. Suppose $i \in C \subseteq S'$ and*

1. *$\mu_i(x) \geq \gamma^{-1}\mu_j(x) \forall j \in S' \setminus C$*

2. *$\mu_i(x) \geq \gamma_*^{-1}\mu_j(x) \forall j \in C_*$ where $\gamma_*^{-1} = \gamma^{-1}\delta K/8$.*

*Let $G_S(x) = \max_{i \in S} ||\nabla V_i(x)||$ then*

$$||\nabla V_S(x) - \nabla V_C(x)|| \leq \frac{4\gamma}{\bar{p}_i} G_S(x)$$

*Proof of Proposition 32 and Proposition 25.* We prove Proposition 32, then Proposition 25 follows immediately by setting $C_* = \emptyset$. For $C' \subseteq S$, let $\bar{p}_{C'} = \sum_{i \in C'} \bar{p}_i$. By Proposition 6, we can write

$$\nabla V_S(x) - \nabla V_C(x) = \frac{\bar{p}_C \mu_C(x) \nabla V_C(x) + \sum_{j \in S \setminus C} \bar{p}_j \mu_j(x) \nabla V_j(x)}{\mu_S(x)} - \nabla V_C(x)$$

$$= \frac{\bar{p}_C \mu_C(x) \nabla V_C(x) + \sum_{j \in S \setminus C} \bar{p}_j \mu_j(x) \nabla V_j(x)}{\bar{p}_C \mu_C(x) + \sum_{j \in S \setminus C} \bar{p}_j \mu_j(x)} - \nabla V_C(x)$$

$$= \sum_{j \in S \setminus C} \frac{\bar{p}_j \mu_j(x)}{\bar{p}_C \mu_C(x) + \sum_{j \in S \setminus C} \bar{p}_j \mu_j(x)} (\nabla V_j(x) - \nabla V_C(x))$$

For $j \in S' \setminus C$,

$$\frac{\bar{p}_C \mu_C(x) + \sum_{j' \in S \setminus C} \bar{p}_j \mu_j(x)}{\bar{p}_j \mu_j(x)} \geq \frac{\bar{p}_i \mu_i(x)}{\bar{p}_j \mu_j(x)} \geq \frac{\bar{p}_i}{\bar{p}_j} \gamma^{-1}$$

and for $j \in C_*$, using the upper bound on $p_j$ and the assumption $\mu_i(x) \geq \gamma_*^{-1}\mu_j(x)$

$$\frac{\bar{p}_C \mu_C(x) + \sum_{j' \in S \setminus C} \bar{p}_{j'} \mu_j(x)}{\bar{p}_j \mu_j(x)} \geq \frac{\bar{p}_i \mu_i(x)}{\bar{p}_j \mu_j(x)} \geq \frac{\bar{p}_i \gamma_*^{-1}}{\bar{p}_j} \geq \bar{p}_i K \gamma^{-1}$$

Next, by Proposition 6, $||\nabla V_C(x)|| \leq G_S(x)$ thus,

$$||\nabla V_S(x) - \nabla V_C(x)|| \leq 2 G_S(x) \gamma \left( \sum_{j \in S \setminus (C \cup C_*)} \frac{\bar{p}_j}{\bar{p}_i} + \sum_{j \in C_*} \frac{1}{K \bar{p}_i} \right) \leq \frac{4 \gamma G_S(x)}{\bar{p}_i}$$

$\square$

The following is a modified version of Lemma 11.

**Lemma 17.** *Fix $\epsilon_{TV}, \tau \in (0, 1/2), \delta \in (0, 1]$. Fix $S \subseteq I$. Let $\bar{p}_i = p_i p_S^{-1}$ and recall that $\mu_S = \sum_{i \in S} \bar{p}_i \mu_i$. Suppose for $i \in S$, $\mu_i$ are $\alpha$-strongly log-concave and $\beta$-smooth with $\beta \geq 1$. Let $u_i = \arg\min_x V_i(x)$ and $D \geq 5\sqrt{\frac{d}{\alpha}}$ be as defined in Lemma 5. Suppose there exists $L \geq 10D$ such that for any $i, j \in S$, $||u_i - u_j|| \leq L$. Fix $p_* > 0$. Let $S' = \{i \in S : \bar{p}_i \geq p_*\}$ and $C_* = S \setminus S'$. Let $\mathbb{G}^\delta := \mathbb{G}^\delta(S', E)$ be the graph on $S'$ with an edge between $i, j$ iff $\delta_{ij} \leq \delta$. Let*

$$T = \frac{2 C_{p_*, K}}{\delta \alpha} \left( \ln(\frac{\beta^2 L}{\alpha}) + \ln \ln \tau^{-1} + 2 \ln \tilde{\epsilon}_{TV}^{-1} \right).$$

*and*

$$\delta' = \frac{\delta^{3/2} \alpha^{3/2} p_*^{5/2} \epsilon_{TV}^2 \tau}{10^5 K^5 d (\beta L)^3 \ln^{3/2}(p_*^{-1}) \ln^{3/2} \frac{\beta^2 L \epsilon_{TV}^{-1} \ln \tau^{-1}}{\alpha} \ln^{2.51} \frac{16 d (\beta L)^2}{\epsilon_{TV} \tau \delta \alpha}}.$$

*Suppose $\max_{i \in C_*} \bar{p}_i \leq \delta'/8$ and for all $i, j$ in $S'$ that are not in the same connected component of $\mathbb{G}^\delta$, $\delta_{ij} \leq \delta'$.*

*For $x \in \mathbb{R}^d$, let $(\bar{X}_t^{\delta_x})_{t \geq 0}$ denote the continuous Langevin diffusion with score $\nabla V_S$ initialized at $\delta_x$. Let $C_{\max,S'}$ be the unique connected component of $\mathbb{G}^\delta$ containing $i_{\max,S'}(x) = \arg\max_{i' \in S'} \mu_{i'}(x)$.*

$$\mathbb{P}_{x \sim \mu_S}[d_{TV}(\mathcal{L}(\bar{X}_T^{\delta_x}|x), \mu_{C_{\max,S'}(x)}) \leq \epsilon_{TV}] \geq 1 - \tau$$

*Proof.* The proof is same as Lemma 11, but we replace Lemma 12 with Lemma 16, Proposition 25 with Proposition 32 and Proposition 21 with Proposition 22. Note that we use $\gamma = \frac{p_* \epsilon_{TV}}{100 \tilde{L}\sqrt{T}}$ and $\tilde{B}_\gamma$ as defined in Lemma 16 to ensure that for $y \notin \tilde{B}_\gamma$, $\|\nabla V_{C_{\max,S'}(y)}(y) - \nabla V_S(y)\| \leq \frac{4\gamma(\beta L \sqrt{\ln(\beta L \epsilon_{TV}^{-1}\tau^{-1}T)})}{p_*} \leq \frac{\epsilon_{TV}}{10\sqrt{T}}$ so that we can bound the total variation distance between the continuous Langevin diffusions with scores $\nabla V_S$ and $\nabla V_{C_{\max,S'}(x)}$ by $\epsilon_{TV}/10$. $\qquad\square$

**Theorem 7.** *Fix $\epsilon_{TV}, \tau \in (0, 1/2)$. Fix $S \subseteq I$. Suppose for $i \in S$, $\mu_i$ are $\alpha$-strongly log-concave and $\beta$-smooth with $\beta \geq 1$. Let $u_i = \arg\min_x V_i(x)$ and $D \geq 5\sqrt{\frac{d}{\alpha}}$ be as defined in Lemma 5. Suppose there exists $L \geq 10D$ such that for any $i, j \in S$, $\|u_i - u_j\| \leq L$. Let $U_{sample}$ be a set of $M$ i.i.d. samples from $\mu_S$ and $\nu_{sample}$ be the uniform distribution over $U_{sample}$. Let $(\bar{X}_t^{\nu_{sample}})_{t\geq 0}$ be the continuous Langevin diffusion with score $\mu_S$ initialized at $\nu_{sample}$. For $M \geq 10^5 (\epsilon_{TV}^3)^{-1} K^3 \log(K\tau^{-1})$ and*

$$T \geq \Theta\left(\alpha^{-1}\left(\frac{10^8 d(\beta L)^3 \exp(K)\ln^5 \frac{16d(\beta L)^2}{\epsilon_{TV}\alpha}}{\epsilon_{TV}^3 \alpha^{3/2}}\right)^{\exp(20(K+1))}\right)$$

*then*

$$\mathbb{P}_{U_{sample}}[d_{TV}(\mathcal{L}(\bar{X}_t^{\nu_{sample}}|U_{sample}), \mu_S) \leq \epsilon_{TV}] \geq 1 - \tau$$

*Proof.* For $i \in S$, let $\bar{p}_i = p_i p_S^{-1}$. As in Lemma 17, fix $p_{0,*} = \frac{1}{K}$ and let $S_0' = \{i \in S : \bar{p}_i \geq p_{0,*}\}$, $C_{0,*} = S \setminus S_0'$ then $S_0' \neq \emptyset$, since there must be at least one $i$ s.t. $\bar{p}_i \geq \frac{1}{K}$. By the same argument as in proof of Theorem 5, we take the sequence $1 = \delta_{0,0} > \delta_{0,1} > \cdots > \delta_{0,K}$ where we use the notation $\delta_{0,s}$ to emphasizes its dependency on $p_{0,*}$. If $\max_{i \in C_{0,*}} \bar{p}_i < \frac{\delta_{0,K}}{8}$ then Lemma 17 applies. More precisely, we will use Proposition 24 and the inductive argument on $\delta_{0,s}$ as in the proof of Theorem 5 to show that the continuous Langevin diffusion initialized at $M$ samples will converge to $\mu_S$ after a suitable time $T$ defined by $\delta_{0,K-1}$. If this is not the case, then we let $p_{1,*} = \frac{\delta_{0,K}}{8}$ and $S_1' = \{i \in S : p_i \geq p_{1,*}\}$ then $|S_1'| \geq |S_0'| + 1$. In general, we inductively set $p_{s+1,*} = \frac{\delta_{s,K}}{8}$. If $\max_{i \in C_{s,*}} p_i \leq p_{s+1,*}$ for some $s \leq K - 2$ then we are done, else $C_{K-1,*} = \emptyset$ thus $\min_{i \in S} \bar{p}_i \geq p_{K-1,*}$ and we can use Theorem 5. In all cases, for $p_* = p_{K-1,*}$, the continuous Langevin diffusion initialized at samples converges to $\mu_S$ after time

$$T \geq \Theta\left(\alpha^{-1} K^2 p_*^{-1} \ln(10 p_*^{-1}) \delta_{K-1,K-1}^{-1}\right)$$
$$= \Theta(\alpha^{-1} \Xi^{-\exp(20(K+1))})$$

To justify the above equation, we lower bound $p_* = p_{K-1,*}$ and $\delta_{K-1,K-1}$.

Let $\tilde{\Gamma}_s = \frac{p_{s,*}^{7/2} \epsilon_{TV}^3 \alpha^{3/2}}{8000 d(\beta L)^3 \exp(K)\ln^{3/2}(p_{s,*}^{-1})\ln^5 \frac{16d(\beta L)^2}{\epsilon_{TV}\alpha}} \geq \frac{p_{s,*}^{3.51} \epsilon_{TV}^3 \alpha^{3/2}}{10^5 d(\beta L)^3 \exp(K)\ln^5 \frac{16d(\beta L)^2}{\epsilon_{TV}\alpha}}$ then

$$\delta_{s,K-1} > \delta_{s,K} \geq \tilde{\Gamma}_s^{2((3/2)^{K+1}-1)} \geq p_{s,*}^{7.02((3/2)^{K+1}-1)}\Xi$$

with $\Xi = \left(\frac{\epsilon_{TV}^3 \alpha^{3/2}}{10^5 d(\beta L)^3 \exp(K)\ln^5 \frac{16d(\beta L)^2}{\epsilon_{TV}\alpha}}\right)^{2((3/2)^{K+1}-1)}$ and we can prove by induction on $s$ that

$$p_{s,*} \geq K^{-\exp(10(s+1))}\Xi^{\exp(2(s+1))} \geq \Xi^{\exp(4.9(s+1))},$$

thus

$$\delta_{K-1,K-1}^{-1} \leq (p_{K-1,*}^{7.02((3/2)^{K+1}-1)}\Xi)^{-1}$$
$$\leq (\Xi^{\exp(4.9(K+1))})^{-7.02((3/2)^{K+1}-1)} \cdot \Xi^{-1}$$
$$\leq \Xi^{-\exp(12(K+1))}$$

$\qquad\square$

**Theorem 8.** *Suppose each $\mu_i$ is $\alpha$ strongly-log-concave and $\beta$-smooth for all $i$ with $\beta \geq 1$. Let $u_i = \arg\min V_i(x)$. Set*

$$L_0 = \Theta\left(\kappa^2 K\sqrt{d}(\ln(10\kappa) + \exp(60K)\ln(d\epsilon_{TV}^{-1}))\right).$$

*Let $S$ be a connected component of $\mathbb{H}^L$, where there is an edge between $i, j$ if $||u_i - u_j|| \leq L := L_0/(\kappa K)$. Let $U_{sample}$ be a set of $M$ i.i.d. samples from $\mu_S$ and $\nu_{sample}$ be the uniform distribution over $U_{sample}$. Let $(X_{nh}^{\nu_{sample}})_{n\in\mathbb{N}}$ be the LMC with score $s$ and step size $h$ initialized at $\nu_{sample}$. Set*

$$T = \Theta\left(\alpha^{-1}\left(\frac{10^8 d(\beta L_0)^3 \exp(K)\ln^5 \frac{16d(\beta L_0)^2}{\epsilon_{TV}\alpha}}{\epsilon_{TV}^3 \alpha^{3/2}}\right)^{\exp(20(K+1))}\right)$$

*Let the step size $h = \Theta(\frac{\epsilon_{TV}^4}{(\beta L_0)^4 dT})$. Suppose $p_S \geq \frac{\epsilon_{TV}}{K}$ and $s$ satisfies Definition 1 with $\epsilon_{score} \leq \frac{\epsilon_{TV}^{5/2}\sqrt{h}}{7\sqrt{KT}} \leq \frac{p_S^{1/2}\epsilon_{TV}^2\sqrt{h}}{7T}$. Suppose the number of samples $M$ satisfies $M \geq 10^7\epsilon_{TV}^{-5}K^3\log(K\epsilon_{TV}^{-1})\log(\tau^{-1})$, then*

$$\mathbb{P}_{U_{sample}}[d_{TV}(\mathcal{L}(X_T^{\nu_{sample}} \mid U_{sample}), \mu_S) \leq \epsilon_{TV}] \geq 1 - \tau$$

*Proof.* The proof is identical to proof of Theorem 6, but we plug in $T$ from Theorem 7 instead of Theorem 5. With the same setup as in proof of Theorem 6, $\epsilon_{score,0}^2 = 3p_S^{-1}(\epsilon_{score}^2 + 8\beta^2 K\exp(-\frac{L^2}{80\kappa}))$, thus as long as we assume $p_S \geq \frac{\epsilon_{TV}}{K}$, we can ensure that with our choice of $L$ and $\epsilon_{score}$, $\epsilon_{score,0} \leq \frac{p_S^{1/2}\epsilon_{TV}^2\sqrt{h}}{7T}$ as required. $\square$

**Corollary 2.** *Suppose $\mu_i$ is $\alpha$ strongly-log-concave and $\beta$-smooth for all $i$ with $\beta \geq 1$. Suppose $s$ satisfies Definition 1. Let $U_{sample}$ be a set of $M$ i.i.d. samples from $\mu$ and $\nu_{sample}$ be the uniform distribution over $U_{sample}$. With $T, h, \epsilon_{score}^2$ as in Theorem 8 and $M \geq 10^8\epsilon_{TV}^{-6}K^4\log(K\epsilon_{TV}^{-1})\log(K\tau^{-1})$. Let $(X_{nh}^{\nu_{sample}})_{n\in\mathbb{N}}$ be the LMC with score $s$ and step size $h$ initialized at $\nu_{sample}$, then*

$$\mathbb{P}_{U_{sample}}[d_{TV}(\mathcal{L}(X_T^{\nu_{sample}} \mid U_{sample}), \mu) \leq \epsilon_{TV}] \geq 1 - \tau$$

*Proof.* This is a consequence of Theorem 8 and Proposition 33. Here we apply Proposition 33 with $M_0 = 10^7\epsilon_{TV}^{-5}K^3\log(K\epsilon_{TV}^{-1})\log(K\tau^{-1})$. $\square$

To remove dependency on $p_*$, we will use the following variant of Proposition 31.

**Proposition 33.** *For a set $U_{sample} \subseteq \mathbb{R}^d$, let $(X_t^{\nu_{sample}})_t$ be a process initialized at the uniform distribution $\nu_{sample}$ over $U_{sample}$. Consider distributions $\mu_C$ for $C \in \mathcal{C}$. Let $\mu = \sum p_C\mu_C$ with $p_C > 0$ and $\sum p_C = 1$. Suppose if $p_C \geq \frac{\epsilon_{TV}}{8|\mathcal{C}|}$, there exists $T > 0, \epsilon_{TV} \in (0, 1)$ s.t. with probability $\geq 1 - \frac{\tau}{10|\mathcal{C}|}$ over the choice of $U_{C,sample}$ consisting of $M \geq M_0$ i.i.d. samples from $\mu_C$, $d_{TV}(\mathcal{L}(X_T^{\nu_{C,sample}}|U_{C,sample}), \mu_C) \leq \epsilon_{TV}/10$ where $\nu_{C,sample}$ is the uniform distribution over $U_{C,sample}$. Then, for $M \geq (\frac{\epsilon_{TV}}{8|\mathcal{C}|})^{-1}\min\{M_0, 20\epsilon_{TV}^{-2}\log(|\mathcal{C}|\tau^{-1})\}$, with probability $\geq 1 - \tau$ over the choice of $U_{sample}$ consisting of $M$ i.i.d. samples from $\mu$, $d_{TV}(\mathcal{L}(X_T^{\nu_{sample}}|U_{sample}), \mu) \leq \epsilon_{TV}$*

*Proof of Proposition 33.* The proof is analogous to Proposition 31. We use the same setup and will spell out the differences between the two proofs. Let $\mathcal{C}' = \{C \in \mathcal{C} : p_C \geq \frac{\epsilon_{TV}}{8|\mathcal{C}|}\}$. We redefine the event $\mathcal{E}_1$ as

$$\forall C \in \mathcal{C}' : |\frac{|U_C|}{M} - p_C| \leq p_C\epsilon_{TV}/8$$

and $\mathcal{E}_2$ as, for $\nu_C$ be the uniform distribution over $\mu_C$

$$\forall C \in \mathcal{C}' : d_{TV}(\mathcal{L}(X_T^{\nu_C}|U_C), \mu_C) \leq \epsilon_{TV}/10$$

Let $U_\emptyset = U_{sample} \setminus \bigcup_{C \in \mathcal{C}'} C$ then

$$\frac{|U_\emptyset|}{M} = \frac{\sum_{C \notin \mathcal{C}'} |U_C|}{M} \leq 1 - \sum_{C \in \mathcal{C}'} p_C(1 - \epsilon_{TV}/8) \leq 1 - (1 - \epsilon_{TV}/8)(1 - \epsilon_{TV}/8) \leq \epsilon_{TV}/4.$$

Suppose $\mathcal{E}_1$ and $\mathcal{E}_2$ both hold, which occur with probability $1 - \tau$ by Chernoff's inequality. Let $\tilde{\mu} = \sum_{C \in \mathcal{C}} \frac{|U_C|}{M} \mu_C$. By part 1 of Proposition 9

$$
\begin{aligned}
d_{TV}(\mathcal{L}(X_T^{\nu_{\text{sample}}} | U_{\text{sample}}), \tilde{\mu}) &= d_{TV}\left( \sum_C \frac{|U_C|}{M} \mathcal{L}(X_T^{\nu_C} | U_C), \sum_C \frac{|U_C|}{M} \mu_C \right) \\
&\leq \sum_{C \in \mathcal{C}'} \frac{|U_C|}{M} \epsilon_{TV}/10 + \sum_{C \notin \mathcal{C}'} \frac{|U_C|}{M} \leq \epsilon_{TV}/10 + \epsilon_{TV}/4 \leq \epsilon_{TV}/2
\end{aligned}
$$

By part 2 of Proposition 9

$$
\sum_C |\frac{|U_C|}{M} - p_C| \leq \sum_{C \in \mathcal{C}'} p_C \epsilon_{TV}/8 + \sum_{C \notin \mathcal{C}'} \max\{\frac{|U_C|}{M}, p_C\} \leq \epsilon_{TV}/8 + \epsilon_{TV}/4
$$

By triangle inequality

$$
\begin{aligned}
d_{TV}(\mathcal{L}(X_T^{\nu_{\text{sample}}} | U_{\text{sample}}), \mu) &\leq d_{TV}(\mathcal{L}(X_T^{\nu_{\text{sample}}} | U_{\text{sample}}), \tilde{\mu}) + d_{TV}(\tilde{\mu}, \mu) \\
&\leq \epsilon_{TV}/2 + 3\epsilon_{TV}/8 \leq \epsilon_{TV}
\end{aligned}
$$

$\square$

# I    ADDITIONAL SIMULATIONS

In this section we give some additional details about the simulations in the main text as well as a few supplementary ones.

For the simulation in Figure 1 of the main text, the estimated score function was learned from data by running $3 \times 10^5$ steps of stochastic gradient descent without batching, using a fresh sample at each step with learning rate $10^{-5}$. The loss function was the vanilla score matching loss from Hyvärinen (2005). The neural network architecture used had a single hidden layer with tanh nonlinearity and 2048 units. The stationary distribution shown in the rightmost subfigure was computed by numerical integration of the estimated score.

For the 32-dimensional simulation in Figure 2 of the main text, to train the network we used ADAM with a batch size of 256 examples, again generated fresh each time; we used 200 batches per epoch and 300 epochs and we learned the vanilla score function using an equivalent denoising formulation as in Vincent (2011). Figure 3 is the same but the network was trained for only 30 epochs. In Figure 4, we performed the same experiment as Figure 2 but we used Contrastive Divergence (CD) training Hinton (2012), which has been used by numerous experimental papers in the literature, instead of score matching as the mechanism to learn the approximate gradient. More precisely, we used CD (again trained over 300 epochs) to learn a distribution of the form $\exp(f(x))$ where the potential $f$ was parameterized by a 8192 unit one-hidden-layer neural network with tanh activations. Once this network is learned, $\nabla f$ was used as the approximate score function since this is the score function of the learned distribution. We also observed in Figure 5 that the score matching loss, which was explicitly trained in the other figures, is also monotonically decreasing over time under CD training. The fact that the behavior is somewhat similar under CD and score matching is morally in agreement with theoretical connections between the two observed by Hyvärinen (2007b). Note that in all three of these figures, the same random seeds were used so that colored trajectories will correspond to each other.

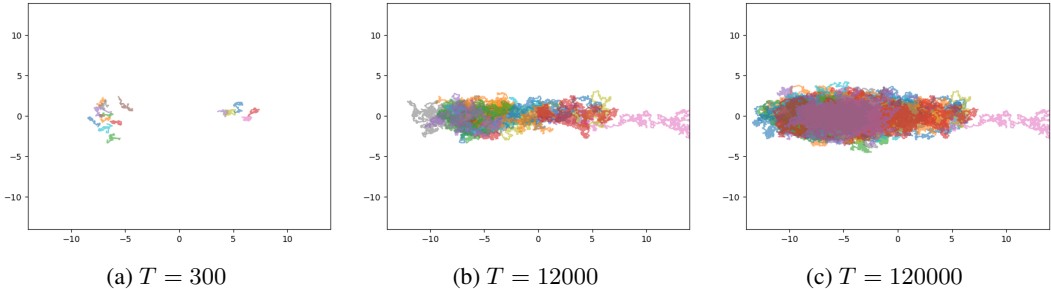

(a) $T = 300$      (b) $T = 12000$      (c) $T = 120000$

Figure 3: Failure to approximate the ground truth with a less accurate score function. This is exact same simulation as Figure 2, except that the network estimating the score function was trained for 30 rather than 300 epochs. We see that while the short time evolution is similar, at moderate times (Figure (b)) the output of the dynamics have drifted away from the true distribution due to accumulation of errors and in particular one trajectory has escaped far right of the rightmost component.

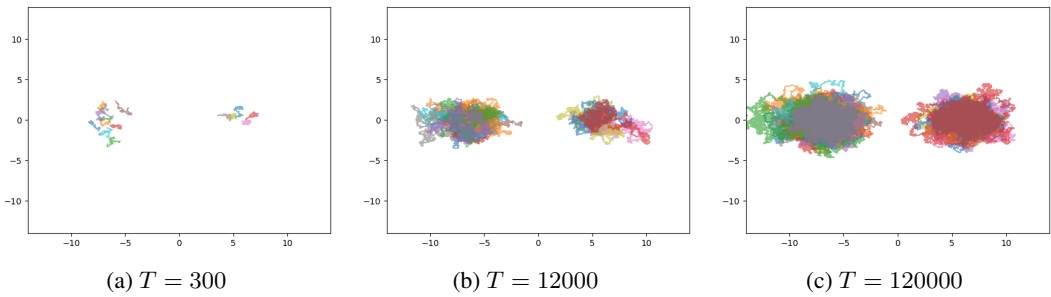

(a) $T = 300$      (b) $T = 12000$      (c) $T = 120000$

Figure 4: Variant of Figure 2 where the approximate score function is learned via Contrastive Divergence (CD) instead of directly trying to match the score function. We used the most basic/efficient version of CD, with only a single step of Langevin dynamics, and we used a larger step size of $0.05$ when sampling in the training loop to compensate for only taking a single step. Qualitatively, the behavior seems similar to Figure 2; at large times, while none of these particular trajectories crossed between components, one trajectory escaped into a low-density region left of the leftmost component.

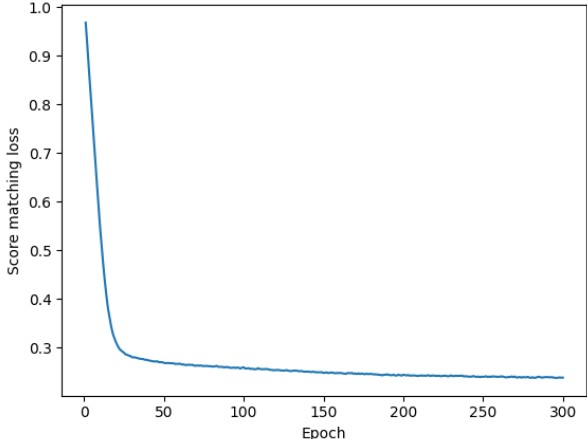

Figure 5: Score matching training loss (precisely the same loss used to train the models in Figures 2 and 3) curve for the CD-trained model in Figure 2. Although the score matching loss is not being explicitly optimized, we see it goes down monotonically over the epochs of CD training nonetheless.

