# OpenReview forum: "Sampling Multimodal Distributions with the Vanilla Score: Benefits of Data-Based Initialization"
_ICLR.cc/2024/Conference — ICLR 2024 poster_

### Official Review · Reviewer_3d8a · 2023-10-23

**Soundness:** 4 excellent
**Presentation:** 3 good
**Contribution:** 3 good
**Rating:** 8
**Confidence:** 3

**Summary:**

This is a theoretical work that addresses the problem, known from the literature, affecting vanilla score matching approaches to learn multimodal distributions from data.
The key idea is to rely on vanilla score matching, and prove that a Langevin diffusion process with early stopping, appropriately initialized at the empirical distribution, successfully generates multimodal distributions.
This result builds on several empirical works that show ways of overcoming the difficulty of using vanilla score functions for the estimation of multimodal distributions, and has the merit of being theoretically sound and -- to some extent -- practical.

**Strengths:**

* I really liked the pedagogical structure of the paper, which is akin to a different kind of literature such as applied mathematics, or conferences on learning theory. The main result is stated, a positioning of the work with respect to both theory and practice is clearly outlined, and open questions are discussed, in a succinct manner. Then, the bulk of the article revolves around the proof strategy used to arrive at the main theorem in the paper.

* I think there are nice connections with recent work such as [1], which deal with methods to factorize the data distribution into a product of conditional probability distributions that are strongly log-concave. Ultimately, the goal is to discover ways of using the vanilla score matching to sample from simpler distributions whereby the noise injection that is typical for score-based generative models is not needed.

* To the best of my understanding, the technical strategy used to prove the main result is correct, well developed and clearly exposed (in the main paper)

[1] Florentin Guth and Etienne Lempereur and Joan Bruna and Stéphane Mallat, Conditionally Strongly Log-Concave Generative Models, ICML 2023

**Weaknesses:**

* I think there is some room for improvement in the overall narrative and exposition of the paper (there are a few typos, easy to fix and not problematic for the technical understanding).
First of all, the assumption that the score is easy to compute and, in particular for some data distributions it can be analytically available, should be emphasized more. Essentially, this work side-steps the problems of learning the score function alltogether: despite citing the seminal work from Hyvarinen 2005, the authors chose not to bring to the readers' attention the fact that in practical cases, even the vanilla score can be hard to compute, as it can require costly computations of the trace of the Hessian of the parametric score, which derives from a rewrite of the Fisher divergence. So, I think it is important to mention that the parametric score function does not come for free in general settings.

* The main results in the paper requires very important ``ingredients'':
1. An early stop mechanism, to set the Langevin Monte Carlo process evolution to stop at a very well defined diffusion time. There is no discussion about the practical implications of the tight bound on $T$ derived in Theorem 1, which is exponential in the data complexity (in the case of the paper, this takes the form of the number of mixture components $K$)

2. The tight bound on step size for the simulation of discrete-time Langevin dynamics decreases exponentially with the data complexity. Although this is not surprising, the more complex the data, the finer-grained your simulation should be, I think a discussion on practical implications is in order.

3. The quality of the approximation of the parametric score function is an assumption that is, in my opinion, very strong, and hardly achievable in practice. This is not a problem in the ``simple'' setting of this work, where data is assumed to be a mixture of $K$ log-concave, smooth components, which is needed to come up with a feasible proof strategy. However, in reality, parametric score functions can only approximate the true score.

4. The number $M$ of i.i.d. samples required to define the initial condition of the discrete Langevin process is not discussed appropriately. From Theorem 1, it seems to me that we need a fairly large number of samples, but I must confess I had a hard time finding the impact of $M$ on the proof strategy outlined in Section 2 of the paper.

* Experiments are weak. Of course we are not talking about using the typical datasets that the current literature on score-based generative modeling. As an example, the results displayed in Figure 2 could have been commented more in the optic of explaining the relation to the above 4 points. Some hints are available in Appendix I, such as details on the learning procedure for the vanilla score network, but I still find it hard to relate such technical details to the hypothesis required for Theorem 1 to be valid.

* [minor weakness] The editorial format of the paper is somehow unconventional. There is no conclusion, and a large fraction of the real-estate available on the 9 pages is dedicated to material that often is given a prime spot in the appendix. If on the one hand I like this presentation style, as it is really helpful to go through the proof strategy detail, the downside is that it substract space to provide more insights on experiments, outline conclusion and compare to prior (albeit experimental) work upon which the authors have drawn inspiration.

**Questions:**

Besides asking authors to provide comments, explanations and eventually additional details for the three main weaknesses discussed above, I have the following question:

* A recent paper [2] studies discrete Langevin processes with approximate scores, and (very informally speaking) also finds that the approximation quality of the distribution obtained by the Langevin process can ``drift away'' in KL terms, from the true distribution if the simulation time $T$ is too large. Do you think there are connections that your work could draw to improve the discussion on the early stop mechanism you devise?

[2] Kaylee Yingxi Yang and Andre Wibisono, Convergence of the Inexact Langevin Algorithm and Score-based Generative Models in KL Divergence, arXiv 2211.01512

** Post rebuttal feedback **
The authors engaged in discussions about my concerns and made an effort to improve their paper. For these reasons I raised my score.

---

> ### Author Response · Authors · 2023-11-15
> **Rebuttal to reviewer 3d8a**
>
> We thank you for your feedback. Our answers to your comments are as followed.
>
> $\textbf{First of all, the assumption that the score is easy to compute and, in particular for some data distributions it can be analytically available, should be emphasized more.
> \\\\
> Essentially, this work side-steps the problems of learning the score function altogether: despite citing the seminal work from Hyvarinen 2005, the authors chose not to bring to the readers' attention the fact that in practical cases, even the vanilla score can be hard to compute, as it can require costly computations of the trace of the Hessian of the parametric score, which derives from a rewrite of the Fisher divergence.
> \\\\
> So, I think it is important to mention that the parametric score function does not come for free in general settings.}$
>
>
> We agree that estimating the vanilla score is an important and nontrivial task.
> First we note that our analysis is following standard practice in theoretical works. In many recent influential papers on generative modeling, the score function/an L2-approximate score function is assumed to be available. See Convergence for score-based generative modeling with polynomial complexity Poster (neurips.cc), ICLR ICLR 2023 Sampling is as easy as learning the score: theory for diffusion models with minimal data assumptions Oral, PMLR Improved Analysis of Score-based Generative Modeling: User-Friendly Bounds under Minimal Smoothness Assumptions (mlr.press). In reality, the score function is assumed to be learned from data, but these previous papers also do not dwell on this fact. We merely follow the convention in the field when presenting our results. In addition, in section 1.1, we have already mentioned that the score function is learned from data. Whether it is easy or not to learn the score function from data is still an active field of research (see Statistical Efficiency of Score Matching: The View from Isoperimetry | OpenReview,  [2306.09332v2] Fit Like You Sample: Sample-Efficient Generalized Score Matching from Fast Mixing Markov Chains (arxiv.org)), and we do not have any intention of claiming that this is an easy task.
>
> The reviewer also mentions some practical concerns regarding computing the Hessian of the score. This happens if we try to fit the score function using Hyvarinen’s algorithm, but all we assume is that it has been fit in some way and alternatives are available. In particular, iIt has been noted in previous works, e.g.  the cited work of Block et al (https://arxiv.org/abs/2002.00107) that in many cases the vanilla score can be reasonably estimated by using denoising score matching with a small amount of noise and this avoids the problem of computing the Hessian.
>
> $\textbf{An early stop mechanism, to set the Langevin Monte Carlo process evolution to stop at a very well defined diffusion time.
> \\\\
> There is no discussion about the practical implications of the tight bound on $T$ derived in Theorem 1, which is exponential in the data complexity (in the case of the paper, this takes the form of the number of mixture components $K$). \\
> The tight bound on step size for the simulation of discrete-time Langevin dynamics decreases exponentially with the data complexity. \\\\
> Although this is not surprising, the more complex the data, the finer-grained your simulation should be, I think a discussion on practical implications is in order.}$
>
> In remark 3, we briefly discuss that T and the number of steps $N=T/h$ depend polynomially in the dimension $d$ and other relevant parameters when $K= O(1).$ Such a polynomial bound is very interesting, since it is known that even for $K=2$, without access to ground truth samples, any algorithm would take exponential time to sample (see the last few lines in “Related theoretical work” in page 4). As we said in the paper, it is a very interesting question for future work whether the dependence on $K$ can be improved in theory.

---

> > ### Author Response · Authors · 2023-11-15
> > **Rebuttal to reviewer 3d8a (part 2)**
> >
> > $\textbf{The quality of the approximation of the parametric score function is an assumption that is, in my opinion, very strong, and hardly achievable in practice. This is not a problem in the ``simple'' setting of this work, where data is assumed to be a mixture of K log-concave, smooth components, which is needed to come up with a feasible proof strategy. However, in reality, parametric score functions can only approximate the true score.}
> >
> > Our assumption on access to a good L2-approximate score function is standard and has been used in highly influential works recently published at NeurIPS Convergence for score-based generative modeling with polynomial complexity Poster (neurips.cc), ICLR ICLR 2023 Sampling is as easy as learning the score: theory for diffusion models with minimal data assumptions Oral, PMLR Improved Analysis of Score-based Generative Modeling: User-Friendly Bounds under Minimal Smoothness Assumptions (mlr.press). See also 2002.00107.pdf (arxiv.org) for a prior result, which shows that a good L2-approximate score function is indeed practically achievable  “optimizing the de-noising autoencoder (DAE) objective does in fact give a L2-accurate score function, with sample complexity depending on the complexity of the function class”. In fact, some recent works, including reference [2] that reviewer 3d8a brings up, even studied a stronger assumption (the MGF assumption).
> >
> > $\textbf{The number M of i.i.d. samples required to define the initial condition of the discrete Langevin process is not discussed appropriately. From Theorem 1, it seems to me that we need a fairly large number of samples, but I must confess I had a hard time finding the impact of M on the proof strategy outlined in Section 2 of the paper}$
> >
> > The number of samples needed is actually quite small, depending only polynomially on the number of components $K$ and the accuracy parameter $\epsilon_{TV}$ (in Theorem 1, there is an additional polynomial dependency on the minimum weight of the components, but this can be removed in the appendix). For concreteness, let’s think of the case when $K = O(1)$ and $\epsilon_{TV}= 1/poly(d)$, then we only need $M = poly(d)$ samples.
> >
> > To see where $M$ appears in the proof, see page 7, the sentence starting with “In a typical set … “. The high-level idea of the proof is, in the low overlap case, we prove that there is a set $\Omega_1$ of samples such that Langevin initialized at a sample from $\Omega_1$ converges to one component $\mu_1$, and we have a similar set $\Omega_2$ for $\mu_2.$. We need $M$ to be large enough so that we have the right proportion of ground truth samples belonging to $\Omega_1$ and $\Omega_2$ i.e. if $\mu= \sum p_i mu_i$ then we need roughly $p_1 M$ samples from $\Omega_1$ and $p_2 M$ samples from $\Omega_2.$
> >
> > This can be guaranteed by Chernoff’s inequality, where the number of samples only need to depend only on the number of components, the accuracy parameters, and the minimum weight of the component.
> > We can further remove the dependency on the minimum weight of the component as well.
> > Intuitively, if a component, say $\mu_1$ have very small weight, then Chernoff bound would have a hard time guaranteeing that the proportion of samples belonging to $\Omega_1$ has sufficient weight; however, we can disregard this component altogether if its weight is small enough compared to the accuracy parameter $\epsilon_{TV}$. Please refer to our answer for a similar question by reviewer vctH for further details.

---

> > > ### Author Response · Authors · 2023-11-15
> > > **Rebuttal to reviewer 3d8a (part 3)**
> > >
> > > Below is the answer to your question:
> > >
> > > $\textbf{A recent paper [2] studies discrete Langevin processes with approximate scores, and (very informally speaking) also finds that the approximation quality of the distribution obtained by the Langevin process can ``drift away'' in KL terms, from the true distribution if the simulation time $T$ is too large. Do you think there are connections that your work could draw to improve the discussion on the early stop mechanism you devise?}$
> > >
> > > We thank you for your interesting question. We are familiar with this work. The assumption on the score approximation used in this work is actually stronger than our L2 error assumption, as they have stated in their abstract “[the MGF error assumption we adopt] is stronger than the L2 error assumption utilized in recent works”. They also assume that the underlying distribution has a good log-Sobolev constant, which is not the case for our setting. With the MGF error assumption, they achieve a stable bound, meaning that the KL divergence between the distribution $\nu_k$ obtained after k steps of Langevin and the target $\nu$ is bounded by a function independent of $k$ (see their Table 1 and Table 2 for a comparison between the unstable bound achieved by previous work and their stable bound). Our work obtains an unstable bound, and this seems inevitable with the L2 score assumption, as noted in their abstract “...which often leads to unstable bounds”. It is possible that with the stronger MGF error assumption, we might achieve a stable bound as well. This is a good direction to study for future work.

---

> > > > ### Comment · Reviewer_3d8a · 2023-11-16
> > > > **rebuttal part 3**
> > > >
> > > > Thanks, I see now that the stronger assumption made in the suggested reference is helpful in deriving stable bounds, and why your weaker assumption does not allow you to achieve the same stability.

---

> > > ### Comment · Reviewer_3d8a · 2023-11-16
> > > **rebuttal part 2**
> > >
> > > Thanks for the clarification on $M$, it makes sense, especially when put in the perspective of $K=O(1)$.

---

> > ### Comment · Reviewer_3d8a · 2023-11-16
> > **rebuttal part 1**
> >
> > Thank you for your clarifications about approximating the score function by learning from data. I have no problems with the assumptions made in the main part of the paper of having the score function directly available. My suggestion was to simply point the reader to the fact that in some practical endeavors score function estimation requires care.
> >
> > Same line of reasoning for the discussion on early stopping. While I greatly appreciate the theoretical result, I was suggesting to point the reader to the fact that when assumptions such as $K=O(1)$ do not hold (or even when data distributions are more complex than mixtures), setting $T$ appropriately requires care.

---

> ### Author Response · Authors · 2023-11-21
> **Learning L2-approximate score function on real datasets**
>
> Thank you for your prompt response. In your review, you commented that one weakness of our paper is "The quality of the approximation of the parametric score function is an assumption that is, in my opinion, very strong, and hardly achievable in practice". In our previous response, we have provided evidence that the assumption of having access to a good L2-approximate score is standard and has been used in several influential prior works. To further corroborate our point, the success of many empirical works is consistent with the belief that L2-approximate score function can be efficiently learned on real datasets, for e.g. MNIST, CelebA, and CIFAR-10.
>
> Prior empirical works show that one can efficiently obtain an L2 estimate of the score/gradient of the log-density function (see the highly cited work [Song-Ermon’19] "https://scholar.google.com/scholar_lookup?arxiv_id=1907.05600"). Note that the score matching algorithm used to estimate the score explicitly minimizes the L2 score error (see also this blog post "Generative Modeling by Estimating Gradients of the Data Distribution | Yang Song (yang-song.net)" https://yang-song.net/blog/2021/score/).[Song-Ermon’19]’s algorithm efficiently learned the score on real datasets (MNIST, CelebA, and CIFAR-10). For more empirical work that employs the framework introduced in [Song-Ermon'19] i.e. learning L2-estimate scores from data samples then using DDPM to produce more samples, see for example [Ho-Jain-Abbeel--NeurIPS'20] https://proceedings.neurips.cc/paper_files/paper/2020/file/4c5bcfec8584af0d967f1ab10179ca4b-Paper.pdf, [Saharia et al.-NeurIPS'22] https://proceedings.neurips.cc/paper_files/paper/2022/file/ec795aeadae0b7d230fa35cbaf04c041-Paper-Conference.pdf.

---

> > ### Comment · Reviewer_3d8a · 2023-11-21
> > **Re: Learning L2-approximate score function on real datasets**
> >
> > Thank you for the additional feedback and the willingness to help readers: I think you have plenty of references to support the claim about having access to good score approximations and the additional material you plan to include can further substantiate the idea.
> >
> > Thank you for the effort you've put in the rebuttal. I will increase the score in my review.

---

> > > ### Author Response · Authors · 2023-11-22
> > >
> > > Thank you for raising the score.

---

### Official Review · Reviewer_7jZP · 2023-10-30

**Soundness:** 2 fair
**Presentation:** 1 poor
**Contribution:** 2 fair
**Rating:** 3
**Confidence:** 5

**Summary:**

This paper utilizes the vanilla score-matching to sample from multimodal distributions. They also show that initialization using data  can help the score matching the ground truth distribution.

**Strengths:**

Sampling from multimodal distributions is an extremely interesting problem.

**Weaknesses:**

- This paper is poorly written and very hard to read and follow. It constantly jumps around. It feels like you are reading lecture notes rather than a paper. The main contribution is hidden in many topics that can easily be moved to the appendix.

- Lack of comprehensive experimental results and applications to real-world data and high dimensional data.

- Lack of comparison to other bounds and theoretical results.

**Questions:**

- How does it compare to other bounds? it seems like this bound provides this on average and not the worst-case. In addition, it is mentioned in previous work: "in high dimensions, it will not be anywhere close to the ground truth distribution unless we have an exponentially large number of samples", how is it not the same case in their paper as well?  Furthermore, What is the computational complexity?
 - The analysis starts with several idealized assumptions, including the exact knowledge of the score function and the assumption that the ground truth distribution is supported within a specific radius. These assumptions might not hold in real-world scenarios, limiting the practical applicability of the results. Could the author please elaborate on that?
- The method relies on various parameters such as $\delta$ (overlap parameter), $H$ (tuning parameter), $\epsilon$ (error threshold), and step size $h$. How can one justify the generalization of the method across different datasets or scenarios due to the fact that sensitivity to these parameters could make the approach highly sensitive to the choice of initial conditions and hyperparameters, making it challenging?
- As mentioned earlier,  the analysis focuses on an idealized scenario and might not directly translate to real-world applications. The conditions and assumptions required for the analysis might be too strict or unrealistic for practical use cases, limiting the method's applicability in real-world data analysis or machine learning tasks. How does this work can be applied in practice? How does it apply to high-dimensional data? Experimental results with high-dimensional data are required to show the efficacy of the procedure.
- The analysis briefly touches upon the scenario where the score function is estimated from data. In practice, obtaining a precise score function estimation can be a challenging task and might introduce significant errors in the analysis.
- The passage does not provide a comprehensive comparison with existing methods or techniques in the field. Without a clear comparison, it's challenging to assess the novelty and superiority of the proposed approach over existing state-of-the-art methods for similar tasks. The analysis primarily focuses on theoretical aspects and lacks empirical validation on real datasets.

---

> ### Author Response · Authors · 2023-11-15
> **Rebuttal for reviewer 7jZP**
>
> We thank you for your feedback, but we respectfully disagree with your assessment.
>
> $\textbf{This paper is poorly written and very hard to read and follow. It constantly jumps around. It feels like you are reading lecture notes rather than a paper. The main contribution is hidden in many topics that can easily be moved to the appendix.}$
>
> We have clearly stated our main result in theorem 1 (the main theorem of the paper). We believe you might be confused by section 2.1, where we present a high-level sketch of our proof. We believe the sketch would be appreciated by readers who want to understand the proof techniques and want to verify the proof, since the full proof in the appendix is long and contains all of the rigorous technical details.
>
> $\textbf{Regarding comparison with previous results:}$
>
> We believe we have provided sufficient comparison with previous theoretical works in page 4 in the paragraph named “related theoretical work”. Please see the answer to your first question for further details.
>
> We have provided experimental results for mixtures of Gaussians in low dimension and relatively high dimension ($d=32$). We also ran the same experiments in other dimensions (e.g. 16, 64,128) and did not observe interesting differences in the results. We didn’t include more experimental results on real datasets since the empirical benefits of Langevin with data-based initialization has been confirmed by previous experimental works such as (Hinton (2012); Xie et al. (2016)).
>
> Answer for your questions (we need to split it into multiple replies due to word limit):
>
> $\textbf{How does it compare to other bounds? Furthermore, What is the computational complexity?}$
>
> We have compared our work with the most closely related previous works, that of Lee et al. (2018) and Ge at al. (2018) on sampling from mixtures of isomorphic Gaussians. Note that because Lee at al. (2018) and Ge et al. (2018) do not assume access to samples from the ground truth, they actually prove an impossibility result when the target mixture distribution $\mu$ is of 2 non-isomorphic Gaussian. In this case, they show that without access to ground truth samples, any algorithm would need exp(d) queries to the log-density and score function, whereas we show that with access to ground truth samples, the standard Langevin can approximately sample from $\mu$ in poly(d) time, where $\mu$ is supported on $\mathbb{R}^d.$  We have emphasized this fact in Remark 3, that, as long as $K$, the number of components, is a constant, the computational complexity of our algorithm is polynomial in all relevant parameters, including the dimension $d$.
>
> $\textbf{it seems like this bound provides this on average and not the worst-case.}$
>
> Our result works with high probability over data sampled from a distribution, which is almost always the setting considered in machine learning and statistics. An efficient algorithm that succeeds in the worst-case scenario is impossible, for example due to the aforementioned impossibility result by Lee et al. (2018) and Ge at al. (2018). Indeed, there is always some possibility that the set of ground truth samples is a fixed set of points (for e.g. the unit sphere centered at 0) that offers no extra information about the target distribution $\mu$, thus you are back in the setting of  Lee et al. (2018) and Ge at al. (2018) where one is not given access to ground truth samples and thus need exp(d) queries to the log-density and score function to produce even one extra sample.
>
> In addition, it is mentioned in previous work: "in high dimensions, it will not be anywhere close to the ground truth distribution unless we have an exponentially large number of samples", how is it not the same case in their paper as well?
>
> We believe that you are referring to the quote in remark 4. What we meant by this is, for the empirical distribution i.e. the uniform distribution over the given set $U_{sample}$ of ground truth samples to be close to the target distribution $\mu$, we’d need $U_{samples}$ to be of size $\exp(d)$ i.e. exponential in the dimension, which is commonly referred to as the “curse of dimensionality.” This is irregardless of the property of the distribution $\mu$, even if $\mu$ is a log-concave distribution or a spherical Gaussian, the above fact still holds.
> As we have stated earlier, our algorithm takes poly(d) time when the target distribution is a mixture of constantly many log-concave distributions, thus escaping this curse of dimensionality.

---

> > ### Author Response · Authors · 2023-11-15
> > **Rebuttal to reviewer 7jZP (continue)**
> >
> > $\textbf{The analysis starts with several idealized assumptions, including the exact knowledge of the score function and the assumption that the ground truth distribution is supported within a specific radius. These assumptions might not hold in real-world scenarios, limiting the practical applicability of the results. Could the author please elaborate on that?}$
> >
> > Our result holds for general log-concave distribution supported on the entire real vector space $\mathbb{R}^d$ and when given access to an approximation of the score function (see Theorem 1). We believe that you are referring to our proof sketch (section 2.1) when you say “the analysis starts with several idealized assumptions, including the exact knowledge of the score function and the assumption that the ground truth distribution is supported within a specific radius”. The purpose of these assumptions is to give a simplified version of the proof so that readers can understand the high-level proof strategy without being bogged down by technical details. Our full proof (in the appendix) does not need these assumptions.  We have emphasized that these assumptions are merely for ease of exposition in the first few lines of “Analysis of idealized diffusion” in page 6.
> >
> > $\textbf{The method relies on various parameters such as $\delta$ (overlap parameter), $H$ (tuning parameter), $\epsilon$ (error threshold), step size $h$. How can one justify the generalization of the method across different datasets or scenarios due to the fact that sensitivity to these parameters could make the approach highly sensitive to the choice of initial conditions and hyperparameters, making it challenging?}$
> >
> > First of all, we would like to make clear that $\delta$ and $H$ only appear in the analysis, and are not hyperparameters of our algorithm. As stated in Theorem 1, the hyperparameters for our algorithm only consist of the step size $h$, the time $T$ to run the Langevin (so the number of steps is $N=T/h$), and the number of samples $M$ to initialize the Langevin at. By $\epsilon$, we believe you are referring to $\epsilon_{TV}$, which is the guaranteed total variation distance between the distribution we sample from and the target distribution $\mu.$ We regard $\epsilon_{TV}$ as part of the input.  We regard $\epsilon_{score}$, which is the bound on the L2-error of the approximate score function, as part of the input as well. The assumption that one is given access to an approximate score function with small L2 error $\epsilon_{score}$  is standard and has appeared in recent influential works on generative modeling (see Convergence for score-based generative modeling with polynomial complexity Poster (neurips.cc) and [2209.11215] Sampling is as easy as learning the score: theory for diffusion models with minimal data assumptions (arxiv.org))
> > Regarding how to choose the hyperparameters. The number of samples, step size $h$, and time $T$ only depends on the dimension $d$, $\epsilon_{TV}$, log-concave and Lipschitz constant $\alpha$ and $\beta,$  and the number of components in the mixture $K.$ Since we know $d$, $\alpha$, $\beta$ and $\epsilon_{TV}$, the only missing information is $K$. Note that an overestimation of $K$ only leads to a longer runtime but doesn’t decrease the quality of the samples produced. So one might either have an upper bound on $K$, or some prior knowledge of what $K$ should be, or one might do a search on $K$.

---

> > > ### Author Response · Authors · 2023-11-15
> > > **Rebuttal to reviewer 7jZP (part 3)**
> > >
> > > $\textbf{As mentioned earlier, the analysis focuses on an idealized scenario and might not directly translate to real-world applications. The conditions and assumptions required for the analysis might be too strict or unrealistic for practical use cases, limiting the method's applicability in real-world data analysis or machine learning tasks. How does this work can be applied in practice?
> > > How does it apply to high-dimensional data? Experimental results with high-dimensional data are required to show the efficacy of the procedure.The analysis briefly touches upon the scenario where the score function is estimated from data. In practice, obtaining a precise score function estimation can be a challenging task and might introduce significant errors in the analysis.}$
> > >
> > > Our work does apply for high-dimensional data: our theoretical result do not assume any upper bound on the dimension $d$, and achieve polynomial in $d$ runtime when $K$ is a constant, and our experiments show that the algorithm produces good result when $d=32$ (and we also ran the experiment in higher dimensions without any interesting difference in results). Our assumption on access to a good L2-approximate score function is standard and has been used in previous highly influential works recently published at NeurIPS Convergence for score-based generative modeling with polynomial complexity Poster (neurips.cc), ICLR ICLR 2023 Sampling is as easy as learning the score: theory for diffusion models with minimal data assumptions Oral, PMLR Improved Analysis of Score-based Generative Modeling: User-Friendly Bounds under Minimal Smoothness Assumptions (mlr.press). See also 2002.00107.pdf (arxiv.org) for a prior result, which shows that a good L2-approximate score function is achievable  “optimizing the de-noising autoencoder (DAE) objective does in fact give a L2-accurate score function, with sample complexity depending on the complexity of the function class”.
> > >
> > > $\textbf{The passage does not provide a comprehensive comparison with existing methods or techniques in the field. Without a clear comparison, it's challenging to assess the novelty and superiority of the proposed approach over existing state-of-the-art methods for similar tasks.}$
> > >
> > > We have provided comparisons to both theoretical and experimental works in the related work sessions. Could you point out any references that we have missed?
> > >
> > > $\textbf{The analysis primarily focuses on theoretical aspects and lacks empirical validation on real datasets.}$
> > >
> > > We agree that our contribution is primarily theoretical. This is because prior works on contrastive divergence training have provided strong empirical evidence that Langevin with data-based initialization converges quickly to the target distribution even when the target is multimodal (see Hinton (2012); Xie et al. (2016)). In particular, the experimental setup in Xie et al (2016) is essentially the same as our contrastive divergence experiment in the appendix, and is run on real data.  See also Using fast weights to improve persistent contrastive divergence | Proceedings of the 26th Annual International Conference on Machine Learning (acm.org), [2205.06924] A Tale of Two Flows: Cooperative Learning of Langevin Flow and Normalizing Flow Toward Energy-Based Model (arxiv.org), [PDF] Implicit Generation and Generalization with Energy Based Models | Semantic Scholar, Learning Non-Convergent Non-Persistent Short-Run MCMC Toward Energy-Based Model (neurips.cc) and *1912.03263.pdf (arxiv.org) (ICLR 2020 oral). Given that Langevin sampling has seen such extensive experimental investigation, we only ran experiments which we hadn’t seen done before (data-based initialization on synthetic data, where we have a good understanding of the ground truth to compare to).

---

> ### Comment · Reviewer_7jZP · 2023-11-22
>
> I'd like to thank the authors for their comprehensive response. I have fully read all the rebuttal arguments and other reviews. Although some of the issues were answered I still believe some of the main issues that were raised by reviewers still stand and I absolutely do not understand some of the reviewers' scores given they think there is a structural issue. At any rate, I very much so appreciate the authors' effort to respond to my concerns.

---

### Official Review · Reviewer_vctH · 2023-11-04

**Soundness:** 4 excellent
**Presentation:** 3 good
**Contribution:** 3 good
**Rating:** 8
**Confidence:** 3

**Summary:**

The paper analyzes Langevin sampling with approximate scores, given some samples of the target distribution are provided.  It is shown that the scheme initialized from a uniform distribution over the available data points generates a sample close to the target in TV distance when stopped at a finite time T. The considered targets are mixtures of strongly log-concave measures and the paper's focus is on showing that with data based initialization, it is possible to remove the dependence on the LSI constant of the whole mixture in Theorem 2.1 of [1] which studied the same scheme.

---
[1] Lee, Holden, Jianfeng Lu, and Yixin Tan. "Convergence for score-based generative modeling with polynomial complexity." Advances in Neural Information Processing Systems 35 (2022): 22870-22882.

**Strengths:**

*The paper formalizes an intuitive result*: Sampling from mixtures is difficult because it is hard to transition in reasonable time from one component to another. If it was known where the mixtures were as well as their relatives weights, then no transitioning would be necessary and sampling could easily be achieved by initializing inside the components' typical sets. This intuitive natural idea (dismissed as too obvious/unrealistic in [1] sec 1.2)  is what is formalized in the paper. Although the result is not very surprising, the authors go through the laborious task of linking the components needed to establish the result: data based initialization with the existing analyses on approximate scores(or inexact langevin) and discretization of langevin diffusions.

 *A clear and easy to follow proof outline*: There is a nice simple setting that is detailed in the main text to understand the paper's strategy, which is quite useful since the paper's result requires lengthy combinations of several results.

---
[1] Lee, Holden, Andrej Risteski, and Rong Ge. "Beyond log-concavity: Provable guarantees for sampling multi-modal distributions using simulated tempering langevin monte carlo." Advances in neural information processing systems 31 (2018).

**Weaknesses:**

- *Structure and unconvincing arguments*: Some sections could be better restructured namely section 1.2. It jumps from motivation to related work to possible extensions. Some remarks can also feel a little unconvincing there. For example, computational hardness arguments are invoked in the motivation but the main idea of noised score learning is to have an annealing scheme where the denoising is only performed from one noise level to a slightly lower noise level. Annealing breaks down the difficulty of denoising, so the paragraph is criticizing an alternative that is never used. A further minor point: remark 5 is an excessively long paragraph to say the mean is not in the typical set in high dimensions.
- *Log sobolev constant for well connected mixtures* : A contribution of the paper is extending the Poincare inequality through decompositions result of Madras & Randall (2002) to the log sobolev inequality. It would be very surprising if such an extension has not already been done as the result is old and functional inequalities are heavily researched. I would kindly ask the authors to check and possibly include references [1] to ensure that they are not missing prior work.
- *On significance*: For the MCMC community this result will not be worthwhile as samples aren't available. For generative modeling where there is a dataset, the better modelling of what practitioners do uses time-varying score functions and so what is done in practice does not correspond to Langevin with a time independent score. The "long line" of experimental work that uses vanilla langevin is claimed to exist but never cited. The paper answers a small interesting curiosity related to data based initialization for sampling from mixtures with limited links with practice.
---
[1] Jerrum, M., Son, J. B., Tetali, P., & Vigoda, E. (2004). Elementary bounds on Poincaré and log-Sobolev constants for decomposable Markov chains.

**Questions:**

- *Number of samples needed, Proposition 23 (and 24)*: From my understanding, for the approach to work, the available samples must have the correct weights of the components. Could the authors explain why the $M$ does not seem to depend on properties of $\mu$ besides the number of components ?
- *Samples to train and samples to initialize*: Presumably the approximate score is learnt from the same available samples $M$ used to initialize. This dependence could break some concentration arguments. Could the authors briefly discuss whether it is necessary to hold out some samples for initialization when learning the score ?

---

> ### Author Response · Authors · 2023-11-15
> **Rebuttal to reviewer vctH**
>
> We thank you for your feedback. Below are our responses to your comments and questions.
>
> $\textbf{Some remarks can also feel a little unconvincing there. For example, computational hardness arguments are invoked in the motivation but the main idea of noised score learning is to have an annealing scheme where the denoising is only performed from one noise level to a slightly lower noise level. Annealing breaks down the difficulty of denoising, so the paragraph is criticizing an alternative that is never used.}$
>
> The point we have made here is correct, nonetheless. We agree that in annealing schemes, the overall task of denoising is broken down into getting from one noise level to a slightly lower noise level. But in this setting, it actually doesn’t matter what learning scheme is used to try to get at the annealed score functions, because the fundamental bottleneck is that the annealed score function (from the displayed equation in the top of page 4) in the sparse spiked Wigner model is hard to approximately compute. That score function is exactly what is needed to slightly denoise the data, and since we cannot compute it in polynomial time, we definitely cannot learn a computationally efficient model to do it.
>
> "A contribution of the paper is extending the Poincare inequality through decompositions result of Madras & Randall (2002) to the log sobolev inequality.
> It would be very surprising if such an extension has not already been done as the result is old and functional inequalities are heavily researched.
> I would kindly ask the authors to check and possibly include references [1] to ensure that they are not missing prior work."
>
> We were also surprised that despite the long history of research on functional inequalities, a result like ours was not known. The most closely related prior results are [1812.06464] Poincaré and log-Sobolev inequalities for mixtures (arxiv.org) (2019) and 2102.11476.pdf (arxiv.org) (2021) but the former only consider mixture of 2 components ($K=2$) while the latter only applies for mixture of isomorphic Gaussians. Given the recent date of these publications, especially the 2019 paper which studies log-Sobolev of mixtures in an identical setting to ours and obtains a much weaker result, it is highly unlikely that our result for log-Sobolev of mixtures was previously known.
> We are familiar with reference [1] (we believe you mean Jerrum, M., Son, J. B., Tetali, P., & Vigoda, E. (2004)., though there are two different references labeled [1] in your response).
> This work applies to a different setting, when the state space (read, $\mathbb{R}^d$) are decomposed into disjoint subsets, and the log-Sobolev of a Markov chain over the entire state space is related to that of Markov chains restricted to within these disjoint subsets (the restriction chains) as well as the projection chain which moves between these subsets. Their result does not apply to our setting, since (1) we don’t have a decomposition of the state space into disjoint subsets, and it’s unclear if any natural decomposition can be made when the Gaussian components have high overlap; (2) the Langevin with respect to one component $\mu_i$ of the mixture $\mu=\sum_i p_i \mu_i$ (i.e. driven by the gradient $\nabla \log \mu_i$ is not the restriction of the Langevin with respect to $\mu$  (i.e. driven by the gradient $\nabla \log \mu$) on any subset of the state space.
> This is why recent works by Schliting (2019) and Chen, Chewi, Niles-Weed (2021) have needed a new analysis and also why we need a new analysis as well.

---

> > ### Author Response · Authors · 2023-11-15
> > **rebuttal to reviewer vctH**
> >
> > $\textbf{On significance: For the MCMC community this result will not be worthwhile as samples aren't available.}$
> >
> > The problem of sampling from mixtures of log-concave distributions or Gaussians is a well-known challenge in the community. The prior works Lee et al. (2018) and Ge et al. (2018) require that the Gaussian components of the mixture are isomorphic i.e. having the same covariance matrix, and show that it takes $\exp(d)$ many log-density and gradient queries to sample from mixtures of 2 non-isomorphic Gaussians, where $d$ is the dimension. Our result offers a way to overcome this impossibility result when given access to ground truth samples, which are freely available in generative modeling. Note that the complexity of our algorithm is polynomial in $d$ and all relevant parameters when the number of components is a constant $K.$
> > Our result can be interpreted more broadly as showing the advantages of non-worst-case initialization on the convergence of Markov chains, which has received lots of recent interest from the MCMC community as well [2106.11296] Low-temperature Ising dynamics with random initializations (arxiv.org) (STOC 2022 and Annals of Applied Probability) and 2305.13239.pdf (arxiv.org) (RANDOM 2023).  One may also consider the setting where one wants to produce many samples from the target $\mu$, and can at first use a more complicated or slower Markov chain to produce some samples, then use those samples as initialization for a simpler chain. This can reduce the amortized cost of sampling, when one needs to produce a large number of samples. The problem of building a faster sampling algorithm given access to a slower sampling algorithm  has attracted a lot of interest from practitioners, see e.g. the cited paper of Lawrence and Yamauchi.
> >
> > "For generative modeling where there is a dataset, the better modelling of what practitioners do uses time-varying score functions and so what is done in practice does not correspond to Langevin with a time independent score. The "long line" of experimental work that uses vanilla langevin is claimed to exist but never cited. "
> >
> > We have cited the use of Langevin with data-based initialization in page 2, 2nd paragraph  “...e.g. Hinton (2012); Xie et al. (2016)”. For more experimental works, see  Using fast weights to improve persistent contrastive divergence | Proceedings of the 26th Annual International Conference on Machine Learning (acm.org), [2205.06924] A Tale of Two Flows: Cooperative Learning of Langevin Flow and Normalizing Flow Toward Energy-Based Model (arxiv.org), [PDF] Implicit Generation and Generalization with Energy Based Models | Semantic Scholar, Learning Non-Convergent Non-Persistent Short-Run MCMC Toward Energy-Based Model (neurips.cc) and *1912.03263.pdf (arxiv.org) (ICLR 2020 oral).
> > Though diffusion models such as DDPM (we believe this is what you have in mind when mentioning time-varying score function) are recently very popular, Langevin with time independent score a.k.a. vanilla Langevin has also seen successes and has been used for a long time with plenty of successes as well. And understanding the simpler vanilla Langevin better may also lead to new insights into other methods down the line. Finally, revisiting older and alternative methods besides the currently most popular one seems to have a clear intellectual value — understanding the weaknesses of vanilla score matching (as discussed e.g. by Song and Ermon) has been one of the directions which led to current progress in diffusion models, whereas at the time they did the work GANs were probably the dominant paradigm in generative modeling.
> >
> > "The paper answers a small interesting curiosity related to data based initialization for sampling from mixtures with limited links with practice."
> >
> > We respectfully disagree with this assessment. Given the success and the popularity of Langevin of data-based initialization, we believe our work is important as the first work to give theoretical justification for this practice. From a purely theoretical perspective, our work circumvents an impossibility result on the well-known problem of sampling from mixtures of log-concave distributions by considering this problem in the context of generative modeling.

---

> > > ### Author Response · Authors · 2023-11-15
> > > **Rebuttal to reviewer vctH (part 3)**
> > >
> > > $\textbf{On number of samples needed}$
> > >
> > > The high-level idea of the proof is, in the low overlap case, we prove that there is a set $\Omega_1$ of samples such that Langevin initialized at a sample from $\Omega_1$ converges to one component $\mu_1$, and we have a similar set $\Omega_2$ for $\mu_2$. We need $M$ to be large enough so that we have the right proportion of ground truth samples belonging to $\Omega_1$ and $\Omega_2$ i.e. if $\mu= \sum p_i mu_i$ then we need roughly $p_1 M$ samples from $\Omega_1$ and $p_2 M$ samples from $\Omega_2.$ This can be guaranteed by Chernoff inequality. The Chernoff bound only uses the number of sets/components $K$, the accuracy parameter and the weights of the component (since they are approximately the mass of the sets $Omega_i$). We can further remove the dependency on the weights as well, by noting that the component with smaller weights can be disregarded since they don’t contribute much to the target distribution anyway. Please refer to the answer for reviewer 3d8a on a similar question for more details.
> > >
> > > $\textbf{Samples to train and samples to initialize: Presumably the approximate score is learnt from the same available samples $M$ used to initialize. This dependence could break some concentration arguments. Could the authors briefly discuss whether it is necessary to hold out some samples for initialization when learning the score?}$
> > >
> > >
> > > This is an interesting question. We agree that it can create a technical difficulty in the analysis if you use the same samples for learning and initialization. We strongly suspect it can be handled at the cost of making the proof more complicated, and this can be an interesting question for future work.
> > > If we want to stick within the realm of the current theorem, it is not a significant issue to save a few samples for initialization, because generally speaking $M$ should be very small compared to the number of samples needed to learn the score function — importantly, $M$ has no dependence on the dimension $d.$
> > >
> > > $\textbf{A further minor point: remark 5 is an excessively long paragraph to say the mean is not in the typical set in high dimensions.}$
> > >
> > > We are always happy to take writing suggestions. To clarify, Remark 5 says that initializing the Langevin at the means of the Gaussian components doesn’t lead to good results, which is not the same as saying that the mean is not in the typical set (though they are both true). After all, in the well-separated case initialization at the centers *does* work, even though centers are not in the typical set. Here, our purpose is to point out that an intuitive way to initialize the Langevin (which we personally considered as an alternative to initialization from samples) doesn’t lead to good results in the general case.
> > >
> > >
> > > $\textbf{This intuitive  [of initializing at the empirical distribution] natural idea (dismissed as too obvious/unrealistic in [1] sec 1.2) is what is formalized in the paper.}$
> > >
> > > We respectfully disagree that the idea of initializing at the empirical distribution has been raised in the reference [1], or that this idea has been dismissed as too obvious.
> > > We are familiar with reference [1] (we believe you mean Lee, Holden, Andrej Risteski, and Rong Ge. (2018) Beyond Log-concavity: Provable Guarantees for Sampling Multi-modal Distributions using Simulated Tempering Langevin Monte Carlo (neurips.cc) though although there are multiple different references labeled [1] in your response). We have carefully reread section 1.2 in that paper and found no mention of the Langevin initialized at the empirical distribution. We wonder if you are referring to the following sentence “Note that importantly the algorithm does not have direct access to the mixture parameters…(otherwise the problem would be trivial)”. If so, this sentence indeed says an obvious thing, but access to the centers of the Gaussians and the weights $p_i$ of the components gives you a description of the mixture distribution $\mu$, whereas access to samples from $\mu$ doesn’t automatically give you either the center(s) or the weights. If this sentence is not what you have in mind, please provide a direct quote for the claim you’ve made.

---

### Official Review · Reviewer_94gx · 2023-11-05

**Soundness:** 2 fair
**Presentation:** 2 fair
**Contribution:** 3 good
**Rating:** 6
**Confidence:** 2

**Summary:**

This work studies vanilla score matching in the context of mixtures of log-concave distributions, providing a recipe to learn multimodal distributions via vanilla score matching, a procedure that has provably failed in most multimodal settings: (1) data-based initialization for the Langevin Monte Carlo chain; and (2) early stopping of the diffusion. The work demonstrates substantive theoretical developments for the proposed method, along with a few toy examples to illustrate the effectiveness of empirical distribution initialization and early stopping in learning mixture of Gaussians.

**Strengths:**

1. This work aims to tackle a widely accepted issue of vanilla score matching that motivated the usage of annealed langevin dynamics in learning multimodal distributions (diffusion models as a class of generative models) — it has the potential to inspire new algorithms for generative modeling.
2. Code for the toy examples is provided via **Supplementary Material** to facilitate reproducibility.

**Weaknesses:**

1. Experiment results do not appear to be convincing enough:
* The ground truth distribution is not plotted in Figure 1 (b); by comparing to Figure 1(c), it’s not hard to tell that the weight of the component on the right is not learned very well — one has a density around $0.15$, while the other around $0.125$. Meanwhile, there is no numerical computation on the learned mean and variance of each component, and how they compare to the ground truth.
* Similar to the issue in Figure 1, the values of the learned projected mean, variance and weight of each component, and their comparisons with the ground truth distribution, are not reported for the experiment presented in Figure 2.
2. It’s not clear how or where early stopping is proposed as a solution throughout the theoretical development.
3. Some minor issues in writing:
* Page 1, first bullet point of positive aspects: “a simple closed form solution <when> the class of models”
* Page 3, the line above **Theorem 1**: “samplling”
* Page 3, strange expression: “Note in particular that we have can draw as many samples as we like”. Perhaps remove “have”?

**Questions:**

1. Could the authors comment on how the contributions in this work might help in improving diffusion models as a class of generative models?
2. Can optimization techniques such as exponential decay learning rate schedule achieve similar results as early stopping, in the context of estimating vanilla score function of a multimodal distribution?

---

> ### Author Response · Authors · 2023-11-15
> **Rebuttal to reviewer 94gx**
>
> We thank you for your feedback. Below are our responses to your comments and questions.
>
> "Experiment results do not appear to be convincing enough:..."
>
> The distributions do not _exactly_ match since we only initialized with only 40 samples. This is a practical choice for the purpose of making a useful visualization to explain the result, since if we use a very large number of samples then eventually figure 1(a) will perfectly match the ground truth as well since it is a 1-dimensional problem. Of course, the theory guarantees that if we use more data the estimator of the ground truth is consistent. Our method *does not* explicitly estimate a mean and variance of each component (i.e. it does not explicitly try to fit a mixture of 2 gaussians to the data), but instead it attempts to estimate the distribution as a density (and the distribution it outputs is not a mixture of 2 gaussians, only an approximation). This is why we plotted the densities instead of giving the parameter error.
>
> "Similar to the issue in Figure 1, the values of the learned projected mean, variance and weight of each component, and their comparisons with the ground truth distribution, are not reported for the experiment presented in Figure 2.
> It’s not clear how or where early stopping is proposed as a solution throughout the theoretical development."
>
> Please see the reply to question 2 below.
>
> Answer to your questions:
> 1. "Could the authors comment on how the contributions in this work might help in improving diffusion models as a class of generative models?"
>
> Our work verifies empirical findings that data-based initialization is helpful for the vanilla Langevin algorithm, showing that this is also a theoretically valid way to deal with multimodality. (As in e.g. cited paper of Xie et al ‘16). In general, it can be difficult to predict the ultimate benefits of building our theoretical understanding. One reason to care about methods which use data-based initialization like contrastive divergence is that they are approximate to Maximum Likelihood, which we know from fundamental results in statistics is the asymptotically optimal way to fit a model as the number of samples goes to infinity. So it wouldn’t be too surprising if these kinds of methods are more data efficient in some applications — this is a very interesting question, but also quite out of scope of the current work which is focused on the theory.
>
> 2. "Can optimization techniques such as exponential decay learning rate schedule achieve similar results as early stopping, in the context of estimating vanilla score function of a multimodal distribution?"
>
> In our paper when we talk about early stopping, we are not talking about in the context of estimating the vanilla score function. Our analysis works with any learning procedure for the vanilla score function, as long as at the end of the day it is accurately learned (so decaying learning rate schedules, early stopping, etc. can all be used in the training of the score matching network). Instead, when we are talking about early stopping we are saying that the Langevin sampler is stopped, rather than run for an arbitrarily long time. This is what is illustrated in Figure 1.

---

### Meta-Review · Area_Chair_yUvD · 2023-12-05

**Metareview:**

This paper investigates the application of Langevin Monte Carlo with vanilla score matching for provably sampling from target distributions comprising mixtures of log-concave distributions. The authors demonstrate that, unlike recent approximate sampling works relying on stronger Log-Sobolev inequality (LSI) assumptions for non-log-concave distributions, their analysis establishes LSVI in cases where mixture components are close together. They show that utilizing data from the target distribution for initialization yields robust coverage even with distant clusters. The problem addressed is significant in the ML community, and the paper is well-written, presenting a novel two-step analysis framework. The majority of reviewers recommend acceptance, though one expresses dissatisfaction with the limited empirical comparison. While I acknowledge the paper's theoretical focus, the authors are encouraged to discuss alternative analysis frameworks' validity under non-idealized assumptions. Additionally, I urge the authors to incorporate the userful discussion with the reviewers in the response to the final version of their paper, such as remarks on sample complexity, discussions with more recent approximate sampling techniques, and inclusion of relevant references.

**Justification For Why Not Higher Score:**

Some clarity and discussion are lacking and needs to be added.

**Justification For Why Not Lower Score:**

The theoretical results presented in this paper are novel and of significant interest to the ML community.

---

### Decision · Program_Chairs · 2024-01-16

Accept (poster)